# Immune resilience despite inflammatory stress promotes longevity and favorable health outcomes including resistance to infection

Some people remain healthier throughout life than others but the underlying reasons are poorly understood. Here we hypothesize this advantage is attributable in part to optimal immune resilience (IR), defined as the capacity to preserve and/or rapidly restore immune functions that promote disease resistance (immunocompetence) and control inflammation in infectious diseases as well as other causes of inflammatory stress. We gauge IR levels with two distinct peripheral blood metrics that quantify the balance between (i) CD8+ and CD4+ T-cell levels and (ii) gene expression signatures tracking longevity-associated immunocompetence and mortality-associated inflammation. Profiles of IR metrics in ~48,500 individuals collectively indicate that some persons resist degradation of IR both during aging and when challenged with varied inflammatory stressors. With this resistance, preservation of optimal IR tracked (i) a lower risk of HIV acquisition, AIDS development, symptomatic influenza infection, and recurrent skin cancer; (ii) survival during COVID-19 and sepsis; and (iii) longevity. IR degradation is potentially reversible by decreasing inflammatory stress. Overall, we show that optimal IR is a trait observed across the age spectrum, more common in females, and aligned with a specific immunocompetence-inflammation balance linked to favorable immunity-dependent health outcomes. IR metrics and mechanisms have utility both as biomarkers for measuring immune health and for improving health outcomes.

Why do individuals manifest such wide differences in lifespan, health status across age, and susceptibility to infectious diseases? One possibility is that variations in an immune trait contribute to these differences. Given that infections are among the most impactful environmental factors that shape the human genome, optimal host responses to these microbial drivers of natural selection may have played a role in increasing longevity[1]. Hence, immune mechanisms may have evolved based on conferred resistance to the ancestral burden of inflammatory stress associated with infectious diseases.

Resistance mechanisms could include higher immunocompetence and prevention of uncontrolled inflammation. In contemporary times, these infection-resistance mechanisms may confer advantages for a lower comorbidity burden and longevity. Additionally, given the importance of immunocompetence for maternal and fetal health, it is conceivable that the immunologic trait associated with resistance to both infections and premature death may have evolved more prominently in females. This sex bias could provide a basis for the observation that females exhibit advantages for immunocompetence and

✉ e-mail: ahujas@uthscsa.edu

longevity[2–5]. For these reasons, we envisaged an immunologic trait with three advantages: longevity, a lower comorbidity burden, and resistance to infections.

Our hypothesis regarding the identity of this advantageous trait is immunologic resilience (IR). We define optimal IR as the capacity to preserve and/or rapidly restore immune functions that promote disease resistance and longevity (immunocompetence), as well as control inflammation during acute, repeated, or chronic immune (antigenic) stimulation associated with inflammatory stressors (e.g., infections or autoantigens)[6]. IR is rooted in the principle that repeated inflammatory (antigenic) exposures are inevitable throughout life, necessitating allostatic processes that mediate adaptation, ideally returning immunocompetence and inflammation to optimal or pre-exposure levels (Fig. 1a, b). With this definition, optimal IR is linked to a conjoined high immunocompetence (IC)-low inflammation (IF) state designated as $IC^{high}$-$IF^{low}$ (Fig. 1a). In contrast, the failure to preserve optimal IR during antigenic exposures and/or rapidly restore IR following such exposures results in suboptimal or nonoptimal IR linked to worse IC-IF states (Fig. 1a). In this framework, suboptimal and nonoptimal IR indicate incomplete or unsuccessful immune allostasis, respectively (i.e., impaired adaptation to inflammatory stress). Hence, a shift from an optimal to suboptimal/nonoptimal IR status associates with a corresponding shift from an $IC^{high}$-$IF^{low}$ to an $IC^{low}$-$IF^{high}$ status (Fig. 1a). In this model, each new antigenic challenge may be met with ever-lower immunocompetence and ever-higher inflammation (i.e., lower IR levels; Fig. 1c), predisposing to increased risk of disease acquisition and severity, as well as mortality.

With this framework (Fig. 1a–c), we envisaged a sequence by which individuals of similar age and sex manifest differences in IR levels, as well as a basis for why some individuals manifest a decline in IR with age. At any age, IR in most individuals experiencing an antigenic exposure will transiently erode/degrade, leading to a temporary $IC^{low}$-$IF^{high}$ state (Fig. 1b). Some persons may resist this degradation or rapidly reconstitute IR to pre-exposure levels. Hence, we envisaged two IR phenotypes. The first is the IR erosion-resistant phenotype signifying successful immune allostasis leading to preservation and/or rapid restoration of optimal IR correlating with $IC^{high}$-$IF^{low}$ (Fig. 1a, c). The second is the IR erosion-susceptible phenotype signifying incomplete/unsuccessful allostasis leading to suboptimal/nonoptimal IR (Fig. 1a, c). Age serves as a proxy, albeit imperfect, for antigenic exposures. Hence, in individuals with the IR erosion-susceptible phenotype, IR may erode with the accumulation of antigenic exposures over lifespan (Fig. 1c). However, some older individuals resist IR erosion (IR erosion-resistant phenotype) (Fig. 1c). In contrast, some younger individuals may exhibit degraded IR similar to that seen with advanced age (IR erosion-susceptible) (Fig. 1c). For these reasons, the lower immune status often observed with age may be driven by two co-existing mechanisms: one is dependent on age (e.g., due to cellular senescence), while the other is associated with incomplete/unsuccessful immune allostasis at any age (age-independent). The latter is the focus of the current research.

Here, to test these concepts, we evaluate IR metrics in individuals represented in varied, well-defined infectious and non-infectious models of acute, repetitive, and chronic immune stimulation. Prospective cohorts include adults in whom the impact of IR status on health outcomes and lifespan could be quantified, after controlling for age, sex, and/or level of immune stimulation (Fig. 1a; Supplementary Fig. 1). We show that preservation of optimal IR (IR erosion-resistant phenotype) is more prevalent in females and associates with advantages for superior immunity-dependent health outcomes such as longevity and resistance to infection risk/severity (Fig. 1a). Conversely, suboptimal/nonoptimal IR predisposes to inferior outcomes and, while more prevalent in males and the elderly, occurs even among younger individuals. These findings have implications for risk

stratification of immune health across the age spectrum, as well as improving health outcomes.

## Results

### IR metric: immune health grades (IHGs) tracking CD8-CD4 profiles

We previously developed two peripheral blood metrics of IR (Fig. 2a)[6]. The first metric was Immune Health Grades (IHGs) I to IV, which reflect the relative proportions of $CD8^+$ and $CD4^+$ T-cell counts that is not inferable through assessments of these two markers or the CD4:CD8 ratio alone (Fig. 2b). IHG-I was assigned as an indicator of optimal IR, as we previously found that preservation of IHG-I during infection with SARS-CoV-2 and HIV was associated with resistance to severe COVID-19 and AIDS[6]. The IHGs were derived by co-indexing the CD4:CD8 T-cell ratio and the $CD4^+$ T-cell count at the indicated cutoffs (Fig. 2b). The basis for deriving IHGs and why they are less-confounded metrics of immune status than the conventional metrics of the $CD4^+$ T-cell count or CD4:CD8 T-cell ratio have been discussed previously[6] and further expanded (Supplementary Note 1).

The cutoffs for the IHGs were based on two principles: (i) a CD4:CD8 ratio value of less than unity (<1.0) is a mathematical representation of or proxy for higher $CD8^+$ T-cell counts that are uncompensated for by higher $CD4^+$ counts, and (ii) 800 $CD4^+$ cells/$mm^3$ approximated the median $CD4^+$ T-cell count in 16,126 HIV-seronegative (HIV−) persons (Supplementary Table 1)[6–8]. IHG-I and IHG-II track relatively lower $CD8^+$ T-cell levels with (IHG-I) or without (IHG-II) higher $CD4^+$ counts, whereas IHG-III and IHG-IV track relatively higher $CD8^+$ T-cell levels with (IHG-III) or without (IHG-IV) higher $CD4^+$ counts (Fig. 2b). Thus, the IHGs are not strata or categories of ratio values or $CD4^+$ counts. These distinctions have clinical relevance: for example, extensive $CD4^+$ T-cell lymphopenia (<200 cells/$mm^3$) may occur with IHG-II during acute COVID-19 vs. with IHG-IV during advanced HIV disease[6]. Based on the relative proportions of $CD8^+$ vs. $CD4^+$ counts, we assigned IHG-I and IHG-II to signify CD8-CD4 equilibrium states, whereas, IHG-III and IHG-IV to signify CD8-CD4 disequilibrium states.

### IR metric: survival- and mortality-associated gene expression signatures

The second metric of IR was transcriptomic (gene expression) profiles that predict survival or mortality (Fig. 2a, c–d). We previously identified a suite of peripheral blood transcriptomic signatures that were associated with COVID-19 outcomes (hospitalization, survival; Supplementary Fig. 2a)[6]. A subset of signatures ($n = 10$) predicted survival during acute COVID-19 as well as aging in participants of the Framingham Heart Study (FHS) without COVID-19 (Supplementary Fig. 2a; Supplementary Information Section 8.2). Here, we focused on the signatures that provided the highest prognostication (by Akaike information criteria) for survival and mortality in both cohorts, after controlling for age and sex (Supplementary Information Section 8.2). These signatures were termed survival-associated signature (SAS)−1 and mortality-associated signature (MAS)−1 (Fig. 2a, c; Supplementary Fig. 2a).

Higher expression of SAS-1 ($SAS-1^{high}$) likely tracked $IC^{high}$, as this signature comprised IC-related genes (e.g., *CCR7*, *IL7R*; Fig. 2d) and higher baseline expression of SAS-1 associated with lower all-cause mortality hazards during acute COVID-19 as well as lower all-cause mortality hazards in the FHS (Fig. 2c, d). Higher expression of MAS-1 ($MAS-1^{high}$) likely tracked $IF^{high}$, as this signature comprised IF-related genes (e.g., *C5AR1*, *MYD88*) and higher baseline expression of MAS-1 associated with *higher* all-cause mortality hazards during acute COVID-19 as well as in the FHS (Fig. 2c, d). Congruently, in the FHS, incrementally higher baseline levels of SAS-1 or MAS-1 predicted progressively longer and shorter lifespans, respectively (Fig. 2c; Supplementary Fig. 2b). In this study, the additive effects of IC and IF status were proxied by four SAS-1/MAS-1 profiles (based on higher and

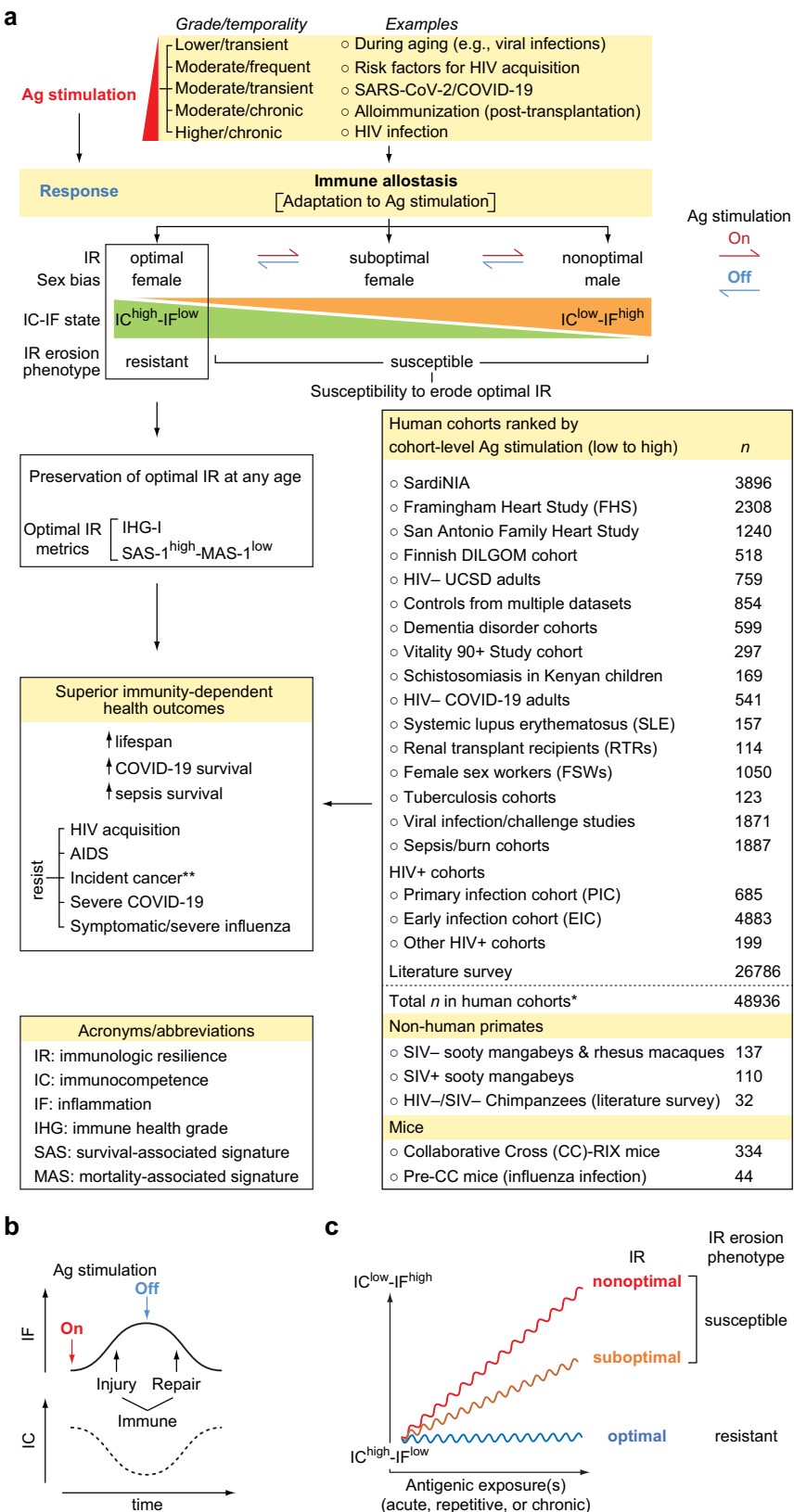

**Nature Communications** | (2023)14:3286

lower levels of these signatures). Because the combination of SAS-1$^{high}$ (IC$^{high}$) and MAS-1$^{low}$ (IF$^{low}$) was predicted to have the best longevity advantage, the combined SAS-1$^{high}$-MAS-1$^{low}$ (IC$^{high}$-IF$^{low}$) profile was considered an indicator of optimal IR that is overrepresented in individuals with IHG-I (Fig. 2e).

## Cohorts: models for low-, moderate- and high-grade antigenic stimulation

Metrics of IR were evaluated in human cohorts ($n = 48,936$ subjects/ samples) that served as proxies for low-, moderate-, and high-grade antigenic stimulation (Fig. 1a; details in Supplementary Data 1).

**Fig. 1 | Study concepts and cohorts. a** Immunologic resilience (IR) erosion-resistant and erosion-susceptible phenotypes and predicted outcomes. Phenotypes are defined by sexually dimorphic immune allostasis responses to antigenic (Ag) stimulation that links high or low immunocompetence (IC) and inflammation (IF) states to the indicated immunity-dependent health outcomes. Possible sources of Ag stimulation, outcomes, and cohorts/datasets are depicted. Arrows depict induction (red) and reversibility (blue) of IR states with Ag stimulation on and off respectively. *number of samples studied; **incident cancer in immunocompromised renal transplant recipients. RIX, recombinant inbred inter-cross. DILGOM dietary, lifestyle and genetic determinants of obesity and metabolic syndrome, SIV simian immunodeficiency virus. Abbreviations frequently used in this study are noted. *n*, number of individuals and/or samples studied. **b** Model. IC and IF changes during an immune injury-repair cycle in response to a single instance of Ag stimulation. **c** Ordinate, IC-IF states associated with the degree of deviation from optimal IR during increased Ag stimulation in individuals with the IR erosion-resistant versus -susceptible phenotypes. The alignment of optimal, suboptimal, and nonoptimal IR status with phenotypes is noted. Abscissa, time window overlapping with a period of increased Ag stimulation that could be acute, chronic, or repetitive irrespective of age. In this model, since age is a proxy, albeit imperfect, for antigenic experience, individuals with the IR erosion-susceptible phenotype may manifest suboptimal or nonoptimal IR with advancing age.

Representative examples were (i) low-grade antigenic stimulation in HIV− persons accrued in aging cohorts (younger adults to >90 years) and included the SardiNIA aging cohort[9] ($n = 3896$), the Offspring subset of the FHS[10,11] ($n = 2308$), San Antonio Family Heart Study[12] ($n = 1240$), the Finnish DILGOM cohort[13] ($n = 518$), and nonagenarians in the Vitality 90 + Study[14] ($n = 297$); (ii) HIV− cohorts with sources of moderate-grade antigenic stimulation, e.g., SARS-CoV-2 infection[6] ($n = 541$), alloimmunization in renal transplant recipients (RTRs)[15] ($n = 114$), autoantigen in systemic lupus erythematosus (SLE)[16] ($n = 157$), and risk factors for acquiring HIV[17] (e.g., in female sex workers [FSWs], $n = 1050$); and (iii) high-grade antigenic stimulation associated with HIV viremia in primary/early HIV infection cohorts[7,8] ($n = 5568$) (Fig. 1a; Supplementary Fig. 1; Supplementary Data 1; Supplementary Data 2; Supplementary Data 3; Supplementary Data 4; Supplementary Data 5; Supplementary Data 6; Supplementary Data 7).

Evolutionary conservation of IR phenotypes was evaluated in 279 nonhuman primates[18–20] and 334 Collaborative Cross-RIX mice, a large panel of recombinant, inbred intercrosses (RIX) designed for complex trait analysis[21] (Fig. 1a). Seropositivity for cytomegalovirus (CMV) has been associated with mortality and age-associated diseases;[22–25] the association between IR status and CMV serostatus was examined in several cohorts (COVID-19, RTRs, and HIV− controls from the University of California at San Diego). The distribution and association of CD8-CD4 disequilibrium grades IHG-III or IHG-IV with health outcomes was examined via a large-scale literature survey of 26,786 humans (Fig. 1a; Supplementary Table 2). Study design features that mitigated confounding are discussed (Supplementary Note 2).

## Overall study guide: study phases 1 to 4
To test the proposed framework (Fig. 1a), we conducted a four-phase study schematized in Fig. 2e. In study phases 1 and 3, we determined whether indicators of optimal IR, namely IHG-I and a transcriptomic proxy for an IC$^{high}$-IF$^{low}$ status (SAS-1$^{high}$-MAS-1$^{low}$ profile) represent primordial states that are eroded to a non-IHG-I grade or non-IC$^{high}$-IF$^{low}$ status in settings of increased antigenic stimulation (Fig. 2e). In study phases I and 3, we also determined the reconstitution patterns of the primordial states following cessation and/or mitigation of antigenic stimulation. In study phases 2 and 3, we examined whether, after controlling for age, resistance to erosion of IR (IR erosion-resistant phenotype) is associated with superior immunity-dependent health outcomes, including longevity (Fig. 2e). In study phase 4, we inquired whether, after controlling for age, the IR erosion-resistant phenotype is linked to immunologic traits typically associated with higher IC and lower IF. For simplicity, the IHGs were evaluated in study phases 1, 2, and 4, while SAS-1/MAS-1 profiles were evaluated in study phase 3.

## Study phase 1: Shifts from IHG-I to non-IHG-I grades across lifespan
In varied cohorts, IHG-I, IHG-II, IHG-III, and IHG-IV tracked the CD8-CD4 profiles of CD8$^{lower}$-CD4$^{highest}$, CD8$^{lowest}$-CD4$^{lower}$, CD8$^{highest}$-CD4$^{higher}$, CD8$^{higher}$-CD4$^{lowest}$, respectively (Fig. 2b; Supplementary Data 3; Supplementary Note 1). In younger participants of the 3893-person,

community-based, HIV− SardiNIA cohort (median age: 49 [IQR: 36-62] years; 42.8% males), IHG-I was the most common and IHG-II was the second most common grade; IHG-III and IHG-IV were less common (<5% in individuals younger than 50 years) (Fig. 2f). Age was associated with a steady decrease in the prevalence of IHG-I (%IHG-I) and reciprocal increases in %IHG-II as well as %IHG-III or %IHG-IV (Fig. 2f, left).

## Study phase 1: Shift from IHG-I to non-IHG-I grades is more common in males
While the progressive shift in %IHGs across age was similar in both sexes, the likelihood of having a non-IHG-I grade vs. preserving IHG-I was more common in males than females (Fig. 2f, g). Across age, the odds of having IHG-I vs. a non-IHG-I grade was greater in females compared with males (Fig. 2g, leftmost). The odds of having IHG-II vs. IHG-I, or IHG-III or IHG-IV vs. IHG-I, increased with age; however, these odds were greater in males than females (Fig. 2g). The odds of having IHG-II vs. IHG-III or IHG-IV did not change significantly with age; however, females compared with males were more likely to have IHG-II than IHG-III or IHG-IV (Fig. 2g, rightmost).

## Study phase 1: IHG-I, an indicator of the IR erosion-resistant phenotype
The IHG distributions during aging (Fig. 2f, g) conveyed four inferences (Fig. 2h). First, across lifespan, there is a strong preference to preserve grades tracking CD8-CD4 equilibrium (IHG-I or IHG-II) than disequilibrium (IHG-III or IHG-IV) states. Second, females compared with males are more likely to preserve CD8-CD4 equilibrium grades IHG-I or IHG-II, including post-menopause. Third, IHG-I is the primordial IHG from which the other IHGs emerge during aging. For this reason, preservation of IHG-I at any age was assigned as an indicator of the IR erosion-resistant phenotype and optimal IR (Fig. 2h). Conversely, having a non-IHG-I grade was assigned as an indicator of the IR erosion-susceptible phenotype and suboptimal or nonoptimal IR (Fig. 2h). Fourth, older and younger persons with the same IHG may share similar immunologic attributes (IR-associated traits after controlling for age); in contrast, a separate set of immunologic traits may track older vs. younger persons preserving IHG-I or IHG-II, the two most-common grades across age (age-associated traits). The validity of these inferences and IHG assignments were tested as described below. In the context of acute antigenic stimulation, the IR erosion phenotypes as gauged by the IHG metric were evaluated in two infection models: schistosomiasis and SARS-CoV-2 infection.

## Study phase 1: IHG shifts during schistosomiasis
In Kenyan children with schistosomiasis, the level of antigenic stimulation was proxied by urinary egg counts (Fig. 3a). Akin to younger SardiNIA participants (Fig. 2f), among children without schistosomiasis, the first and second-most prevalent IHGs were IHG-I and IHG-II, respectively (Fig. 3a). Progressively higher urinary egg counts of *Schistosoma haematobium* were associated with incrementally lower % IHG-I and increases in %IHG-IV (Fig. 3a). Signifying the IR erosion-resistant phenotype, in the higher egg count stratum (≥500 eggs/mL), some children preserved IHG-I. Signifying the IR erosion-susceptible

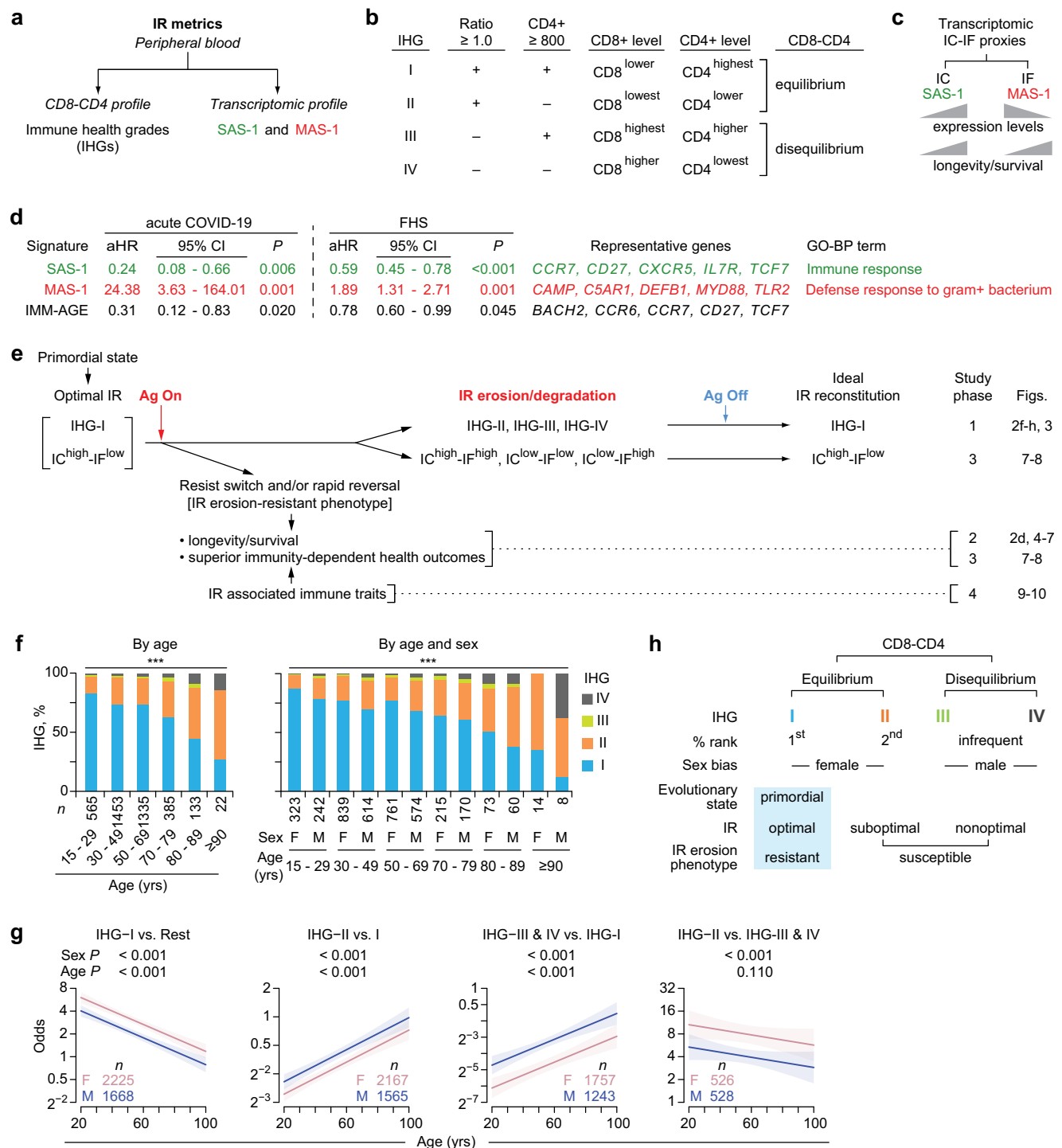

Fig. 2 | **Metrics of immunologic resilience (IR) and association of the immune health grade (IHG) metric in the SardiNIA cohort. a** IR metrics. IHGs are described in panel (**b**). Two gene expression (transcriptomic) signatures termed survival-associated signature-1 (SAS-1) and mortality-associated signature-1 (MAS-1) are prognosticators of survival and mortality, respectively, after controlling for age and sex. **b** CD8-CD4 profiles by IHGs and cutoffs of the CD4:CD8 T-cell ratio and CD4⁺ T-cell counts (cells/mm³) used to derive IHGs. **c** Predicted associations of expression levels of transcriptomic proxies for immunocompetence (IC) and inflammation (IF) with longevity/survival. **d** Hazard ratios adjusted for age and sex (aHR) with 95% confidence intervals (CIs) for the indicated gene signatures associated with all-cause mortality in the acute COVID-19 cohort (90-day mortality) and Framingham Heart Study (FHS; survival over 9 years since first sampling). Representative genes and gene ontology biological process (GO-BP) terms are shown. +,

positive. **e** Model and study phases 1 to 4. Far right, figures specific to the outcomes are noted. During antigenic (Ag) stimulation, preservation of and/or rapid restoration of a primordial status defined by IHG-I and a higher IC and lower IF (IC^high^-IF^low^) state is associated with superior immunity-dependent health outcomes, including a longevity/survival advantage. **f** Distribution of IHGs in the HIV⁻ SardiNIA cohort. ***P < 0.001. F, female; M, male. **g** Odds of having the indicated IHG (with 95% confidence bands) by age and sex in the SardiNIA cohort. *P*, for differences in odds by sex and age are depicted. Rest, all other IHGs. **h** Features of CD8-CD4 equilibrium and disequilibrium grades. Assignment of IHG-I as an indicator of the IR erosion-resistant phenotype. A non-IHG-I grade signifies the IR erosion-susceptible phenotype. Two-sided tests were used. Statistics are outlined in Supplementary Information Section 11.3.2., *P* values are in Supplementary Data 14, and Source data are provided as a Source Data file.

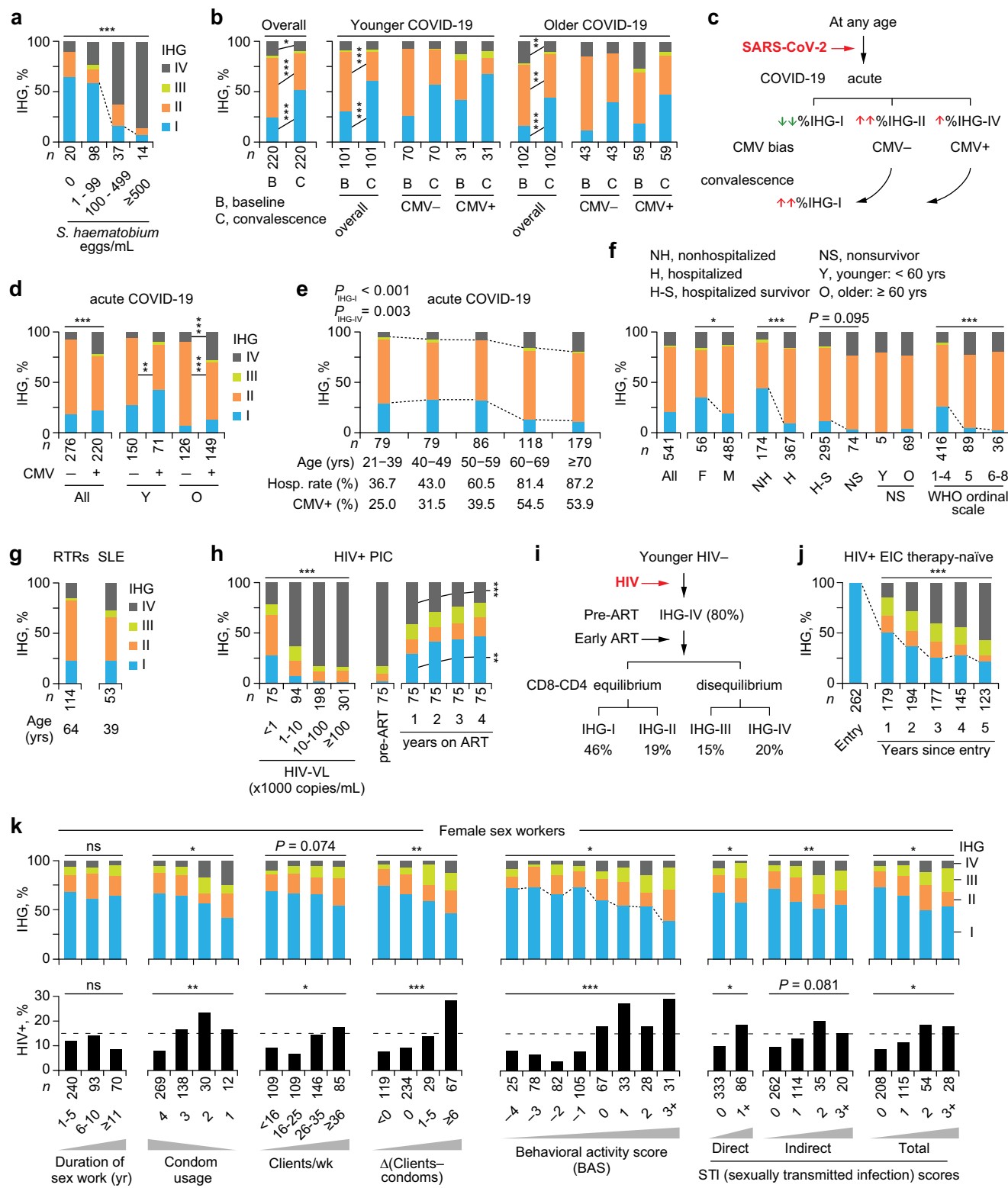

phenotype, a greater proportion of children in the lower (1–99 eggs/mL) vs. the zero-egg count stratum had IHG-IV. Higher egg counts were associated with elevated peripheral blood levels of CD25$^+$CD127$^-$ CD4$^+$ T-cells (Supplementary Fig. 2c), a marker of regulatory CD4$^+$ T cells[26]. These findings highlight two points: (i) erosion of IHG-I (IHG-I → IHG-IV) was proportionate to levels of immune (antigenic) stimulation, and (ii) among children with comparable levels of antigenic stimulation, there was interindividual variation in susceptibility to IHG-I erosion.

## Study phase 1: IHG shifts during acute and convalescent COVID-19

By comparing IHG distribution patterns at presentation in HIV-seronegative patients with COVID-19 (baseline) vs. during recovery (convalescence), we determined whether erosion/degradation followed by age-appropriate reconstitution of IHG-I occurs during the acute and convalescent phases of COVID-19, respectively. These comparisons revealed that, (i) irrespective of age or CMV serostatus, acute COVID-19 was associated with lower %IHG-I at baseline vs.

**Fig. 3 | Shift from immune health grade (IHG)-I to non-IHG-I grades in settings associated with increased antigenic stimulation. a** %IHG (prevalence) in Kenyan children according to *Schistosoma haematobium* egg counts in urine. **b–f** Acute COVID-19 cohort. **b** %IHGs at baseline vs. convalescence (paired): overall, by age and cytomegalovirus (CMV) serostatus. **c** IHG degradation and reconstitution during COVID-19 by CMV serostatus. Baseline %IHG-II and %IHG-IV, higher during COVID-19, is overrepresented in CMV− and CMV+ patients, respectively. **d** Baseline %IHGs by CMV serostatus: overall, and age. **e** Baseline %IHGs, hospitalization (hosp.) rates, and CMV seropositivity rates by age strata. $P_{IHG-I}$ and $P_{IHG-IV}$, for the change in %IHG-I vs. other grades and %IHG-IV vs other grades across age strata, respectively. **f** Baseline %IHGs overall and stratified by sex and outcomes. F female, M male. Disease severity status defined by World Health Organization (WHO) ordinal scale: 1-4 [mild]; 5 [moderate]; 6–8 [severe]. **g** %IHGs in renal transplant recipients (RTRs) and patients with systemic lupus erythematosus (SLE). **h** Primary HIV infection cohort (PIC). Left, Baseline %IHGs by HIV viral load (HIV-VL). Right, %

IHGs before (pre-antiretroviral therapy [ART]) and during 4 years of ART. **i** Schema for panel h with %IHGs in year 4 of ART. **j** %IHGs during 5 years of therapy-naïve HIV disease course in the subset with IHG-I at entry into the early infection cohort (EIC). **k** Baseline %IHG (top) and subsequent HIV seroconversion rates (bottom) in female sex workers who were HIV− at baseline stratified according to behavioral and biological (sexually transmitted infection [STI]) risk factors. Behavioral risk factors: duration of sex work, condom usage (1, never; 2, <50%; 3, ≥50%; and 4, always), clients/week, and Δ (clients − condoms) (the difference between the number of clients/wk and condoms used/wk). Behavioral acitivty score (BAS) is the sum of scores of these risk factors. STI scores were derived based on direct and indirect indicators of STI. *$P < 0.05$; **$P < 0.01$; ***$P < 0.001$; ns nonsignificant. Two-sided tests were used. Statistics are outlined in Supplementary Information Section 11.3.3., $P$ values are in Supplementary Data 14, and Source data are provided as a Source Data file.

convalescence, with preferential emergence of IHG-II and less so of IHG-IV (Fig. 3b, c), and (ii) IHG distributions reconstituted during convalescence mirrored those of age-matched SardiNIA participants (compare Fig. 3b - convalescence vs. Fig. 2f).

CMV serostatus influenced the nature of the IHGs that emerged during COVID-19 (Fig. 3b–d). Presentation with IHG-IV and IHG-III was more common in CMV+ vs. CMV− patients, especially older CMV+ patients (Fig. 3d). In contrast, presentation with IHG-II was more common in CMV− patients. The higher %IHG-IV in older CMV+ patients paralleled our finding that rates of both %IHG-IV and CMV seropositivity increased with age (Fig. 3e). Since CMV seropositivity rates increase with age (Fig. 3e; Ref. 27), during acute COVID-19, older CMV+ persons are predisposed to present with IHG-IV and, less commonly, IHG-III (Fig. 3b). The association between CD8-CD4 disequilibrium grades IHG-III or IHG-IV and high rates of CMV seropositivity was confirmed in persons without COVID-19 (Supplementary Note 3). However, development of IHG-IV during acute COVID-19 can be temporary, as reflected by our finding that, among older CMV+ persons, % IHG-IV was lower during convalescence vs. baseline (Fig. 3b, rightmost).

The IHG distribution patterns in cohorts of persons without (SardiNIA) or with acute COVID-19 showed three similarities. First, while %IHG-I was lower in persons with vs. without acute COVID-19, in both cohorts, %IHG-I was higher in younger persons and declined progressively with age, with reciprocal increases in %IHG-II and %IHG-IV (compare Fig. 2f vs. 3e). In the COVID-19 cohort, decreases in %IHG-I with age paralleled an increase in hospitalization rates (Fig. 3e). Second, within each age stratum of both cohorts, some persons resisted erosion of IHG-I (Figs. 2f, 3e). In the COVID-19 cohort, preservation of IHG-I was associated with better outcomes, as %IHG-I was greater in nonhospitalized vs. hospitalized patients, those with mild disease severity status (indexed by WHO ordinal scale[28] of 1–4), and survivors (Fig. 3f). Third, females preserved IHG-I to a greater extent than males (Figs. 2f–g, 3f).

Taken together, these findings convey two key inferences. First, the IHG at presentation with acute COVID-19 is dependent on five factors: age, sex, CMV serostatus, the IR erosion phenotype, and the IHG present before COVID-19, as IHG-I during acute COVID-19 is mostly possible in persons who had the same grade before SARS-CoV-2 infection. Thus, persons preserving IHG-I before and at presentation with acute COVID-19 have the IR erosion-resistant phenotype. Second, erosion of IHG-I can be temporary, and even older persons can retain the capacity to reconstitute IHG-I during convalescence.

**Study phase 1: IHG shifts during chronic antigenic stimulation**
In the context of chronic antigenic stimulation, the IR erosion phenotypes were examined in three settings: HIV– RTRs, HIV– SLE patients, and those with primary/early HIV infection. Compared with age-matched SardiNIA participants (Fig. 2f), %IHG-I was lower, whereas %

IHG-II and %IHG-IV were higher among RTRs [$n = 114$; median age: 64 (IQR: 56–71) years; Fig. 3g] and younger individuals with SLE [$n = 53$; median age: 39 (IQR: 30–51 years); Fig. 3g].

Two HIV infection cohorts were evaluated: the primary infection cohort (PIC) from the University of California at San Diego[7] ($n = 685$; median [IQR] age, 33 [26–40] years; 96.2% males) and the early infection cohort (EIC) ($n = 4883$; median [IQR] age, 28 [24–34] years; 93.0% males)[8] (Fig. 1a; Supplementary Fig. 1a–b, Supplementary Data 2; Supplementary Data 6). Level of HIV-associated antigenic stimulation was proxied by HIV viral load (HIV-VL). Incrementally higher HIV-VL in PIC participants was associated with progressively lower %IHG-I and increases in %IHG-IV (Fig. 3h, left). However, within each HIV-VL stratum, a small subset preserved IHG-I (IR erosion-resistant phenotype). Suppression of HIV-VL with antiretroviral therapy (ART) resulted in the progressive reconstitution of IHG-I followed by IHG-II; in year 4 of ART, 46% of the cohort had reconstituted IHG-I (Fig. 3h–i). The interval between the estimated date of infection and starting ART was 3.66 (IQR: 2.67–7.55) months. Thus, suppression of HIV-VL during early HIV infection was amenable to reconstitution of the primordial IHG-I, despite a high proportion of patients having IHG-IV (~80%) before ART (Fig. 3h–i).

Untreated HIV infection provided an experimental system to witness the stepwise erosion/degradation of IHG-I. We focused on the elite subset (5.2%) of EIC participants who preserved IHG-I during early infection ($n = 262$; median [IQR] age: 26 [23–32] years) (Fig. 3j). During five years of the therapy-naïve disease course, the capacity to preserve IHG-I decreased, resulting in the emergence of the other grades (Fig. 3j). However, in year 5, nearly 20% continued to preserve IHG-I (Fig. 3j). These findings support IHG-I as the primordial IHG and the capacity to preserve/express the IR erosion-resistant phenotype even in settings of high-grade chronic antigenic stimulation.

**Study phase 1: IHG shifts during repetitive antigenic stimulation**
IR erosion phenotypes in the context of repetitive, moderate-grade antigenic stimulation was examined in 1050 FSWs (Supplementary Fig. 1c; median [IQR] age: 31 [27–37] years). All FSWs were HIV− at presentation (baseline); 127 seroconverted within the study period. The extent of moderate-grade antigenic stimulation was proxied by behavioral (frequency of unprotected sex) and biological [sexually transmitted infection (STI)] risk factors for HIV acquisition. Behavioral risk factors and baseline IHG status were available for 762 FSWs (Supplementary Fig. 3a–b; Supplementary Data 4a). To mitigate confounding attributable to a false-negative HIV seronegative test, the association between baseline IHG and subsequent (incident) HIV seroconversion was restricted to 449 FSWs with at least 2 HIV seronegative tests performed at least 3 months apart (Supplementary Fig. 3a; Supplementary Data 4b). Of these, 53 women subsequently seroconverted (Supplementary Fig. 3a). The median interval between baseline and HIV seroconversion was 4.70 (IQR: 2.40–10.28) years.

At baseline, nearly 60% of FSWs had IHG-I and 19% had IHG-III or IHG-IV. Prevalence of IHGs was similar regardless of the duration of sex work (Fig. 3k, top-leftmost). Overall behavioral risk was quantified by a behavioral activity score (BAS) that is also a proxy for HIV exposure risk (fewer condoms, more clients, more clients than condoms used), scaled from −4 to +5. Biologic risk was quantified by a total STI score; this score was positively associated with the BAS ($r = 0.22$; $P < 0.001$; Supplementary Fig. 3c). An incrementally higher BAS or total STI score was associated with progressively lower %IHG-I and reciprocally higher %IHG-III or %IHG-IV (Fig. 3k, top). Among those without IHG-III or IHG-IV at baseline, a higher baseline BAS was associated with an increased hazard of subsequently developing these grades (Supplementary Fig. 3d). Hence, higher BAS and STI scores were risk factors for having or developing IHG-III or IHG-IV. Congruent data were observed in the 762 HIV− FSWs (Supplementary Fig. 3b).

After baseline measurements, FSWs were provided education and interventions (e.g., condoms) for practicing safe sex. Mitigation of behavioral risk factors (lower BAS) was associated with reconstitution of IHG-I in FSWs who remained HIV− for (i) 10 years with available IHG data within each 2-year interval (Fig. 4a; Supplementary Fig. 4a) and (ii) at least 4 years (Fig. 4b; Supplementary Fig. 4b–d). Reconstitution of IHG-I was attributable to lowering CD8+ counts ($P < 0.001$), as CD4+ counts did not change significantly (Supplementary Fig. 4a, c, d).

Pre- and post-HIV seroconversion IHG data were available on 43 FSWs. Akin to the elite group of individuals accrued during early HIV infection who preserved IHG-I at presentation (Fig. 3j), an elite subset of these 43 FSWs preserved IHG-I post-HIV infection (group 1 Fig. 4c, left; Supplementary Fig. 5). Nearly 30% ($n = 13$) of FSWs had IHG-III or IHG-IV before HIV seroconversion (group 2 Fig. 4c, left), whereas post seroconversion, nearly 75% had IHG-IV (Fig. 4c; Supplementary Fig. 5a). Thus, HIV+ persons may have IHG-III or IHG-IV attributable to two causes of increased antigenic stimulation: risk factors that antedated infection and HIV infection per se.

## Study phase 1: Evolutionary conservation of IHGs in nonhuman species

Sooty mangabeys without and with natural simian immunodeficiency virus (SIV) infection allowed for evaluation of the additive impact of a single (non-SIV) source vs. two (non-SIV and SIV)[18] sources of antigenic stimulation on erosion of IHG-I. Two sources of antigenic stimulation had additive negative effects, as IHG-I was present in only 23% of SIV+ vs. 48% of SIV− sooty mangabeys ($P = 0.001$) (Fig. 4c, right). Akin to humans (Fig. 2f), IHG-I was the primordial IHG in non-human primates, present in nearly 75% of 3- to 5-year-old SIV− sooty mangabeys (Fig. 4d; Supplementary Fig. 6) and 2- to 3-year-old Chinese rhesus macaques (Fig. 4e).

Evolutionary parallels were also observed in the Collaborative Cross-RIX mice[29]. Groups of mice strains categorized into those who manifested relative resistance vs. susceptibility to lethal Ebola virus infection[21]. We examined the IHGs of the uninfected counterparts of these mice strains: %IHG-I was greater in mice strains that survived after Ebola infection; in contrast, %IHG-IV was more common in mice strains that died after infection (Fig. 4f; Supplementary Note 4). Thus, resistance vs. susceptibility to lethal Ebola in mice may partly relate to a genetically associated capacity to preserve IHG-I or develop IHG-IV, respectively, before infection. We identified a human corollary: a single nucleotide polymorphism in the MHC locus (rs2524054-A) that associated with relatively lower levels of CD8+ T-cells[30] may associate with preservation of IHG-I or IHG-II (Supplementary Note 5).

## Study phase 1: Convergent findings across human and nonhuman primate cohorts

Consistent with our model (Fig. 2e), the sum of the findings in study phase 1 were convergent for an association between antigenic stimulation and erosion of optimal IR indexed to IHG-I. The juxtaposition of findings in human vs. nonhuman primate cohorts suggest three evolutionary parallels. First, in both species, IHG-I is the primordial grade from which non-IHG-I grades emerge with increased antigenic stimulation. Second, %IHG-I is higher in female vs. male human and nonhuman primates, and there is a progressive decline in %IHG-I with age (Figs. 2f, 4d–e). Third, akin to FSWs who acquired HIV, sooty mangabeys categorized into those preserving IHG-I after SIV infection (group 1 Fig. 4c) and those with IHG-III or IHG-IV before SIV infection (group 2 Fig. 4c). Overall, CD8-CD4 disequilibrium grades IHG-III and IHG-IV were more frequent in nonhuman primates, present in nearly 69% of SIV+ and 42% of SIV− sooty mangabeys, 52% of SIV− rhesus macaques[19] (Fig. 4c, right; Fig. 4e), and 22% of SIV− chimpanzees ($n = 32$)[20]. Thus, a key evolutionary difference was that IHG-III and IHG-IV were much less frequent in otherwise healthy humans (Fig. 2f) than nonhuman primates. The higher prevalence of IHG-III and IHG-IV in nonhuman primates vs. humans may be attributable to differences in types and levels of antigenic exposures between species and suggests a potential survival benefit for humans to preserve CD8-CD4 equilibrium grades IHG-I or IHG-II vs. disequilibrium grades IHG-III or IHG-IV.

The juxtaposition of findings from HIV− vs. HIV+ cohorts yielded five inferences relevant to understanding the impact of host x environment (antigenic stimulation) interactions on IHG distributions. First, at any age, increased antigenic stimulation induces a shift from IHG-I to non-IHG-I grades. Second, the extent of the deviation or shift from IHG-I is proportionate to the level of antigenic stimulation. For example, seven groups with contrasting host characteristics but relatively lower levels of antigenic stimulation manifested relatively similar IHG distribution patterns with the trifecta of lower %IHG-I, higher %IHG-II, and higher %IHG-III or %IHG-IV: male octogenarians (Fig. 2f); older CMV+ patients with acute COVID-19 (Fig. 3d); RTRs (Fig. 3g); younger individuals with SLE (Fig. 3g); younger HIV+ adults with lower HIV-VL (<1000 copies/mL; Fig. 3h); HIV+ individuals on ART (Fig. 3h); and HIV− FSWs with higher BAS and STI scores (Fig. 3k), including those who subsequently seroconverted (Fig. 4c, left). These similarities across human cohorts have clinical relevance, as they suggest that (i) cohorts with varying host characteristics may comprise individuals with similar levels of immunosuppression linked to a non-IHG-I grade, (ii) immunosuppression may antedate HIV seroconversion, and (iii) development of a non-IHG-I grade may explain why some younger patients with HIV or SLE prematurely manifest immune and clinical features of age-associated diseases[31,32].

Third, reconstitution of IHG-I is possible. For example, in three different contexts [COVID-19 (Fig. 3b), HIV+ patients on ART (Fig. 3h, right), and HIV− FSWs (Fig. 4a, b)], mitigation of antigenic stimulation was associated with reconstitution of IHG-I. Fourth, individuals may have multiple concurrent sources of increased antigenic stimulation; hence, reconstitution of IHG-I may be impaired without mitigation of all sources. Thus, the age-associated erosion of IHG-I to a non-IHG-I grade may be partly attributable to accumulated antigenic experience. Fifth, consistent with our model (Fig. 2e), among persons of similar ages and levels of antigenic stimulation, some individuals resist erosion of IHG-I, i.e., manifest the IR erosion-resistant phenotype. In study phase 2 (below), we examined whether preservation of IHG-I was associated with superior immunity-dependent health outcomes.

## Study phase 2: IR erosion-resistant phenotype and less-severe COVID-19

Juxtaposition of the IHG distribution patterns across age in persons without (Fig. 2f) vs. with (Fig. 3e) acute COVID-19 predicts that based on the IHG status at presentation with COVID-19 and the theoretical IHG status possible before COVID-19, patients may stratify into three groups (Fig. 4g, top). Group A comprises patients presenting with IHG-I; based on the above-noted results (Fig. 3b), most of Group A are predicted to have IHG-I before COVID-19. Group B is a conflated group of individuals presenting with IHG-II or IHG-IV; these grades before

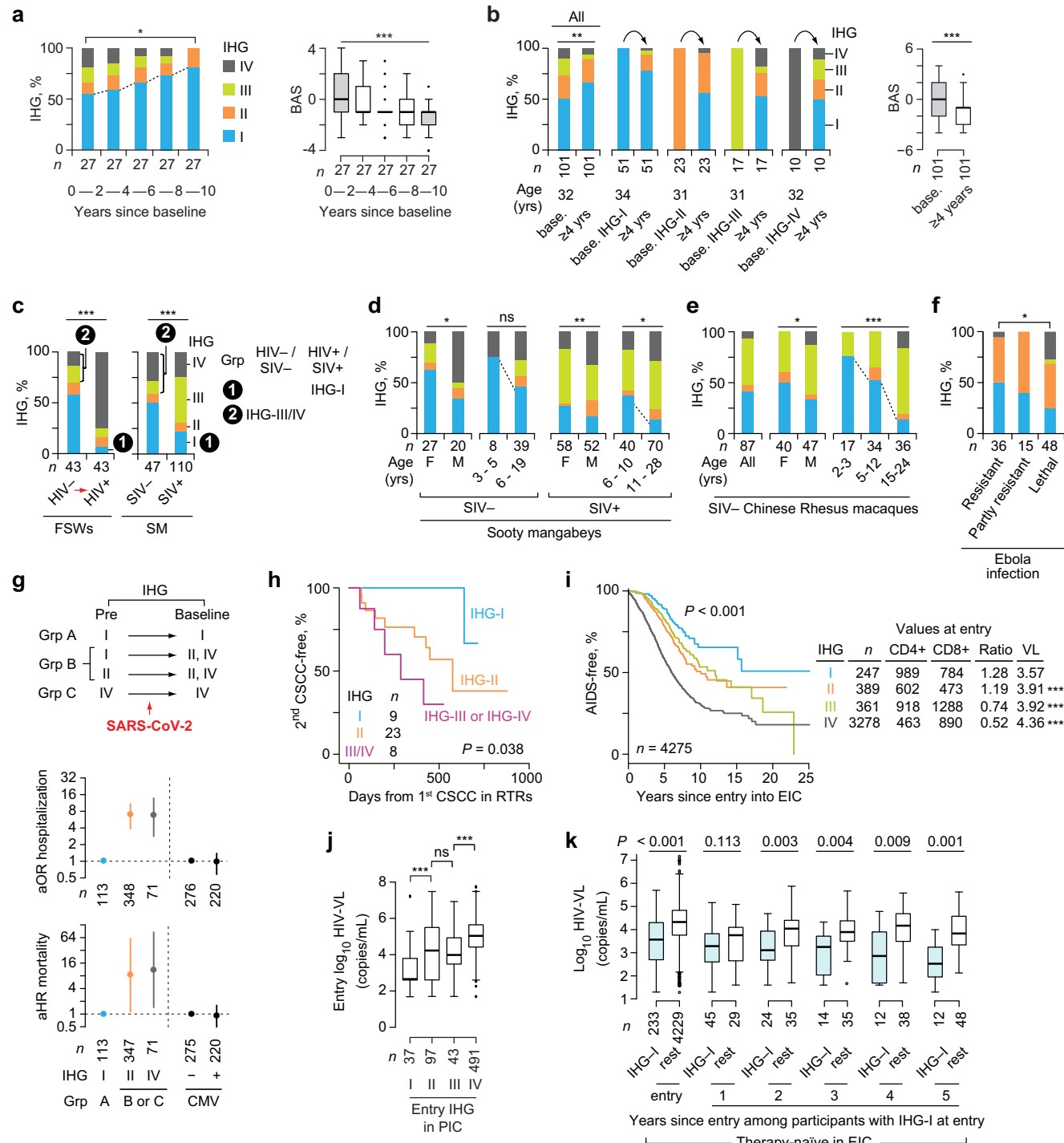

**Fig. 4 | Reconstitution of immune health grade (IHG)-I and associations of IHGs with immunity-dependent health outcomes. a, b** Left, reconstitution of IHG-I in female sex workers (FSWs) who remained HIV− (**a**) during 10-year follow-up and (**b**) for at least 4 years according to the IHG at baseline (base.). Right, behavioral activity score (BAS). **c–f** %IHGs in (**c**) Left, 43 FSWs (paired) pre- and post-seroconversion with HIV. Right, SIV− and SIV+ sooty mangabeys (SM) (unpaired); (**d**) sooty mangabeys by SIV serostatus, sex, and age; (**e**) SIV− Chinese rhesus macaques by sex and age; and (**f**) the uninfected counterparts of Collaborative Cross-RIX mice grouped by outcomes after Ebola infection. **g** Groups based on IHG at baseline and predicted IHG before COVID-19 (top). Adjusted odds ratio (aOR) with 95% confidence interval (CI) for hospitalization (middle), and adjusted hazard ratio (aHR) with 95% CI for all-cause, 30-day mortality (bottom) from two separate models, adjusted by age strata. Model 1, by baseline IHG; and model 2, by CMV serostatus. % IHG-III was low (Fig. 3b) and not included in the models. **h** Time to second

occurrence of cutaneous squamous cell carcinoma (CSCC) by IHG at time of first occurrence of CSCC in renal transplant recipients. **i** Time to AIDS (CDC 1993 criteria) by baseline IHG with median CD4+ and CD8+ counts, CD4:CD8 ratio and HIV viral load (HIV-VL) values at entry to the early HIV infection cohort (EIC). *P*, for differences in HIV-VL vs. IHG-I is shown. **j** HIV-VL by entry IHG in the primary HIV infection cohort (PIC). **k** HIV-VL at entry and subsequent 5 years of therapy-naïve follow-up in EIC participants. Differences in HIV-VL are between participants with IHG-I vs. rest (i.e., IHG-II, IHG-III, or IHG-IV) at the indicated timepoints. *$P < 0.05$; **$P < 0.01$; ***$P < 0.001$; ns nonsignificant. For box plots: center line, median; box, interquartile range (IQR); whiskers, rest of the data distribution and outliers greater than ±1.5 × IQR are represented as points. Two-sided tests were used. Statistics are outlined in Supplementary Information Section 11.3.4., *P* values are in Supplementary Data 14, and Source data are provided as a Source Data file.

COVID-19 could have been IHG-I or IHG-II. Group C was envisaged based on having IHG-IV at presentation and before COVID-19. While age was associated with a stepwise increase in the likelihood of hospitalization and death (Supplementary Fig. 7), presentation with IHG-II or IHG-IV (represented in groups B or C) vs. IHG-I (group A) was associated with a significantly higher odds ratio of hospitalization (Fig. 4g, middle) and hazard ratio of all-cause, 30-day mortality (Fig. 4g, bottom), after controlling for age. CMV serostatus was not associated with hospitalization or death (Fig. 4g, middle and bottom). These findings suggest that (i) the capacity to preserve IHG-I both before and during early SARS-CoV-2 infection was associated with less-severe COVID-19 (nonhospitalization, survival), and (ii) while CMV serostatus may influence the nature of the IHG that emerges during COVID-19 (Fig. 3b–d), serostatus may not directly influence COVID-19 outcomes.

### Study phase 2: IR erosion-resistant phenotype and resistance to cancer in HIV− RTRs

RTRs are at a heightened (up to 100-fold) risk of developing recurrent cutaneous squamous cell carcinoma (CSCC)[15]. We examined the risk of a second episode of CSCC according to the IHG at the time of initial diagnosis of CSCC (baseline). In a prospective RTR cohort (Supplementary Data 5)[15], the hazard of a second episode of CSCC was lowest, intermediate, and highest in individuals who, at the time of the first episode of CSCC, had IHG-I, IHG-II, and IHG-III or IHG-IV, respectively (Fig. 4h). In persons with recurrent CSCC, duration of immunosuppression or age did not differ substantially by baseline IHG (Supplementary Data 5). CD57$^+$CD8$^+$ T-cells are a marker of incomplete differentiation of CD8$^+$ T-cells with functional properties of both early effector memory cells and terminally differentiated effector cells[33]. CD57$^+$CD8$^+$ T-cells are an independent determinant of recurrent CSCC[15]; proportions of these cells were highest in individuals with IHG-III or IHG-IV at baseline (Supplementary Fig. 8). All RTRs with IHG-III or IHG-IV were CMV+ (Supplementary Data 5). Thus, 22.5% ($n = 9$; Fig. 4h) of 40 RTRs had preserved IHG-I at the time of the first CSCC; this elite group appeared to have resistance against progression to a second episode of CSCC.

### Study phase 2: IR erosion-resistant phenotype and resistance to AIDS

In participants of the early HIV infection cohort, the rates of progression to AIDS were slowest, intermediate, and fastest in patients who at presentation had IHG-I, IHG-II or IHG-III, and IHG-IV, respectively (Fig. 4i). HIV-VL in participants from the early (Fig. 4i) and primary (Fig. 4j) HIV infection cohorts showed a gradient (highest to lowest) by baseline IHG: IHG-IV > IHG-III - IHG-II > IHG-I. Individuals in the elite group of therapy-naïve HIV+ persons shown in Fig. 3j who presented with and preserved IHG-I in each year of therapy-naïve disease exhibited lower HIV-VL vs. those who developed IHG-II, IHG-III, or IHG-IV (Fig. 4k). Thus, the elite capacity to preserve IHG-I during HIV infection was associated with greater immunocompetence as proxied by lower AIDS risk and restriction of HIV viral replication.

### Study phase 2: IR erosion-resistant phenotype and resistance to HIV acquisition

In FSWs, higher baseline BAS and total STI scores were associated with two outcomes: higher rates (Fig. 3k, bottom; Fig. 5a; Supplementary Fig. 3b) and odds (Fig. 5b) of (i) having IHG-III or IHG-IV, and (ii) HIV seroconversion. However, baseline IHG-III or IHG-IV vs. IHG-I was also associated with an increased likelihood of HIV seroconversion (Fig. 5b, rightmost). In multivariate analysis (Supplementary Data 8a), IHG-IV independently associated with a nearly 3-fold increased risk of HIV seroconversion (adjusted OR, 2.97; 95% CI, 1.05–8.38), after controlling for age, as well as BAS and total STI scores.

These findings suggest that risk factor-associated antigenic stimulation increases the risk of developing IHG-III or IHG-IV, and IHG-III

and especially IHG-IV prognosticate HIV seroconversion risk after controlling for BAS, a proxy for the level of HIV exposure. This inference was supported by our literature survey (Fig. 5c; Supplementary Table 2), as we found that %IHG-III or %IHG-IV was higher in (i) geographic areas with increased microbial exposures, including helminthic infections associated with HIV infection[34,35] (akin to Kenyan children with schistosomiasis shown in Fig. 3a), and (ii) cohorts with an increased risk of HIV acquisition [drug users and men who have sex with men (akin to FSWs shown in Figs. 3k, 5a)]. %IHG-III or %IHG-IV was nearly twice as high in men who have sex with men with higher- vs. lower-risk behavior [-22% vs. 10%; $P < 0.001$; Fig. 5c (akin to FSWs shown in Fig. 5a)].

### Study phase 2: IR erosion-resistant phenotype in other immunity-dependent conditions

Our literature survey showed that IHG-III or IHG-IV is also associated with increased mortality, a trend for reduced cognitive function, cancers in HIV+ persons, rapid progression of leukemia in HIV− persons, and a trend for lower influenza vaccine responsiveness, including in younger adults (Fig. 5c; Supplementary Table 2). This survey also affirmed that (i) prevalence of IHG-III or IHG-IV increases with age and is higher in males and (ii) CMV seropositivity rates in HIV− persons increase with age and IHG-III or IHG-IV associated with CMV seropositivity.

### Study phase 2: Convergence of IR status with disease continuity spectrum

Consistent with our model (Fig. 2e), the sum of the findings in study phase 2 were convergent for an association between preservation of IHG-I (IR erosion-resistant phenotype) and superior immunity-dependent health outcomes. Furthermore, these findings suggest that IR status indexed by the IHGs may shape the continuity spectrum from disease susceptibility to outcomes in the context of HIV-AIDS (Fig. 5d), COVID-19 (Fig. 5e), CSCC in RTRs (Fig. 5f), and possibly other conditions (Fig. 5c). The singular feature of progression along these continuity spectrums is that having a non-IHG-I grade (eroded IR) before and/or during disease may be associated with detrimental health outcomes.

### Study phase 2: Core set of detrimental non-IHG-I grades across varied conditions

Toward defining the precise level of IR eroded that prognosticates inferior immunity-dependent health outcomes, we characterized the full repertoire of IHGs that emerge in settings of antigenic stimulation. For this characterization, we derived subgrades a, b, and c of IHG-II and IHG-IV indexed to CD4$^+$ count thresholds of 200 and 500 cells/mm$^3$ (Fig. 6a). These two CD4$^+$ cutoffs are clinically relevant immunosuppression thresholds: ≤200 CD4$^+$ cells/mm$^3$ indicates AIDS[36] and 500 cells/mm$^3$ is the median CD4$^+$ count during primary/early HIV infection[7,8]. Thus, subgrades a, b, and c signify progressively lower CD4$^+$ counts, tracking incrementally greater immunosuppression but in the context of either CD8-CD4 equilibrium (IHG-II a, b, or c) or disequilibrium (IHG-IV a, b, or c) (Fig. 6a).

IHG-I and IHG-IIa were the first and second-most prevalent grades during aging (SardiNIA; Fig. 6b), convalescent COVID-19 (Fig. 6c, rightmost), and HIV− FSWs with lower BAS (Fig. 6d, rightmost). The b and c subgrades of IHG-II and the a and b subgrades of IHG-IV were more prevalent in settings of increased antigenic experience that included older SardiNIA participants, patients with acute COVID-19, patients with SLE, RTRs, therapy-naïve HIV+ persons, and FSWs with a higher BAS (Fig. 6b–d). In comparison with age-matched controls in the SardiNIA cohort (Fig. 6b), IHG distributions were restored to age-appropriate levels in persons recovering from COVID-19, but not in HIV+ individuals receiving ART during primary/early HIV infection (Fig. 6c–d).

Thus, the IHG repertoires provide a unifying framework of IR: a shared subset of detrimental non-IHG-I grades associated with worse

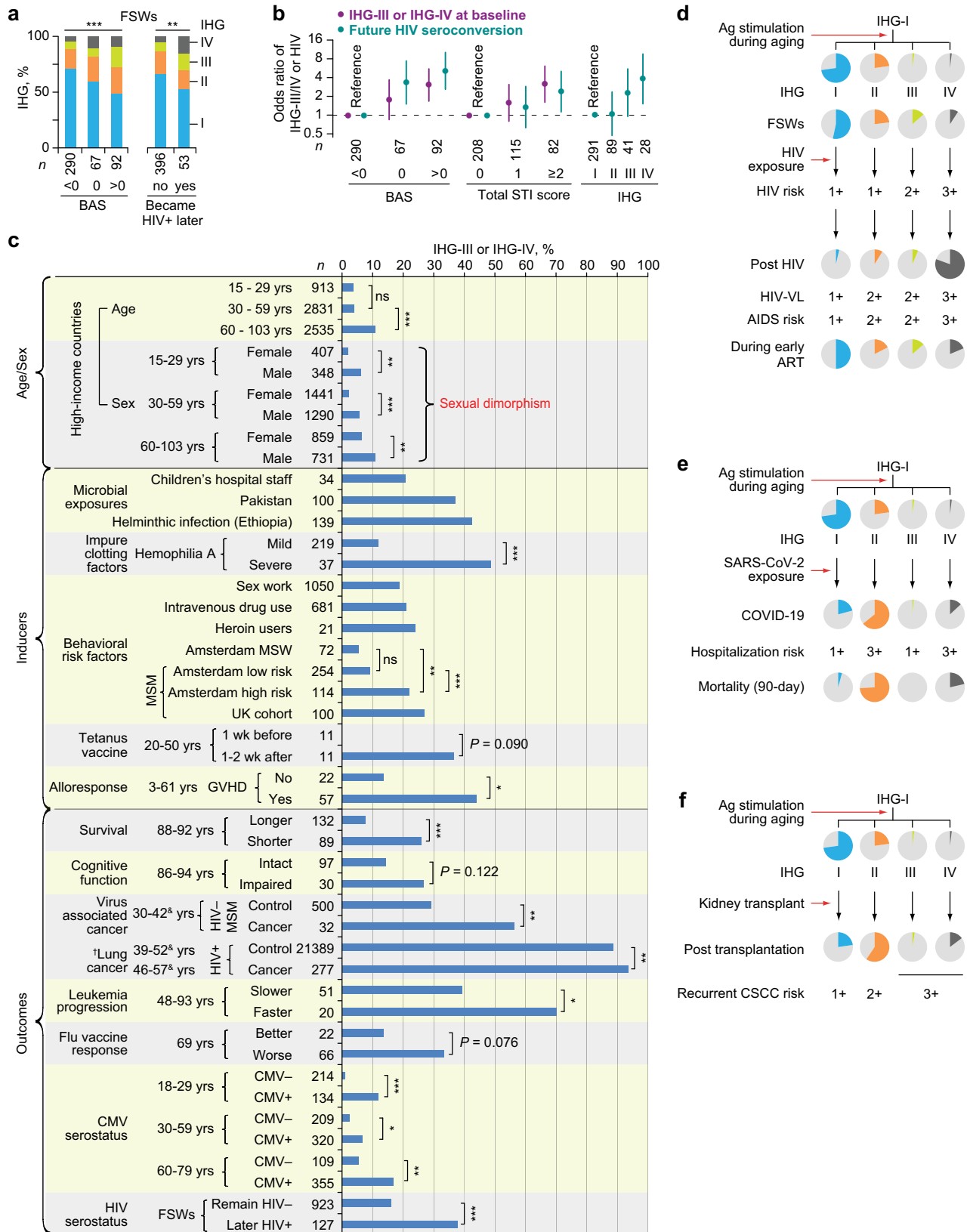

health outcomes emerges in settings of lower (e.g., aging), moderate (SARS-CoV-2, HIV risk factors), and higher (HIV) antigenic stimulation (Fig. 6b–e). While the antigenic stimulation noted with aging, SARS-CoV-2, and HIV infection was associated with CD4+ T-cell lymphopenia, the lymphopenia associated with aging (Supplementary Fig. 9; Supplementary Note 6) and COVID-19 occurs mainly within the context of

subgrades tracking CD8-CD4 equilibrium (e.g., IHG-IIb and IHG-IIc; Fig. 6b–c, e). In contrast, the CD4+ T-cell lymphopenia seen with HIV occurs within the context of subgrades tracking disequilibrium (IHG-IVa, IHG-IVb) (Fig. 6d–e). The subgrades may provide more precise risk prognostication attributable to where a person may reside along an IR continuum: (i) we previously found that presentation with subgrades b

**Fig. 5 | Inferior immunity-dependent health outcomes associated with immune health grades (IHGs) that correspond to the immunologic resilience (IR) erosion-susceptible phenotype. a** Female sex workers (FSWs) stratified first by baseline behavioral activity score (BAS) and then by subsequent HIV seroconversion status. **b** Odds ratio (OR) with 95% confidence interval (CI) of having IHG-III or IHG-IV at baseline (purple) or future HIV seroconversion (blue) in FSWs according to baseline BAS and total sexually transmitted infection (STI) score. Far right, OR for HIV seroconversion by baseline IHG. **c** Associations of CD8-CD4 disequilibrium grades IHG-III and IHG-IV with age and sex; inducers of these grades; and outcomes. Findings are from the literature survey (also see Supplementary Table 2 for details

and references) and our primary datasets. Flu, influenza; CMV, cytomegalovirus; MSW, men who have sex with men; &, interquartile range for age. †The original data from[106] stratified the CD4:CD8 ratio as ≤1.0 and >1.0. **d–f** Models depicting risk of indicated outcomes is lower in persons with the IR erosion-resistant phenotype (IHG-I). **d** HIV-AIDS, (**e**) COVID-19, and (**f**) recurrent cutaneous squamous cell cancer (CSCC) in renal transplant recipients. Pie charts depict relative proportions of the IHGs in the study group. Risk scaled from 1 to 3. Ag, antigenic; VL, viral load. *$P < 0.05$; **$P < 0.01$; ***$P < 0.001$; ns nonsignificant. Two-sided tests were used. Statistics are outlined in Supplementary Information Section 11.3.5., $P$ values are in Supplementary Data 14, and Source data are provided as a Source Data file.

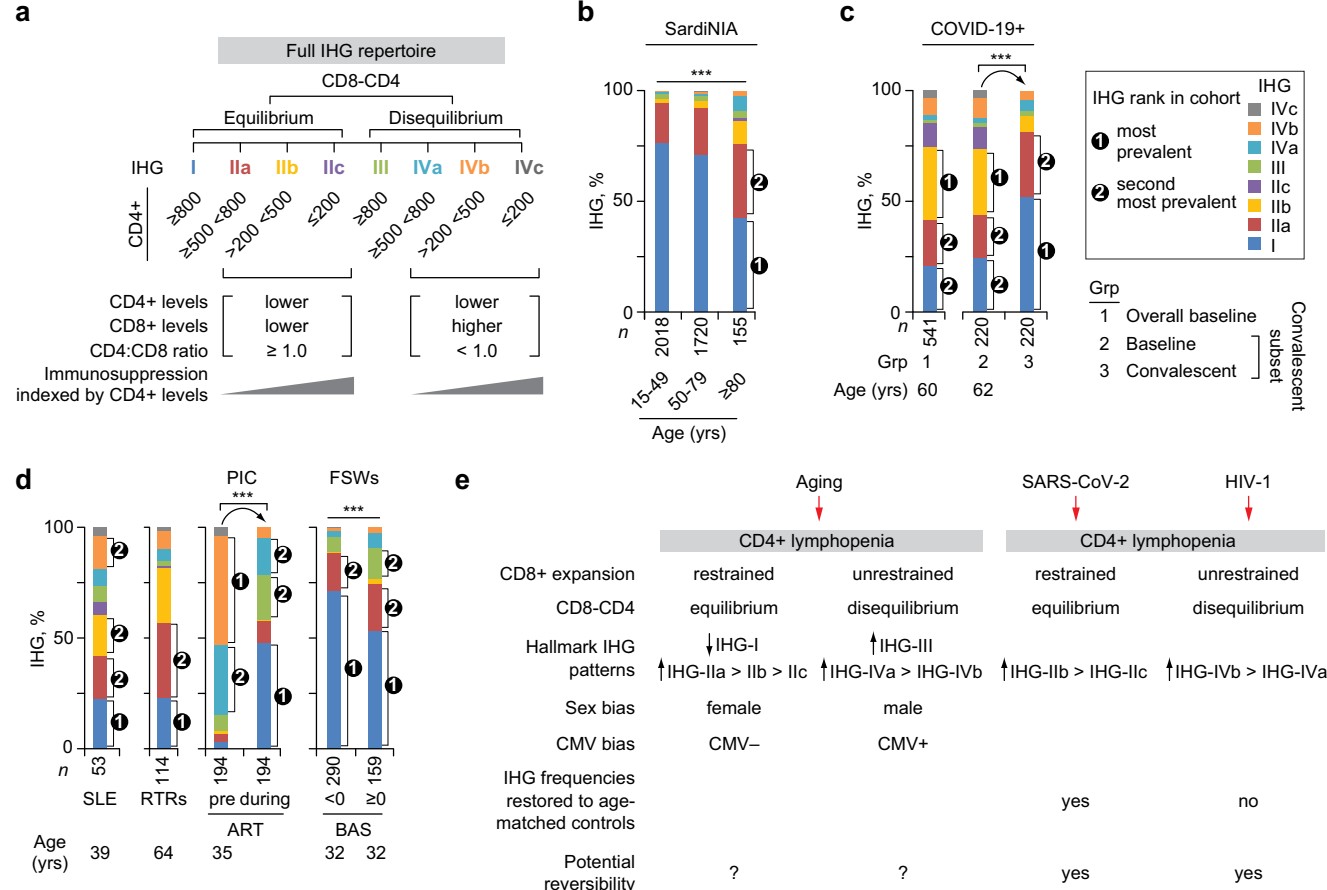

**Fig. 6 | Immune health grade (IHG) repertoire across HIV− and HIV+ cohorts with parallels between aging, COVID-19, and HIV disease. a** Schema for defining the full repertoire of IHGs. Subgrades of IHG-II and IHG-IV defined by the CD4⁺ T-cell count thresholds. **b** Distribution of IHGs with subgrades in the SardiNIA cohort by age strata. **c** IHGs with subgrades in the overall acute COVID-19 cohort at baseline ($n = 541$) and the subset of 220 individuals with available IHG data at baseline and convalescence. **d** IHGs with subgrades in persons with systemic lupus erythematosus (SLE), renal transplant recipients (RTRs), participants from the

primary HIV infection cohort (PIC) before initiation of antiretroviral therapy (ART) and following ART, and female sex workers (FSWs) by baseline behavioral activity score (BAS, <0 vs. ≥0). Age, median age at IHG assessment, baseline or pre-ART are shown. **e** Model depicting the enrichment of non-IHG-I grades during aging and at presentation with COVID-19 or HIV infection. ***$P < 0.001$. Two-sided tests were used. Statistics are outlined in Supplementary Information Section 11.3.6., $P$ values are in Supplementary Data 14, and Source data are provided as a Source Data file.

and c of IHG-II or IHG-IV predicted higher risk of COVID-19-associated mortality[6], after controlling for age; (ii) HIV acquisition occurred mainly in FSWs presenting with IHG-III and IHG-IVa (Supplementary Fig. 10; Supplementary Data 8b); and (iii) while IHG-IVc is an indicator of AIDS, this grade is also observed in patients with acute COVID-19, those with SLE, and RTRs.

**Convergence of study phases 1 and 2: IHG repertoire defines an IR continuum**
Our findings suggest that the IHG repertoire defines three tiers of IR (Fig. 7a). Tiering was based on (i) the similarity in the prevalence of the IHG repertoire in HIV− settings associated with lower antigenic

stimulation, i.e., aging cohorts (Figs. 2f, 6b), convalescent COVID-19 (Fig. 6c), and FSWs with lower BAS (Fig. 6d), and (ii) our finding that IHG-I was associated with varied superior immunity-dependent health outcomes, whereas a shared set of detrimental non-IHG-I grades (IHG-IIb, IHG-IIc, IHG-III, and subgrades of IHG-IV) was associated with inferior health outcomes in specific contexts. While IHG-III prevalence is low in HIV− persons with or without COVID-19, the prevalence of IHG-III is increased in specific contexts with relevance to HIV infection. IHG-III prevalence was elevated mainly in two distinct settings: HIV− FSWs with higher BAS and STI scores (Figs. 3k, 5a, 6d) and HIV+ patients with low HIV-VL or receiving ART (Figs. 3h, 3j, 6d). For these reasons, IHG-III was classified as a detrimental non-IHG-I grade in this

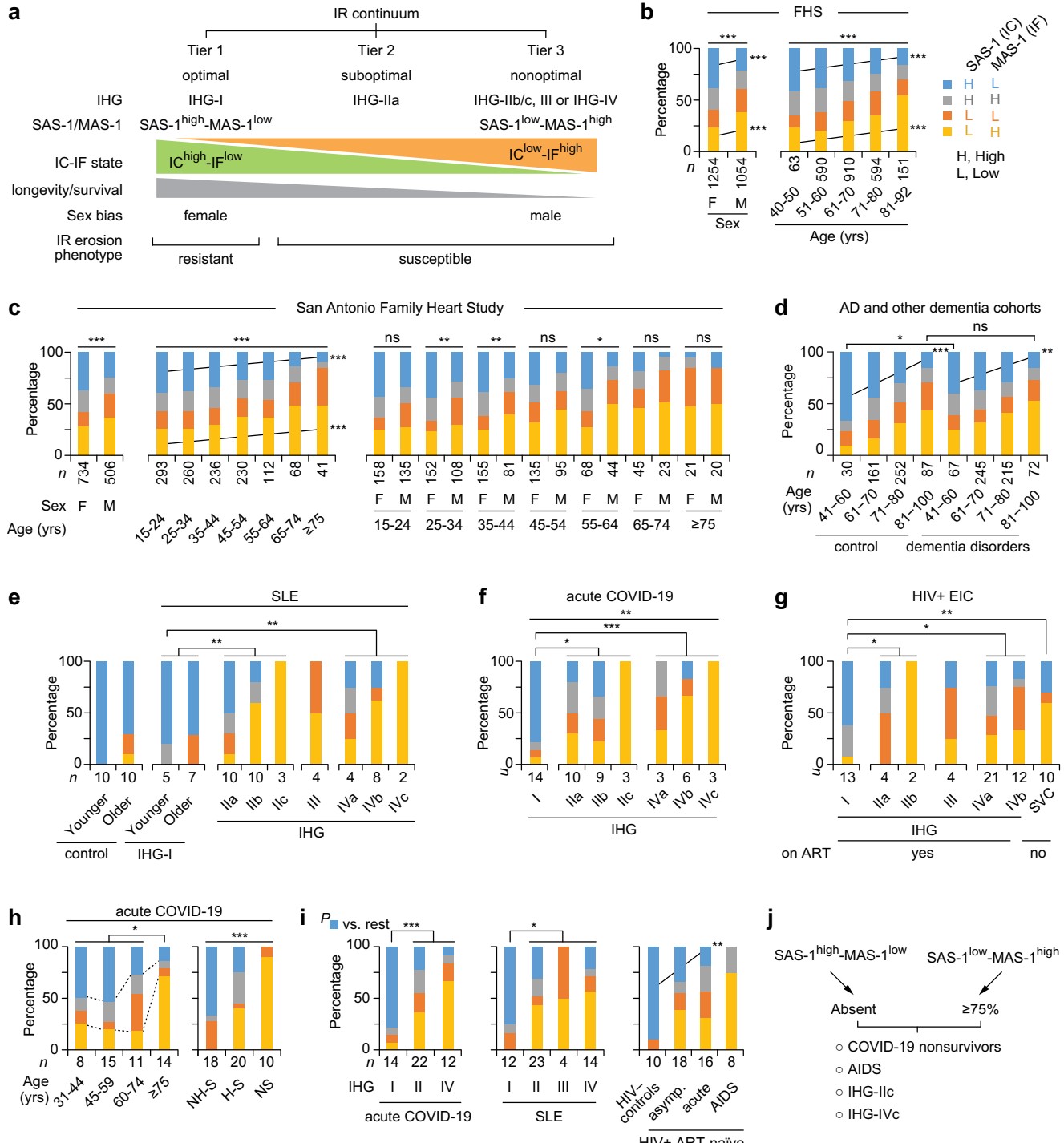

**Fig. 7 | Immunologic resilience (IR) continuum defined by transcriptomic metrics of IR associated with immune health grades (IHGs).** **a** Schema for IR continuum. IR tiers and erosion phenotypes defined by the IR metrics IHGs, survival-associated signature (SAS)-1, and mortality-associated signature (MAS)-1. Higher expression of SAS-1 and MAS-1 serve as transcriptomic proxies for immunocompetence (IC) and inflammation (IF), respectively. Groupings of SAS-1 and MAS-1 based on higher or lower levels of these signatures are depicted. **b–i** Distribution of the SAS-1/MAS-1 groupings/profiles in (**b**) the Framingham Heart Study (FHS) stratified by sex and age; (**c**) the San Antonio Family Heart Study categorized by sex, age, and both; (**d**) a meta-analysis of persons without (controls) vs. with Alzheimer disease (AD) and other dementia disorders; (**e**) persons without (control) vs. with systemic lupus erythematosus (SLE) stratified by IHGs with subgrades and age strata; (**f**) the acute COVID-19 cohort stratified by IHGs with

subgrades; (**g**) participants of the early HIV infection cohort (EIC) stratified by IHGs with subgrades reconstituted during virally suppressive antiretroviral therapy (ART) or in therapy-naïve spontaneous virologic controller (SVC); (**h**) the acute COVID-19 cohort sampled at baseline stratified by age, hospitalization, and survivor status; (**i**) acute COVID-19 cohort and patients with SLE stratified by IHGs, and healthy controls and therapy-naïve (without ART) HIV+ patients by disease stage. Asymp, asymptomatic. **j** Schema, proportions of SAS-1/MAS-1 groupings/profiles. In panels (**b–i**) the SAS-1/MAS-1 groupings are based on cohort-level higher or lower expression (above or below median, respectively) of SAS-1 and MAS-1. Cohort characteristics and sources of gene expression profile data are in Supplementary Data 13a. *$P < 0.05$; **$P < 0.01$; ***$P < 0.001$; ns nonsignificant. Two-sided tests were used. Statistics are outlined in Supplementary Information Section 11.3.6., $P$ values are in Supplementary Data 14, and Source data are provided as a Source Data file.

study. Thus, the IHGs define a continuum: IHG-I, the most prevalent grade, signified optimal IR (tier 1); IHG-IIa, the second-most prevalent grade, signified suboptimal IR (tier 2); and the detrimental and less-frequent non-IHG-I grades signified nonoptimal IR (tier 3) (Fig. 7a).

**Study phase 3: IHG repertoire links SAS-1/MAS-1 profiles to the IR continuum**

Our hypothesis (Figs. 1a, 2e) predicts that the IR continuum indexed to the IHG repertoire tracks a survival spectrum (Fig. 7a). To test this proposition, we examined whether the transcriptomic (gene expression) metrics of IR, namely, survival- vs. mortality-associated SAS-1/MAS-1 transcriptomic profiles (Fig. 2a, c–d), were associated with the IHG repertoire in a non-stochastic manner. We posited that: (i) IHG-I tracks optimal IR, as it is strongly linked with a transcriptomic proxy (SAS-1$^{high}$-MAS-1$^{low}$) for an IC$^{high}$-IF$^{low}$ state that is associated with a longevity/survival advantage (tier 1); (ii) the detrimental non-IHG-I grades track nonoptimal IR, as they are strongly linked with a transcriptomic proxy (SAS-1$^{low}$-MAS-1$^{high}$) for an IC$^{low}$-IF$^{high}$ state that is associated with a longevity/survival disadvantage (tier 3); and (iii) IHG-IIa tracks suboptimal IR, attributable to a weaker association with an IC$^{high}$-IF$^{low}$ state (tier 3; Fig. 7a). In this schema, the SAS-1/MAS-1 profiles track an IR continuum partly attributable to their association with the IHGs (Fig. 7a).

To corroborate that SAS-1$^{high}$ was a transcriptomic proxy for IC$^{high}$ and not IF$^{low}$, we focused on the findings of Alpert et al., who characterized the IMM-AGE gene expression signature[11]. Higher levels of IMM-AGE (based on gene expression) associated with lower levels of an immune-aging metric based on immune senescence-associated T-cell subset frequencies[11] as well as survival in the FHS cohort (Fig. 2d). We found that, akin to higher SAS-1 expression, higher expression of IMM-AGE was also associated with lower mortality hazards in the COVID-19 cohort (Fig. 2d). Congruently, expression of SAS-1 and IMM-AGE was positively correlated; conversely, SAS-1 and IMM-AGE expression was negatively correlated with MAS-1 expression (Supplementary Fig. 11).

Our finding that SAS-1, MAS-1, and IMM-AGE associated with survival/mortality after controlling for age and sex (Fig. 2d) was consistent with two observations in the aging cohorts. First, the correlation between expression of these gene signatures and age, while statistically significant, was low (Supplementary Fig. 11). Second, while expression levels of SAS-1 and IMM-AGE declined and those of MAS-1 increased with age (Supplementary Fig. 12), levels in older persons approximated those of younger individuals with conditions associated with lower immunocompetence and inflammation (e.g., tuberculosis, HIV) (Supplementary Fig. 12c). Thus, the age-associated changes in SAS-1 and MAS-1 levels appeared to be more closely related to accumulated antigenic experience than the direct effects of age per se.

Together, these findings and the gene composition of the signatures (Fig. 2d) suggest that SAS-1 and IMM-AGE appear to track similar longevity- and IC-associated immune mechanisms, whereas MAS-1 appears to track distinct mortality- and IF-associated mechanisms, after controlling for age. This distinction provided the rationale to derive combined SAS-1/MAS-1 profiles and determine their association with the IHGs. Based on higher and lower levels of SAS-1 and MAS-1, we derived four SAS-1/MAS-1 profiles representative of four IC-IF states (Fig. 7b, key code). SAS-1$^{high}$-MAS-1$^{low}$, SAS-1$^{high}$-MAS-1$^{high}$, SAS-1$^{low}$-MAS-1$^{low}$, and SAS-1$^{low}$-MAS-1$^{high}$ profiles are considered as representative of IC$^{high}$-IF$^{low}$, IC$^{high}$-IF$^{high}$, IC$^{low}$-IF$^{low}$, and IC$^{low}$-IF$^{high}$ states, respectively (Fig. 7b, key code).

**Study phase 3: SAS-1/MAS-1 profiles by age, sex, and IHGs**

Four findings made across multiple cohorts support the proposed association between SAS-1/MAS-1 profiles and the IHG repertoire that define the IR continuum (Fig. 7a). First, akin to the age-associated shift from IHG-I to non-IHG-I grades (Fig. 2f), in varied aging cohorts, the proportion of individuals with the SAS-1$^{high}$-MAS-1$^{low}$ profile declined progressively with age, whereas the proportion with SAS-1$^{low}$-MAS-1$^{high}$ increased (Fig. 7b–d; Supplementary Fig. 13a–b). Second, akin to the overrepresentation of IHG-I in females across age strata (Fig. 2f), SAS-1$^{high}$-MAS-1$^{low}$ vs. SAS-1$^{low}$-MAS-1$^{high}$ profiles were more prevalent in females than males (Fig. 7b–c; Supplementary Fig. 13a–b). These findings were consistent with our observation that, across all ages in the FHS, females compared with males preserved higher levels of SAS-1 and lower levels of MAS-1 (Supplementary Fig. 12a).

Third, even in cohorts with participants experiencing increased antigenic stimulation (SLE, acute COVID-19, and HIV+ on ART), IHG-I was nearly universally overrepresented with the SAS-1$^{high}$-MAS-1$^{low}$ profile, whereas SAS-1$^{low}$-MAS-1$^{high}$ was absent or underrepresented (Fig. 7e–g). Conversely, representation of SAS-1$^{low}$-MAS-1$^{high}$ was progressively greater with the a, b, and c subgrades of IHG-II and IHG-IV (Fig. 7e–g). IHG-III lacked representation of the SAS-1$^{high}$-MAS-1$^{low}$ profile. Thus, IHG-I was hallmarked by nearly complete representation of the SAS-1$^{high}$-MAS-1$^{low}$ profile and underrepresentation of the SAS-1$^{low}$-MAS-1$^{high}$ profile. In contrast, IHG-IIc and IHG-IVc were hallmarked by complete representation of the SAS-1$^{low}$-MAS-1$^{high}$ profile and absence of the SAS-1$^{high}$-MAS-1$^{low}$ profile. IHG-IIa had some representation of the SAS-1$^{high}$-MAS-1$^{low}$ profile. Congruent with these findings, expression of SAS-1 was higher, whereas expression of MAS-1 was lower in IHG-I vs. the other grades in three distinct cohorts (Supplementary Fig. 13c).

These findings support our proposition that the representation of SAS-1/MAS-1 profiles in IHGs that define the IR continuum is not stochastic (Fig. 7a). Congruently, we found that the baseline IHG status and SAS-1/MAS-1 profiles showed similar prognostication during acute COVID-19. In the COVID-19 cohort, there was a stepwise decrease in IHG-I with age (Fig. 3e) and presentation with IHG-I was associated with nonhospitalization and survival (Fig. 4g). Paralleling these findings with IHG-I, the representation of SAS-1$^{high}$-MAS-1$^{low}$ (i) decreased with age (Fig. 7h, left); (ii) was higher in nonhospitalized vs. hospitalized survivors and absent in nonsurvivors (Fig. 7h, right); and (iii) was enriched (~75%) in patients with IHG-I (Fig. 7i, leftmost). Conversely, representation of the SAS-1$^{low}$-MAS-1$^{high}$ profile was higher in older persons, nonsurvivors, and individuals with IHG-IV; intermediate in hospitalized survivors and those with IHG-II; and lower or absent in nonhospitalized survivors or those with IHG-I (Fig. 7h–i).

Fourth, consistent with our finding that some younger persons develop non-IHG-I grades that are more common in older persons (Figs. 2f, 6b), we found that some otherwise younger healthy persons manifest the SAS-1$^{low}$-MAS-1$^{high}$ profile that is more common in older persons (Fig. 7b–d) and individuals with advanced immunosuppression with HIV infection (Fig. 7i). Furthermore, the relative representation of SAS-1$^{high}$-MAS-1$^{low}$ vs. SAS-1$^{low}$-MAS-1$^{high}$ observed in therapy-naïve HIV+ patients was similar to that observed in individuals with non-IHG-I grades in the acute COVID-19 and SLE cohorts (Fig. 7e, f, i), as well as in older persons (Fig. 7b–d). Hence, individuals with a survival disadvantage (COVID-19 nonsurvivors, patients with AIDS) share the hallmark features found in IHG-IIc and IHG-IVc, namely, absence of SAS-1$^{high}$-MAS-1$^{low}$ and enrichment of SAS-1$^{low}$-MAS-1$^{high}$ (Fig. 7j).

Taken together, these findings suggest that the non-stochastic representation of the SAS-1/MAS-1 profiles in the IHGs define an IR continuum (Fig. 7a). We suggest that (i) the SAS-1$^{high}$-MAS-1$^{low}$ profile is a transcriptomic proxy for IHG-I and that both of these metrics of optimal IR are overrepresented in females (Fig. 2f–g; Fig. 7b–c; Supplementary Fig. 13a–b); (ii) SAS-1$^{low}$-MAS-1$^{high}$ is a transcriptomic proxy for detrimental non-IHG-I grades (nonoptimal IR); (iii) indicative of the IR erosion-susceptible phenotype, some younger persons, especially males, have a predilection to manifest SAS-1$^{low}$-MAS-1$^{high}$ (Fig. 7b–d) and non-IHG-I grades (Fig. 2f); and (iv) in each age stratum, indicative of the IR erosion-resistant phenotype, some persons, including older individuals, preserve SAS-1$^{high}$-MAS-1$^{low}$ (Fig. 7b–d) and IHG-I (Fig. 2f).

## Study phase 3: SAS-1/MAS-1 profiles predict survival during aging

Compared with the SAS-1^high-MAS-1^low profile, the hazard of dying, after controlling for age and sex, was higher and similar in persons with the SAS-1^high-MAS-1^high and SAS-1^low-MAS-1^low profiles and highest in persons with the SAS-1^low-MAS-1^high profile (Fig. 8a; Supplementary Data 9e). Correspondingly, SAS-1^low-MAS-1^high was overrepresented and SAS-1^high-MAS-1^low was underrepresented at baseline in nonsurvivors (Fig. 8b). Two additional findings supported the associations of the SAS-1/MAS-1 profiles with survival rates, after controlling for age. First, among older FHS participants, females lived longer than males, and levels of SAS-1 and MAS-1 further stratified survival rates (Supplementary Fig. 13d). The survival rates in older (66–92 years) persons were highest in females with SAS-1^high or MAS-1^low, intermediate in females with SAS-1^low or MAS-1^high and males with SAS-1^high or MAS-1^low, and lowest in males with SAS-1^low or MAS-1^high (Supplementary Fig. 13d). This survival hierarchy and our findings in Fig. 8a suggest that age, sex, and IR status indexed to SAS-1/MAS-1 profiles were independent determinants of lifespan (Fig. 8c; Supplementary Note 7). Second, in the Vitality 90+ study, MAS-1^low in the context of SAS-1^high-MAS-1^low or SAS-1^low-MAS-1^low provided a further survival advantage to nonagenarians (Supplementary Fig. 13e; Supplementary Note 7).

## Study phase 3: SAS-1/MAS-1 profiles predict survival during sepsis

Based on gene expression profiles obtained at baseline (admission), Knight and colleagues categorized four cohorts of individuals into sepsis risk groups that predicted mortality vs. survival in individuals admitted to intensive care units with severe sepsis due to community-acquired pneumonia or fecal peritonitis[37,38]. Our evaluations revealed that, irrespective of age, the survival-associated SAS-1^high-MAS-1^low profile was highly underrepresented, whereas SAS-1^low-MAS-1^high and SAS-1^low-MAS-1^low profiles were disproportionately overrepresented in the sepsis risk group associated with mortality (G1 group) vs. survival (G2 group) (Fig. 8d, left – sepsis #1; Supplementary Fig. 13f). Corroborating these findings, we found that the sepsis cascade (systemic inflammatory response syndrome→sepsis→septic shock with survival→septic shock with death), as well as worsening Sequential Organ Failure Assessment score during sepsis and higher total burn surface area (>20% vs ≤20%) was associated with a progressive under-representation of SAS-1^high-MAS-1^low with a reciprocal increase in the representation of the SAS-1^low-MAS-1^high profile (Fig. 8d, right – sepsis #2; Supplementary Fig. 13g–h)[39–42]. Thus, consistent with our model (Fig. 7a), we found that among persons without (Fig. 8a–b) and with sepsis (Fig. 8d), preservation of SAS-1^high-MAS-1^low profile was indicator of the IR erosion-resistant phenotype and was associated with a longevity/survival advantage.

## Study phase 3: SAS-1/MAS-1 profiles predict respiratory viral infection severity

In 18- to 49-year-old adults, acute respiratory infection (ARI) with common seasonal viruses (influenza A and B, rhinovirus, and others) associated with the mortality-associated SAS-1^low-MAS-1^high profile within 48 h of symptom onset (day 0; Fig. 8e, left). SAS-1^low-MAS-1^high was represented in 24% of individuals pre-ARI vs. in nearly 84% on day 0 ($P < 0.001$). During recovery (convalescence), representation of SAS-1^low-MAS-1^high progressively declined with a reciprocal increase in the representation of SAS-1^high-MAS-1^low; by day 21, representation of the SAS-1/MAS-1 profiles resembled pre-ARI levels (Fig. 8e, left). However, the SAS-1/MAS-1 profiles reconstituted during convalescence was dependent on the pre-ARI profile. Most of the individuals who had SAS-1^high-MAS-1^low pre-ARI manifested SAS-1^low-MAS-1^high at day 0 and nearly 67% reconstituted SAS-1^high-MAS-1^low by day 21 and beyond (Fig. 8e, middle). Thus, despite having the SAS-1^high-MAS-1^low profile pre-ARI, nearly 30% of these individuals did not reconstitute this survival-

associated profile post-ARI. Recovery of SAS-1/MAS-1 profiles in individuals with SAS-1^low-MAS-1^high pre-ARI was variegated, and only 29% reconstituted a SAS-1^high-MAS-1^low profile post-ARI (Fig. 8e, right).

We next examined whether asymptomatic ARI was associated with the IR erosion-resistant phenotype, i.e., asymptomatic status related to the capacity to resist induction of the mortality-associated SAS-1^low-MAS-1^high profile after intranasal challenges with common respiratory viruses. Resistance was determined by comparing the representation of the SAS-1/MAS-1 profiles in persons with asymptomatic vs. symptomatic infection after viral challenge at two timepoints: baseline (T1) vs. when symptomatic patients had peak symptoms (T2) (Fig. 8f). Figure 8g shows the combined results of three different viral challenges (influenza virus, respiratory syncytial virus, rhinovirus). After intranasal inoculations of virus, nearly 50% of the participants had an asymptomatic infection (Fig. 8g). Among symptomatic participants, SAS-1^low-MAS-1^high was enriched at T2 vs. T1 (Fig. 8g). In contrast, among persons who remained asymptomatic, proportions of the SAS-1^low-MAS-1^high profile did not change substantially between T1 and T2; instead at T2, there was a significant enrichment of SAS-1^high-MAS-1^low compared to symptomatic participants (Fig. 8g). Similar results were observed in another study in which participants were challenged with influenza virus (Fig. 8h). Thus, the capacity to preserve and/or induce expression of the survival-associated SAS-1^high-MAS-1^low profile following experimental viral inoculations was a marker of asymptomatic infection. Supporting these findings in humans, among pre-Collaborative Cross-RIX mice strains infected with influenza, SAS-1^high-MAS-1^low was overrepresented, whereas SAS-1^low-MAS-1^high was underrepresented in strains that manifested histopathologic features of mild (low response) vs. severe (high response) infection (Supplementary Fig. 13i).

Paralleling the time series shown in Fig. 8e, the mortality-associated SAS-1^low-MAS-1^high profile was overrepresented at baseline in hospitalized patients with influenza followed by reconstitution of SAS-1^high-MAS-1^low at least 4 weeks after hospitalization (Fig. 8i, left-most). However, regardless of age, the hallmark of less-severe vs. most-severe influenza infection was the capacity to reconstitute a survival-associated SAS-1^high-MAS-1^low profile more quickly (Fig. 8i, right).

## Convergent findings in study phases 1–3 define IR phenotypes

Paralleling our findings with IHG-I status in study phase 2, our findings in study phase 3 suggest that preservation and/or rapid reconstitution of SAS-1^high-MAS-1^low, the transcriptomic proxy for IHG-I, is an indicator of the IR erosion-resistant phenotype, as it was associated with three superior outcomes: (i) a longevity advantage during aging (Fig. 8a), (ii) a survival advantage during sepsis (Fig. 8d; Supplementary Fig. 13f, h), and (iii) less-severe or asymptomatic infection during natural or experimental ARI with common seasonal respiratory viruses (e.g., influenza) (Fig. 8e–i). Figure 9a synthesizes the key findings from study phases 1, 2, and 3. Viral challenge studies in humans (Fig. 8g–h) suggest that the longevity/survival advantage linked to the IR erosion-resistant phenotype may reflect two immune allostatic responses or their combination during life: minimal or no susceptibility to develop the mortality-associated IC^low-IF^high state (SAS-1^low-MAS-1^high profile) during acute antigenic stimulation (Fig. 9b, left) vs. rapid restoration of the survival-associated IC^high-IF^low state (SAS-1^high-MAS-1^low profile) during the convalescence phase (Fig. 9b, right).

## Study phase 4: Immune correlates of IR – an immunologic trifecta

To further support the idea that the SAS-1^high-MAS-1^low profile tracks an IC^high-IF^low state, we determined the correlation between expression levels of genes comprising SAS-1 and MAS-1 with indicators of T-cell responsiveness and dysfunction in peripheral blood[8,43], as well as systemic inflammation (plasma IL-6, a biomarker of age-associated diseases and mortality[44–46]) (Fig. 9c; right). Genes correlating positively

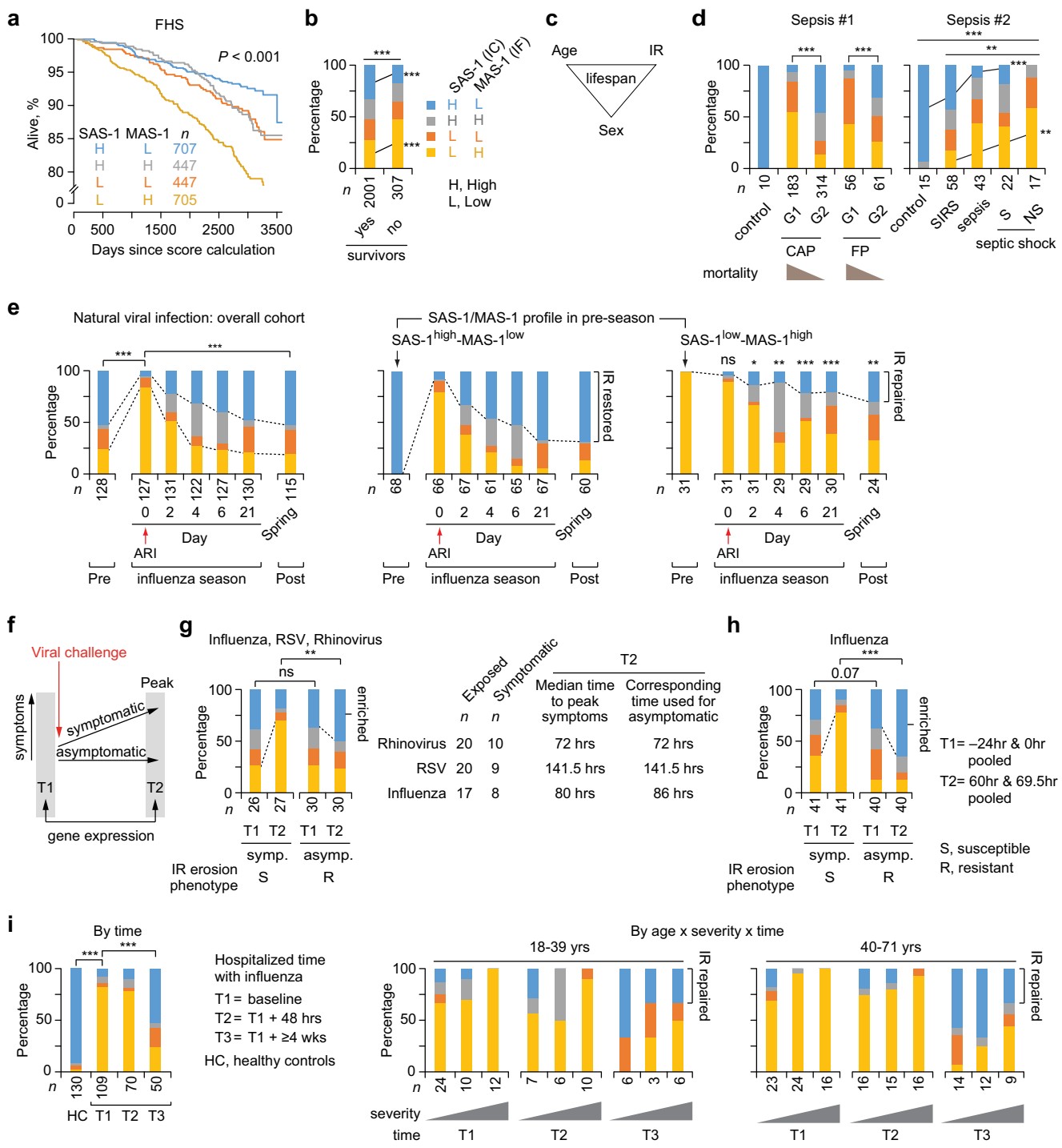

with T-cell responsiveness and negatively with T-cell dysfunction or plasma IL-6 levels were considered to have pro-IR functions; genes with the opposite attributes were considered to have IR-compromising functions (Fig. 9c, bottom right). We found that SAS-1 was enriched for genes whose expression levels correlated positively with pro-IR functions; several of these genes have essential roles in T-cell homeostasis (e.g., *TCF7*[47,48], *IL24*[49], *IL17R*[50], *CD27*[51], *ICOS*[52]) (Fig. 9c, left; Supplementary Data 10). Compared with SAS-1, MAS-1 was enriched for genes whose expression levels correlated with IR-compromising functions (e.g., *TLR2*[53], *MYD88*[54], *ADAM17*[55], *TBK1*[56]) (Fig. 9c, middle; Supplementary Data 10). These correlation patterns support the idea that SAS-1^high is a transcriptomic proxy for IC^high (pro-IR functions) and MAS-1^high is a transcriptomic proxy for IF^high (IR compromising functions)

and provide a possible mechanistic basis by which the SAS-1^high-MAS-1^low profile was associated with a longevity/survival advantage.

## Study phase 4: Immune traits associated with IHG status vs. age vs. both

In study phases 2 and 3, we found that the IR metrics [IHGs and their transcriptomic proxies (SAS-1/MAS-1 profiles)] associated with immunity-dependent outcomes, including longevity/survival, after controlling for age. These associations, coupled with the distribution patterns of the IR metrics across age, raised the possibility that levels of immune traits differed by (i) IR (IHG) status, after controlling for age (age-independent) vs. (ii) age, regardless of IR (IHG) status (age-dependent), vs. (iii) both. Additionally, because we observed

**Fig. 8 | Survival-associated signature (SAS)-1 and mortality-associated signature (MAS)-1 associate with mortality and acute respiratory viral infection outcomes. a** Proportion survived in the Framingham Heart Study (FHS) by SAS-1/MAS-1 groupings/profiles calculated at time 0. **b** Distribution of SAS-1/MAS-1 profiles in the FHS by survival status. **c** Model: age, sex, and immunologic resilience (IR) levels influence lifespan. **d**–**g** Representation of SAS-1/MAS-1 profiles. **d** Sepsis #1 comprises healthy controls and meta-analysis of patients with community-acquired pneumonia (CAP) and fecal peritonitis (FP) stratified by sepsis response signature groups (G1 and G2 associated with higher and lower mortality, respectively). Sepsis #2 comprises healthy controls and patients with systemic inflammatory response syndrome (SIRS), sepsis, and septic shock survivors (S) and nonsurvivors (NS). **e** Participants in a natural influenza season cohort (age: 18–49 years) sampled at pre and during acute respiratory infection (ARI) and at spring follow-up: overall (left) or according to the indicated SAS-1/MAS-1 profile during the pre-ARI season (right). $P$ values (asterisks, ns) for participants with SAS-1$^{low}$-MAS-1$^{high}$ at pre-ARI (right) are for their cross-sectional comparison to the profiles at the corresponding timepoints for participants with SAS-1$^{high}$-MAS-1$^{low}$ at pre-ARI (middle). **f** Schema of the timing of gene expression profiling in experimental intranasal challenges with respiratory viral infection in otherwise healthy young adults with data presented in panels g and h. T, time. **g** Participants inoculated intra-nasally with respiratory syncytial virus (RSV), rhinovirus, or influenza virus stratified by symptom status and sampling timepoint. Symp. symptomatic, Asymp. asymptomatic. **h** Participants inoculated intra-nasally with influenza virus stratified by symptom status and sampling timepoint. **i** Individuals with severe influenza infection requiring hospitalization collected at three timepoints, overall, and by age strata and severity. Patients were grouped by increasing severity levels: no supplemental oxygen required, oxygen by mask, and mechanical ventilation. Cohort characteristics and sources of biological samples and gene expression profile data are in Supplementary Data 13a. *$P < 0.05$; **$P < 0.01$; ***$P < 0.001$; ns nonsignificant. Two-sided tests were used. Statistics are outlined in Supplementary Information Section 11.3.8., $P$ values are in Supplementary Data 14, and Source data are provided as a Source Data file.

---

evolutionary parallels between humans and nonhuman primates (Figs. 2f, 4c–e), it was conceivable that similar trait patterns are observed in both species.

Compared with humans, a hallmark feature of nonhuman primates was the overrepresentation of IHGs tracking CD8-CD4 disequilibrium grades signifying unrestrained CD8$^+$ expansion with higher (IHG-III) or lower (IHG-IV) CD4$^+$ counts (Fig. 4d–e). We therefore first examined whether CD8$^+$ T-cell subsets expressing genes represented in SAS-1 [*CCR7* and the receptor for IL-7 (*CD127*)], differed by IHG status vs. age. In SIV+ sooty mangabeys, compared with IHG-I or IHG-II, IHG-III and IHG-IV were associated with lower levels of CD127$^+$CD8$^+$ and higher levels of CD28$^-$CD8$^+$ senescent/terminally differentiated T-cells (Fig. 9d). IHG-III and IHG-IV was also associated with lower levels of other traits noted with superior immunologic health (CCR7$^+$ and CXCR4$^+$ bearing CD8$^+$ T-cells[57], and naïve CD8$^+$ T-cells) (Fig. 9d). In SIV− rhesus macaques, expression of PD-1 on CD4$^+$ and CD8$^+$ T-cells increased with age (Fig. 9e, left; Supplementary Fig. 14; Supplementary Data 11). However, overall and within each age stratum, PD-1-expressing CD4$^+$ and CD8$^+$ T-cells were higher in rhesus macaques with IHG-III or IHG-IV (Fig. 9e, right; Supplementary Fig. 14; Supplementary Data 11). Trait levels in both species differed to a greater extent by IHG status than age (Supplementary Data 11). Thus, CD8-CD4 disequilibrium grades IHG-III and IHG-IV were highly prevalent in nonhuman primates (Fig. 4d–e) and associated with immune features linked to immune dysfunction after controlling for age. In general, IHG-I appeared to be associated with a better immune trait profile (e.g., higher levels of CCR7$^-$ expressing CD8$^+$ T-cells and naïve CD8$^+$ T-cells in SIV+ sooty mangabeys and lower levels of PD-1-expressing CD8$^+$ and CD4$^+$ T-cells in rhesus macaques).

Contrary to nonhuman primates, CD8-CD4 equilibrium grades IHG-I and IHG-II vs. disequilibrium grades IHG-III or IHG-IV are much more prevalent across age in humans (Fig. 2f). However, emphasizing evolutionary parallels, we identified similar traits associated with IHG status after controlling for age in both humans and nonhuman primates. In an analysis of the SardiNIA cohort ($n = 3896$; cohort description[9] and see Supplement Information), 75 immune traits (Supplementary Fig. 15–18) classified into four groups (Fig. 10a). Group 1 comprised 13 immune traits whose levels differed between CD8-CD4 equilibrium vs. disequilibrium grades (IHG-I vs. IHG-III or IHG-II vs IHG-IV), after controlling for age and sex. Group 2 comprised 22 immune traits whose levels differed by age (<40 vs. ≥70 years) after controlling for sex among persons with IHG-I or IHG-II—the two most prevalent grades in humans. Group 3 comprised 10 immune traits that differed by attributes of both groups 1 and 2 after controlling for sex. Group 4 (neutral) comprised 30 immune traits that did not differ by group 1 or 2 attributes (Fig. 10a; Supplementary Note 8; Supplementary Data 12).

Group 1 immune traits included natural killer T-cells (NKT), CD8$^+$ NKT-like cells, CD127$^-$CD8$^{bright}$ T-cells (effector-memory),

CD25$^{++}$CD8$^{bright}$ (activated/proliferating) T-cells, and CD28$^-$CD8$^{dim}$ (senescent/terminally differentiated) T-cells (signatures 1–4; Fig. 10a; Supplementary Data 12). Thus, levels of a representative trait in signature 1 (CD127$^-$CD8$^{bright}$ T-cells) was similar in older and younger persons with IHG-I or IHG-II but was higher in persons with IHG-III or IHG-IV vs IHG-I or IHG-II, irrespective of age (Fig. 10b). Congruently, across age, levels of CD127$^-$CD8$^{bright}$ T-cells differed significantly by IHG status ($P < 0.001$) relative to their association ($P = 0.015$) with age (Fig. 10c).

Group 2 immune traits included increased proportions of memory subsets within the CD4$^+$ T-cell compartment (except for the central memory subset) and decreased proportions of naïve CD8$^+$ T-cells, B cells (CD19$^+$), and plasmacytoid dendritic cells (CD123$^+$ CD11c$^-$) (e.g., signatures 5–8 and lower levels of signature 10; Supplementary Data 12). Figure 10b shows that the levels of a representative trait in signature 6 (naïve CD8$^{bright}$) differed between older vs. younger persons with IHG-I or IHG-II but did not differ by IHG status. Thus, group 2 immune traits represent traits that are associated with aged CD8-CD4 equilibrium.

Group 3 immune traits include CD28$^-$CD8$^{bright}$ T-cells that are viewed by some as a hallmark of aging[33]. While levels of CD28$^-$CD8$^{bright}$ T-cells were higher with both age and IHG-III and IHG-IV, they were disproportionately higher in persons with IHG-III and IHG-IV (signature 9; Fig. 10b) compared to IHG-I and IHG-II, respectively. Congruently, levels of CD28$^-$CD8$^{bright}$ T-cells both significantly increased with age and differed by IHG status (Fig. 10c). Neutral immune traits (Group 4) included mature dendritic cells (CD86$^+$), monocytes, and some Treg subsets (Supplementary Note 8; Supplementary Data 12). Additional trait features of Groups 1–4 are discussed (Supplementary Note 8).

Thus, suggesting evolutionary parallels, we identified similar immunologic features (e.g., CD127$^-$ and CD28$^-$ CD8$^+$ T-cells) that are associated with IR (IHG) status in SIV+ sooty mangabeys and HIV− humans, after controlling for age. However, since the prevalence of IHG-III or IHG-IV increases with age (Fig. 2f), immune traits associated with Groups 1 and 3 may also become more prevalent with age and be potentially misattributed to the effects of age alone.

## Discussion
Our study addresses a fundamental conundrum. Why do some younger individuals manifest attributes consistent with an immunosuppressive-proinflammatory state predisposing to increased disease risks/severity and premature mortality? Conversely, why do some older persons resist manifesting these attributes? This conundrum pointed to an immunosuppressive-proinflammatory process that is not directly attributable to age; we posited this process relates to the failure, at any age, to preserve and/or restore optimal IR when experiencing inflammatory (antigenic) stressors. This failure indicates

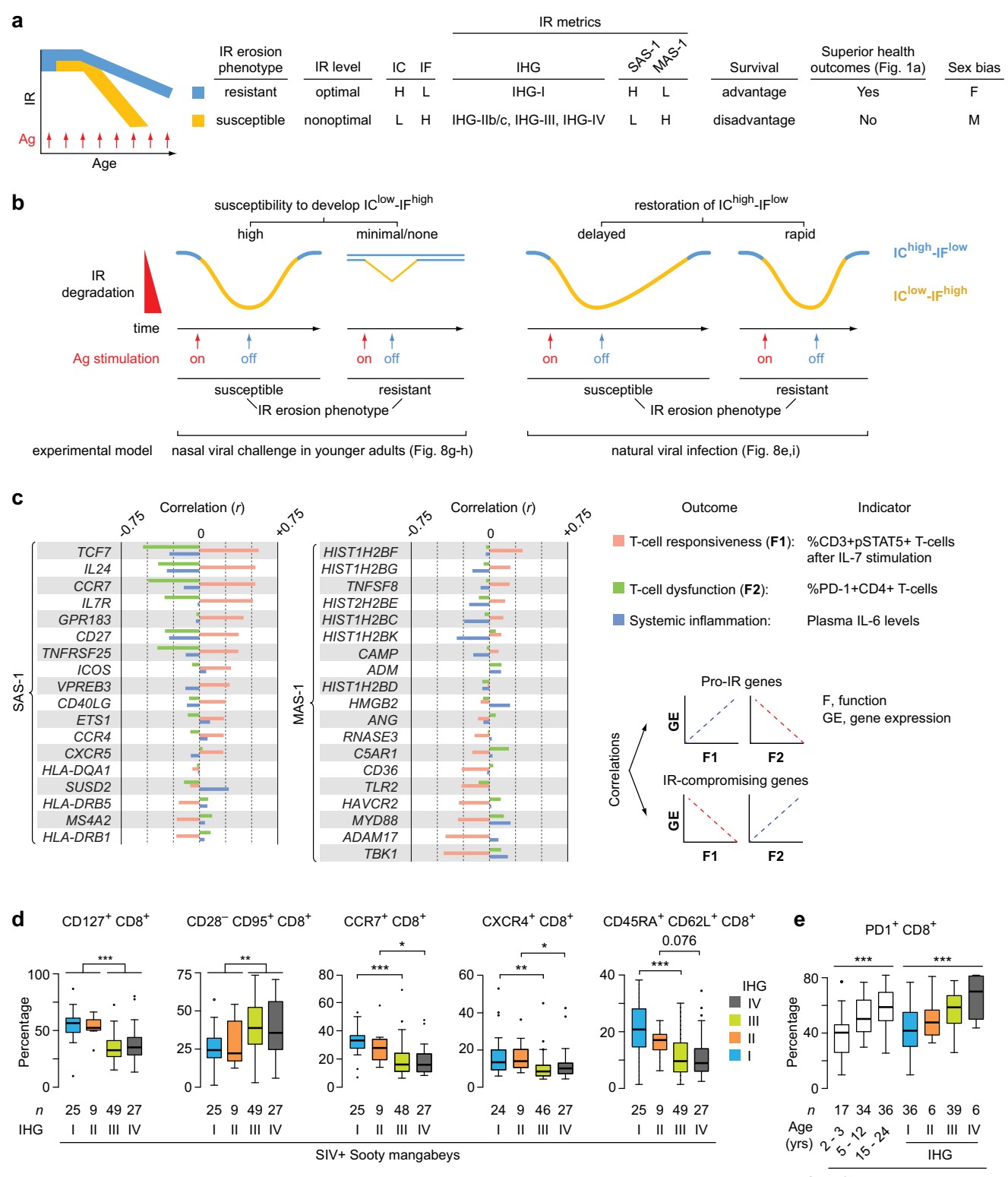

the IR erosion-susceptible phenotype. We examined IR levels and responses in varied human and nonhuman cohorts that are representative of different types and severity of inflammatory (antigenic) stressors. The sum of our findings supports our study framework that optimal IR is an indicator of successful immune allostasis (adaptation) when experiencing inflammatory stressors, correlating with a distinctive immunocompetence-inflammation balance (IC^high-IF^low) that associates with superior immunity-dependent health outcomes, including longevity (Fig. 1a).

The importance of monitoring immune allostasis (IR status) across a person's lifetime is underscored by our finding that, even in young adults with pre-existing optimal IR, the acute inflammatory stress associated with symptomatic common respiratory viral infections (e.g., influenza) is a strong signal for rapid degradation of IR. This IR degradation correlates with a gene expression signature profile (SAS-1^low-MAS-1^high) tracking an IC^low-IF^high status linked to mortality both during aging and COVID-19, as well as immunosuppression (e.g., AIDS). Despite clinical recovery from such common viral infections,

**Fig. 9 | Associations of Immunologic Resilience (IR) erosion phenotypes and immune correlates of IR. a** Schema of features associated with IR erosion phenotypes defined by immune health grade (IHG) status, and survival-associated signature (SAS)-1/mortality-associated signature (MAS)-1 profiles. Ag antigenic, F female, H high, IC immunocompetence, IF inflammation, L low, M male **b** IR erosion-resistant and IR erosion-susceptible phenotypes based on experimental models. **c** Correlation (*r*; Pearson) between expression levels of genes within SAS-1 and MAS-1 signatures with levels of an indicator for T-cell responsiveness, T-cell dysfunction, and systemic inflammation. Measures of T-cell responsiveness, T-cell dysfunction, and plasma IL-6 are from 55, 56, and 50 HIV+ individuals, respectively, on virally suppressive antiretroviral therapy from the early HIV infection cohort. **d**, **e** Levels of the indicated immune traits by IHGs in (**d**) sooty mangabeys seropositive for simian immunodeficiency virus (SIV) and (**e**) SIV-seronegative Chinese rhesus macaques. Comparisons were made between IHG-I vs. IHG-III and IHG-II vs. IHG-IV to mitigate the confounding effects of higher and lower CD4+ counts, respectively. *$P < 0.05$; **$P < 0.01$; ***$P < 0.001$. For box plots: center line, median; box, interquartile range (IQR); whiskers, rest of the data distribution and outliers greater than ±1.5 × IQR are represented as points. Two-sided tests were used. Statistics are outlined in Supplementary Information Section 11.3.9., *P* values are in Supplementary Data 14, and Source data are provided as a Source Data file.

some younger adults were unable to reconstitute optimal IR. Among HIV-seronegative individuals without acute or chronic infections, the mortality/immunosuppression-associated SAS-1^low-MAS-1^high profile was represented in approximately 25% of 15–25-year-olds (Fig. 7c). Taken together, we suggest that a person's lifetime is punctuated by inflammatory signals that degrade IR, predisposing individuals of all ages to manifest, either transiently or permanently, a mortality/immunosuppression-associated gene expression profile. However, since the prevalence of the SAS-1^low-MAS-1^high profile increases steadily with age, it may give the misimpression that this profile relates to the aging process vs. IR degradation.

Consistent with these observations, individuals preserving optimal IR metrics (IR erosion-resistant phenotype) manifested advantages for longevity/survival as well as resistance to severe COVID-19, HIV-AIDS, common acute respiratory viral infections, recurrent skin cancer, and sepsis-associated mortality (expanded discussion in Supplementary Note 9). These advantages were observed after controlling for age, sex, and/or level of antigenic stimulation. Collectively, our findings suggest that the lower immune status observed with age may be driven by two distinct mechanisms, one being dependent on age and the other attributable to IR erosion/degradation, which occurs concurrently with age but is not dependent on age per se (Fig. 10d). Thus, among persons of similar age, an individual's susceptibility to disease risks/severity and mortality may relate to their antecedent and current capacity for preservation and/or restoration of optimal IR when experiencing inflammatory stress.

To test our study framework (Fig. 1a), we evaluated two sets of metrics to gauge IR status – IHGs and SAS-1/MAS-1 gene expression profiles associated with lifespan. These complementary metrics provide an easily implementable method to monitor the IR continuum irrespective of age (Figs. 7a, 9a). Metrics signifying optimal IR relate to two attributes: a balanced CD8+ and CD4+ T-cell count profile, defined by IHG-I (restricted expansion of CD8+ T-cells with higher CD4+ counts), and a gene expression profile (SAS-1^high-MAS-1^low) aligned with an IC^high-IF^low status associated with longevity. Paralleling the observation that females manifest advantages for immunocompetence and longevity[2–5], the IR erosion-resistant phenotype was more common in females (including postmenopausal). We also noted IR responses suggesting that IR preservation/degradation has ancient evolutionary origins (Fig. 4c–f). Congruently, immune traits associated with some nonoptimal IR metrics were similar in humans and nonhuman primates. Additionally, in Collaborative Cross-RIX mice, the IR erosion-resistant phenotype was associated with resistance to lethal Ebola and severe influenza infection.

We accrued direct evidence of the benefits of optimal IR during exposure to a single inflammatory stressor by examining young adults during experimental intranasal challenge with common respiratory viruses (e.g., influenza, rhinovirus, RSV; Fig. 8g–h). Following challenge with the viral inocula used in these experimental studies, only about 50% of the individuals became symptomatic (Fig. 8g–h). The hallmark of asymptomatic status after intranasal inoculation of respiratory viruses was the capacity to preserve, enrich, or rapidly restore the survival-associated SAS-1^high-MAS-1^low profile (Figs. 8g–h, 9b). In contrast, in these challenge studies, symptomatic viral infections in most

young adults associated with the rapid degradation of IR as exemplified by the induction of the mortality/immunosuppression-associated gene expression profile (SAS-1^low-MAS-1^high).

Could incomplete reconstitution of optimal IR following inflammatory stress explain why some younger individuals manifest suboptimal/nonoptimal IR (unsuspected immunosuppression)? Findings noted on longitudinal monitoring of IR degradation and reconstitution during natural infection with common respiratory viruses supported this possibility. With symptomatic infection, nearly all individuals manifested the mortality/immunosuppression-associated gene expression profile (Fig. 8e–i). During recovery, reconstitution of optimal IR was greater and faster in persons who before infection had the survival-associated SAS-1^high-MAS-1^low vs. the SAS-1^low-MAS-1^high profile (Fig. 8e). However, despite the elapse of several months from initial infection, some younger persons with the SAS-1^high-MAS-1^low profile before infection failed to reconstitute this profile (exemplifying residual deficits in IR) (Fig. 8e). An impairment in the capacity for reconstitution of optimal IR was also observed in prospective cohorts with other inflammatory contexts (FSWs, COVID-19, HIV infection). These findings support our viewpoint that the deviation from optimal IR that tends to occur with age could be due to an impairment in the reconstitution of IR in individuals with the IR erosion-susceptible phenotype (Fig. 1c; expanded supplementary discussion in Supplementary Note 9).

After intranasal inoculation with SAS-CoV-2, nearly 50% of the recipients were uninfected[58]. There is significant interest in identifying host genetic factors that mediate resistance to acquiring SARS-CoV-2 or developing severe COVID-19[59–61]. In addition to the identified genetic factors, our data from experimental viral infection challenges, coupled with our findings in patients with acute COVID-19, suggest a complementary possibility: resistance to SARS-CoV-2 acquisition and/or severe COVID-19 relates to the IR erosion-resistant phenotype. We are currently investigating whether failure to reconstitute optimal IR after acute COVID-19 may contribute to postacute sequelae.

Favorable outcomes during HIV exposure and after infection may serve as indicators or proxies for the elite capacity to resist degradation of optimal IR despite the inflammatory stress associated with HIV risk factors and/or viremia. Resistance to HIV acquisition despite exposure to the virus is a distinctive trait[62] observable in some FSWs. Among FSWs with comparable levels of risk factor-associated antigenic stimulation, HIV seronegativity was an indicator of the IR erosion-resistant phenotype, whereas seropositivity was an indicator of the IR erosion-susceptible phenotype. Having baseline IHG-IV, a nonoptimal IR metric, associated with a nearly 3-fold increased risk of subsequently acquiring HIV, after controlling for level of risk factors. We found that a subset of FSWs had the capacity for preservation of optimal IR, both before and after HIV infection. Preservation of optimal IR associated with a lower HIV-VL during primary/early HIV infection and, since HIV-VL predicts AIDS progression rates[63], slower AIDS progression rates as well. Thus, restriction of HIV replication and progression to AIDS may serve as indicators of HIV+ persons with the IR erosion-resistant phenotype. By analogy, we suggest that CMV seropositivity may have similar indicator functions (Supplementary Notes 3, 9). While CMV seropositivity has been associated with age-

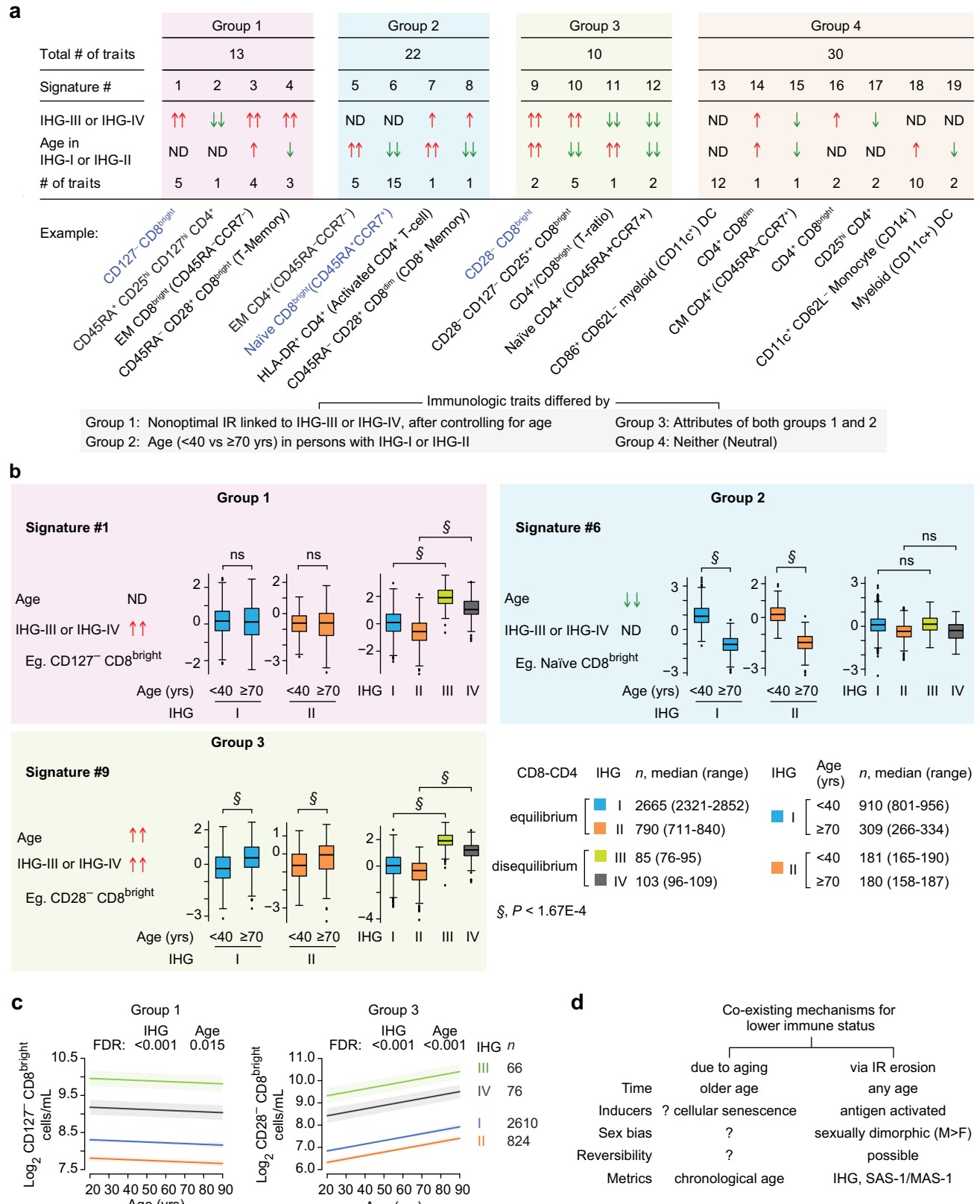

a

Immunologic traits differed by

Group 1: Nonoptimal IR linked to IHG-III or IHG-IV, after controlling for age
Group 2: Age (<40 vs ≥70 yrs) in persons with IHG-I or IHG-II
Group 3: Attributes of both groups 1 and 2
Group 4: Neither (Neutral)

b

**Group 1**

Signature #1

**Group 2**

Signature #6

**Group 3**

Signature #9

CD8-CD4 IHG n, median (range)

equilibrium
I 2665 (2321-2852)
II 790 (711-840)

disequilibrium
III 85 (76-95)
IV 103 (96-109)

IHG / Age (yrs) / n, median (range)
I <40 910 (801-956)
≥70 309 (266-334)
II <40 181 (165-190)
≥70 180 (158-187)

§, P < 1.67E-4

c

Group 1
IHG FDR: <0.001 Age 0.015

Group 3
IHG FDR: <0.001 Age <0.001

IHG n
III 66
IV 76
I 2610
II 824

d

Co-existing mechanisms for lower immune status

| | due to aging | via IR erosion |
|---|---|---|
| Time | older age | any age |
| Inducers | ? cellular senescence | antigen activated |
| Sex bias | ? | sexually dimorphic (M>F) |
| Reversibility | ? | possible |
| Metrics | chronological age | IHG, SAS-1/MAS-1 |

associated diseases and mortality[22-25], we suggest that these associations are tracking CMV+ persons with the IR erosion-susceptible phenotype (expanded discussion in Supplementary Notes 3, 9).

The IR framework points to the commonalities in the HIV and COVID-19 pandemics. Our findings suggest that these pandemics may be driven by individuals who had IR degradation before acquisition of

viral infection. With respect to the HIV pandemic, nonoptimal IR metrics are overrepresented in persons with behavioral and non-behavioral risk factors for HIV, and these metrics predict an increased risk of HIV acquisition. Correspondingly, HIV burden is greater in geographic regions where the prevalence of nonbehavioral risk factors is also elevated (e.g., schistosomiasis in Africa[64]). With respect to the

**Fig. 10 | Immune traits associated with immune health grade (IHG)-III or IHG-IV vs age vs both. a** In the HIV− SardiNIA cohort, 75 immune traits categorized into four groups. Within each group, traits were clustered into signatures according to whether their levels were higher or lower with IHG-III or IHG-IV, after controlling for age and sex; by age in older or younger persons with IHG-I or IHG-II, after controlling for sex; both; or neither. cDC, conventional dendritic cells. Arrows indicate significant difference at $P < 1.67E\text{-}4$; ND indicates no significant difference at $P < 1.67E\text{-}4$. Two arrows indicate both comparisons for IHG-I vs. IHG-III and IHG-II vs. IHG-IV or age within IHG-I and IHG-II are significant, one arrow indicates only one of the comparisons for IHG status or age is significant. **b** Representative traits by age in persons with IHG-I or IHG-II and by IHG status. Comparisons for the indicated traits were made between IHG-I vs. IHG-III and IHG-II vs. IHG-IV to mitigate the confounding effects of higher and lower CD4+ counts, respectively. Trait levels (y-axis) were normalized using inverse normal transformations with values ranging from −3 to 3; boxplots show covariate-adjusted residuals. Median number of individuals evaluated by IHG status and age within IHG-I or IHG-II. ns nonsignificant. **c** Linear regression was used to analyze the association between $\log_2$ transformed cell counts (outcome) with age and IHG status (predictors). The linear model was used to generate the fitted lines and 95% confidence bands and significance was determined by likelihood ratio test. FDR, false discovery rate $P$ values adjusted for multiple comparisons. **d** Model differentiating features of processes associated with lower immune status that occur due to aging or via erosion of IR. SAS-1, survival-associated signature-1; MAS-1, mortality-associated signature-1. For box plots: center line, median; box, interquartile range (IQR); whiskers, rest of the data distribution and outliers greater than ±1.5 × IQR are represented as points. Two-sided tests were used. Statistics are outlined in Supplementary Information Section 11.3.10., $P$ values are in Supplementary Data 14, and Source data are provided as a Source Data file.

COVID-19 pandemic, the proportion of individuals preserving optimal IR metrics decreases with age and age serves as a dominant risk factor for developing severe acute COVID-19. Controlling for age, the likelihood of being hospitalized was significantly lower in individuals preserving optimal IR at diagnosis with COVID-19. Thus, individuals with the IR erosion-susceptible phenotype may have contributed substantially to the burden of these pandemics.

Our study has several limitations (expanded limitations in Supplementary Note 10). The primary limitation is our inability to examine the varied clinical outcomes assessed here in a single prospective human cohort. Such a cohort that spans all ages with these varied inflammatory stressors and outcomes is nearly impossible to accrue, necessitating the juxtaposition of findings from varied cohorts. Additionally, we were unable to evaluate immune traits in peripheral blood samples bio-banked from the same individual when they were younger vs. older. However, we took several steps to mitigate this limitation (discussed in Supplementary Notes 2, 8). Furthermore, we cannot ascribe cause-effect relationships (eroded IR → inferior immunity-dependent health outcomes). However, our findings satisfy the nine Bradford-Hill criteria[65], the most frequently cited framework for causal inference in epidemiologic studies (Supplementary Note 11). We acknowledge that, in addition to inflammatory stressors, the changes in IR metrics observed during aging (Figs. 2f, 7b–d) might be driven by multiple factors, such as thymic involution and decline in stem cell production, as well as cell proliferation, tissue migration, and residency of memory cells[66]. Possible confounders regarding the generation of the IHGs and their distribution patterns in varied settings of increased antigenic stimulation are discussed (Supplementary Notes 1, 2, 3 and 6). While we focused on the association between antigenic stimulation associated with inflammatory stressors and shifts in IHG status, psychosocial stressors may contribute, as they associate with age-related T lymphocyte percentages in older adults[67]. However, the latter lymphocyte changes can be indirect, as psychosocial stressors may predispose to infection[68,69]. As a final limitation, we could not evaluate whether eroded IR mitigates autoimmunity. This possibility needs consideration, as autoimmunity rates are higher in females, a risk group that is more likely to preserve/manifest optimal IR.

Supporting our conclusion that age-independent mechanisms contribute to IR status, we provide evidence that host genetic factors (in MHC locus) associate with the IR erosion phenotypes (Supplementary Note 5). Our suggestion that a lower immune status may be due to eroded/degraded IR vs. age (Fig. 10d) has two practical implications. First, while a significant effort is placed on targeting the immune traits associated with age, we show that immune traits group into those associated (i) uniquely with IR status irrespective of age, (ii) uniquely with age, and (iii) both age and IR status (Fig. 10a−c). Some of the immune traits that associate with uniquely nonoptimal IR metrics have been misattributed to age (e.g., signatures 1 to 4; Fig. 10a). Hence, a comparison of immune traits between younger and older persons

conflates these groupings, obscuring the immune correlates of age. Second, the reversibility of eroded IR suggests that immune deficits linked to this erosion are separable from those linked directly to the aging process and may be more amenable to reversal. However, our findings in FSWs and during natural respiratory viral infections indicate that this reversal may take months to years to occur. Additionally, data from FSWs and sooty mangabeys illustrate that multiple sources of inflammatory stress have additive negative effects on IR status (Fig. 4c). Hence, reconstitution of optimal IR may require cause-specific interventions. For example, reconstitution of optimal IR metrics in HIV+ individuals may require a two-pronged strategy, reversing the IR degradation due to HIV risk factors and HIV viremia. The immunosuppression of HIV infection is potentially reversible: the mortality/immunosuppression-associated gene expression profile was underrepresented, whereas the survival-associated profile was overrepresented in HIV+ persons who reconstituted optimal IR (IHG-I) during antiretroviral therapy (Fig. 7g).

In summary, our findings support the principles of our framework (Fig. 1a) wherein IR is a trait distinct from processes that anchor immune status or other attributes (e.g., epigenetic modifications) to chronologic age as a reference (e.g., biological age[70–72], inflammaging[73], and premature aging[74]). The principles of our framework are intuitive with health/survival implications irrespective of age, sex, underlying comorbidities, and HIV or CMV serostatus. Irrespective of these factors, most individuals do not have the capacity to preserve optimal IR when experiencing common inflammatory insults such as symptomatic viral infections. Deviations from optimal IR associates with an immunosuppressive-proinflammatory, mortality-associated gene expression profile. This deviation is more common in males. Those individuals with capacity to resist this deviation or who during the recovery phase rapidly reconstitute optimal IR manifest health and survival advantages. However, under the pressure of repeated inflammatory (antigenic) stressors experienced across their lifetime, the number of individuals who retain capacity to resist IR degradation declines. Persons with residual deficits in IR (suboptimal/nonoptimal IR) have health and survival disadvantages.

How might these framework principles inform personalized medicine, development of therapies to promote immune health, and public health policies? First, individuals with suboptimal or nonoptimal IR can potentially regain optimal IR through reduction of exposure to infectious, environmental, behavioral, and other stressors. Second, IR metrics provide a means to gauge immune health regardless of age, sex, and underlying comorbid conditions. Disease risks/severity (e.g., risk of recurrent skin cancer, COVID-19 severity) and responses to therapies may differ according to IR status. Thus, early detection of individuals with IR degradation could prompt a work-up to identify the underlying inflammatory stressors. Third, balancing trial and placebo arms of a clinical trial for IR status may mitigate the confounding effects of this status on outcomes that are dependent on differences in immunocompetence and inflammation. Fourth, while

senolytic agents are being investigated for the reversal of age-associated pathologies[75], the findings presented herein provide a rationale to consider the development of strategies that, by targeting the IR erosion-susceptible phenotype, may improve vaccine responsiveness, healthspan, and lifespan. Finally, population-level differences in the prevalence of IR metrics may help to explain the racial, ethnic, and geographic distributions of diseases such as viral infections and cancers. Hence, strategies for improving IR and lowering recurrent inflammatory stress may emerge as high priorities for incorporation into public health policies.

## Methods

### Ethics and IRB approval

All studies were approved by the institutional review boards (IRBs) at the University of Texas Health Science Center at San Antonio and institutions participating in this study. The IRBs of participating institutions are listed in the reporting summary. All studies adhered to ethical and inclusion practices approved by the local IRB.

### Cohorts

The cohorts and study groups (Fig. 1a) were assembled to evaluate the hypothesis and outcomes of this study noted in Fig. 1a. In total, $n = 48,936$ human subjects/samples, 279 non-human primates, and 378 mice were studied. The source of the cohorts/participants are summarized in Supplementary Data 1. Baseline characteristics of all adult HIV− cohorts are listed in Supplementary Data 2.

### SardiNIA

The SardiNIA study investigates genotypic and phenotypic aging-related traits in a longitudinal manner. The main features of this project have been described in detail previously[9,76,77]. All residents from 4 towns (Lanusei, Arzana, Ilbono, and Elini) in a valley in Sardinia (Italy) were invited to participate. By November 2001, the recruited participants correspond to approximately >60% of the population eligible for recruitment in the area. Immunophenotype data from 3896 participants (age 15 to 103 years) were included in this study. Details provided in Supplementary Information Section 1.1.1.

### Kenya Majengo observational cohort study (MOCS)

The Majengo sex worker cohort[17] is an open cohort dedicated to better understanding the natural history of HIV infection, including defining immunologic correlates of HIV acquisition and disease progression. The present study comprised 1050 initially HIV-negative FSWs with data available for analysis and were evaluated from the time they were enrolled (see criteria in Supplementary Fig. 1c). Of these, 127 subsequently seroconverted. 762 HIV-negative FSWs had both CD4+ count and CD4:CD8 ratio values available, as well as baseline risk behavior data. The characteristics of these 762 FSWs are listed in Supplementary Data 4a. The association of risk behavior (e.g., duration of sex work, frequency of condom use, clients per week) with prevalence of CD8-CD4 disequilibrium grades IHG-III and IHG-IV, as well as future HIV seroconversion were evaluated in these 762 FSWs. To mitigate confounding, in the Results section, we show data for the 449 women who met the following criteria: had concurrent CD4+ T-cell count and CD4:CD8 ratio measurements, as well as risk behavior data and at least 2 HIV seronegative follow-up visits 3 months apart. Among these, 53 subsequently seroconverted. Prior to seroconversion, the 53 FSWs were followed for 309.31 person-years; 396 FSWs who remained HIV− were followed for 1,664.81 person-years. The characteristics of these 449 FSWs are listed in Supplementary Data 4b. Of the 53 FSWs who seroconverted, 43 had at least one CD4+ and CD8+ T-cell count measurement within 1 year of the seroconversion date and the data from these participants are presented in Fig. 4c in Results section. Details provided in Supplementary Information Section 1.1.2.

### Renal transplant recipient (RTR) cohort

To investigate the associations of IHG status with cancer development, we assessed the hazard of developing CSCC within a predominantly White cohort of long-term RTRs. A total of 114 RTRs with available clinical and immunological phenotype were evaluated. The characteristics of the RTRs are as described previously[15] and summarized in Supplementary Data 5. Briefly, 65 eligible RTRs with a history of post-transplant CSCC were identified, of whom 63 were approached and 59 participated. Seventy-two matched eligible RTRs without a history of CSCC were approached and 58 were recruited. Fifteen percent of participants received induction therapy at the time of transplant, and 80 percent had received a period of dialysis prior to transplantation. Details provided in Supplementary Information Section 1.1.3.

### Cohort of Kenyan HIV− children with Schistosomiasis

The Kenyan HIV− children with *S. haematobium* urinary tract infection were from a previous study[78]. Briefly, all participants were examined by ultrasound for *S. haematobium* infection and associated morbidity in the Msambweni Division of the Kwale district, southern Coast Province, Kenya, an area where *S. haematobium* is endemic. No community-based treatment for schistosomiasis had been conducted during the preceding 8 years of enrollment in this population. From this initial survey, we selected all children 5–18 years old residing in 2 villages, Vidungeni and Marigiza, who had detectable bladder pathology and *S. haematobium* infection. Details provided in Supplementary Information Section 1.1.4.

### HIV− cohort from the University of California San Diego (UCSD)

The HIV-seronegative UCSD cohort was accessed from HIV Neurobehavioral Research Center, UCSD, and derived from the following three resources: (a) those who enrolled as a normative population for ongoing studies funded by the National Institute of Mental Health; (b) those who enrolled as a normative population for studies funded by the National Institute on Drug Abuse; and (c) those who enrolled as HIV− users of recreational drugs for studies funded by the National Institute on Drug Abuse. In the present study, we evaluated 759 participants pooled from the three abovementioned sources. Details provided in Supplementary Information Section 1.1.5.

### South Texas veterans health care system COVID-19 cohort

Details provided in Supplementary Information Section 1.1.7. This was a prospective observational cohort study of patients testing positive for SARS-CoV-2 evaluated at the Audie L. Murphy VA Medical Center, South Texas Veterans Health Care System (STVHCS), San Antonio, Texas, from March 20, 2020, through November 15, 2020. Patients were followed during hospitalization and/or a minimum of 30 days from inclusion. The cohort characteristics and samples procedures are described in Supplementary Data 2 and Supplementary Data 7. The cohort features of a smaller subset of patients studied herein and samples procedures have been previously described[6]. COVID-19 progression along the severity continuum was characterized by hospitalization and death. Standard laboratory methods in the Flow Cytometry Core of the Central Pathology Laboratory at the Audie L. Murphy VA Medical Center were used to determine peripheral blood CD4+ and CD8+ T-cell levels. The overview of this cohort is shown in Supplementary Fig. 1d. All measurements evaluated in the present study were conducted prior to the availability of COVID-19 vaccinations. RNA-Seq was performed on a subset of this cohort as previously described[6]. The phenotype and processed gene expression data generated are available in the Figshare database at the following accession link (https://figshare.com/projects/Ahuja_Lab_COVID-19_dataset/158732).

### Primary infection cohort (PIC) from University of California, San Diego

The PIC cohort comprised 723 HIV+ participants[7]. These participants were recruited between June 1996 and June 2010 and then followed prospectively. Details of the cohort are as described previously[7]. We evaluated only participants in whom an estimated date of infection could be calculated through a series of well-defined stepwise rules that characterize stages of infection based on our previously described serologic and virologic criteria[7]. Of the 723 participants, 685 were evaluated in the present study while they were therapy-naïve (see criteria in Supplementary Fig. 1a; Supplementary Data 6). 194 who commenced ART between April 26, 1997 and April 26, 2013 (with clinical data collected until October 13, 2014), and met other inclusion criteria were also evaluated (Supplementary Data 6). The inclusion criteria are outlined in Supplementary Fig. 1a. Participants in the cohort self-selected ART or no ART, and those who chose not to start therapy were followed in a manner identical to those who chose to start ART. Rules of computing time to estimated date of infection are as reported by us previously[7]. 75 of the 194 were on ART for at least 4 years and had at least 1 IHG measurement in each of those 4 years. Details provided in Supplementary Information Section 1.2.1.

### Early HIV infection cohort (EIC)

The US Military HIV Natural History Study is designated as the EIC. This is an ongoing, continuous-enrollment, prospective, multicenter, observational cohort study conducted through the Uniformed Services University of the Health Sciences Infectious Disease Clinical Research Program. The EIC has enrolled approximately 5723 active-duty military service members and beneficiaries since 1986 at 7 military treatment facilities (MTFs) throughout the United States. The US military medical system provides comprehensive HIV education, care, and treatment, including the provision of ART and regular visits with clinicians with expertise in HIV medicine at MTFs, at no cost to the patient. Mandatory periodic HIV screening according to Department of Defense policy allowed treatment initiation to be considered at an early stage of infection before it was recommended practice. Eighty-eight percent of the participants since 1995 have documented seroconversion (i.e., a documented negative HIV test preceding a positive HIV test), with a median seroconversion window of approximately 15 months. In the present study, 4883 of 5723 EIC participants were available for evaluation (Supplementary Fig. 1b; Supplementary Data 6). Details provided in Supplementary Information Section 1.2.2.

Additional details of the SardiNIA[9,76,77], FSW-MOCS[17], PIC-UCSD[7], RTR cohort[15], *S. haematobium*-infected children cohort[78], and EIC[8,79–82] have been described previously. Some features of the entire populations or subsets of the SardiNIA, COVID-19, SLE (Supplementary Information Section 8.3.1), FHS (Supplementary Information Section 8.3.2), San Antonio Family Heart Study (Supplementary Information Section 8.3.3), and HIV cohorts studied herein have been described previously by us in a recent study[6].

### Sooty mangabeys

One hundred sixty sooty mangabeys were evaluated in the current study. Of these, 50 were SIV seronegative (SIV−) and 110 were naturally infected with SIV (Figs. 4c–d, 9d). Data from a subset of these sooty mangabeys have been reported by Sumpter et al.[18]. All sooty mangabeys were housed at the Yerkes National Primate Research Center and maintained in accordance with National Institutes of Health guidelines. In uninfected animals, negative SIV determined by PCR in plasma confirmed the absence of SIV infection. Based on longitudinal serologic surveys, the majority of SIV+ sooty mangabeys are known to have acquired their infection by 3 to 4 years of age. IL-7R (CD127) levels of CD8+ T-cells, as well as CD8+ effector T-cells, were assessed as the proportion of CD127+CD8+ T-cells and CD28−CD95+CD8+ T-cells, respectively (Fig. 9d). Other immune traits studied are reported in

Supplementary Data 11. Details provided in Supplementary Information Section 1.3.1.

### Chinese rhesus macaques

Forty-seven male and 40 female SIV− Chinese rhesus macaques from a previous study were evaluated (Fig. 4e)[19]. All animals were colony-bred rhesus macaques (*M. mulatta*) of Chinese origin. All animals were without overt symptoms of disease (tumors, trauma, acute infection, or wasting disease); estrous, pregnant, and lactational macaques were excluded. A marker of T-cell dysfunction was evaluated, which was expression levels of PD-1 on CD8+ T-cells (Fig. 9e; Supplementary Fig. 14), using methods described previously[19]. Details provided in Supplementary Information Section 1.3.2.

### Collaborative cross intercrossed (CC-RIX) mice (Ebola infection)

In a study by Rasmussen et al.[83], the role of mice genetics in Ebola virus disease was studied using the CC resource[83]. CC is a genetically diverse panel of recombinant inbred (CC-RI) mice obtained through a systematic cross of 8 inbred founder mouse strains, 5 of which are classic laboratory strains (C57BL/6J, A/J, 129S1/SvImJ, NOD/ShiLtJ, and NZO/H1LtJ) and 3 of which are wild-derived inbred strains (CAST/EiJ, PWK/PhJ, and WSB/EiJ). Different strains were crossed with one another to generate CC-RIX F1 progeny. The authors reported that the CC-RIX mice exhibited distinct disease phenotypes after mouse-adapted Ebola virus infection and the phenotypes ranged from complete resistance (0% mortality) to lethal disease (>50% mortality). Details provided in Supplementary Information Section 1.4.

### Immunologic resilience (IR) metric: immune health grades (IHGs).

IHGs I to IV, which reflect the relative proportions of CD8+ and CD4+ T-cell counts that are not inferable through assessments of these markers or the CD4:CD8 ratio alone (Fig. 2b). The cutoffs for the IHGs are based on 2 principles: (i) a CD4:CD8 ratio value of less than unity (<1.0) is a mathematical representation of or proxy for higher CD8+ T-cell counts that are uncompensated for by higher CD4+ counts, and (ii) 800 CD4+ cells/mm³ approximated the median CD4+ T-cell count in 16,126 HIV-seronegative (HIV−) persons (Supplementary Table 1)[6–8]. We selected those cutoffs based on the following rationale.

We previously reported that CD4+ T-cell count of approximately 800 cells/mm³ approximated the lower bounds of the median CD4+ T-cell count in >12,000 HIV uninfected participants[7,8]; this survey was conducted by a Medline search. On this basis, we had previously used CD4+ count of 800 cells/mm³ as an outcome during ART of HIV+ persons[7]. Additionally, we found that attainment of CD4+ counts equal to or above 800 cells/mm³ associated with restoration of markers of T-cell health to levels observed in HIV-seronegative persons[8].

Most HIV-seronegative individuals maintain a CD4:CD8 T-cell ratio ≥1.0[30,84], sometimes even in conditions associated with CD4+ lymphocytopenia (e.g., infections other than HIV-1, malignancy)[85]. In sharp contrast, a universal feature of untreated chronic HIV infection is the inversion of the CD4:CD8 T-cell ratio (<1.0)[86–88], and an inverted ratio in HIV-seronegative participants correlates with adverse events[44,86,89–95]. Additional details regarding the IHGs are described in Supplementary Note 1.

### Immunophenotyping.

Immune correlates (markers) that associated with IHG status vs. age in the SardiNIA cohort were assessed on fresh blood samples. Cells were processed within 2 h after sample collection to avoid time-dependent artefacts. A set of multiplexed fluorescent surface antibodies was used to characterize the major leukocyte cell populations circulating in peripheral blood belonging to both adaptive and innate immunity. Briefly, with the antibody panel designated as T-B-NK in Supplementary Data 12, we identified T-cells, B-cells, and NK-cells and their subsets. We also used the HLA-DR marker to assess the activation status of T and NK cells. The regulatory T-cell panel (Treg in

Supplementary Data 12) was used to characterize regulatory T-cells subdivided into resting, activated, and secreting nonsuppressive cells[96,97]. Moreover, in selected T-cell subpopulations, we assessed the positivity for the ectoenzyme CD39 and the CD28 co-stimulatory antigen[98]. The antibody panel labelled T-cell maturation (Mat) in Supplementary Data 12 accounted for the chemokine receptor CCR7 and the CD45RA marker to distinguish between naïve, central memory (CM), effector memory (EM), and terminally differentiated (TD) subsets in CD4[+] and CD8[bright] and CD4[−]CD8[−] T-cells[99]. Finally, by the circulating dendritic cells (DC) panel, we divided circulating DCs into myeloid (conventional DC, cDC) and plasmacytoid DCs (pDC) and assessed the expression of the adhesion molecule CD62L and the co-stimulatory ligand CD86[100,101]. The circulating DC panel is labelled DC in Supplementary Data 12. Overall, through this process we measured 75 distinct informative, immune traits/markers (Supplementary Data 12). Detailed protocols and reproducibility of the measurements have been described[9]. Leukocytes were characterized on whole blood by polychromatic flow cytometry with 4 antibody panels, namely T-B-NK, regulatory T-cells (Treg), Mat, and circulating DCs, as described elsewhere[9] and detailed in Supplementary Information Section 5.

**T-cell responsiveness, T-cell dysfunction, and systemic inflammation.** These indicators were evaluated in a subset of EIC participants whose characteristics were described previously[8,43]: ART-naïve controls ($n = 28$), individuals virally suppressed with long-term ART ($n = 124$), and HIV-uninfected controls ($n = 13$), as described in Supplementary Information Section 1.1.6. Integrity of the IL-7/IL-7 receptor axis (T-cell responsiveness) was investigated by determining the level of responsiveness of T-cells assessed as the proportion of T-cells responding to IL-7, based on the percentage of CD3[+] T-cells positive for phosphorylated signal transducer and activator of transcription (STAT5) (%CD3[+]pSTAT5[+] T-cells) after in vitro stimulation of peripheral blood mononuclear cells with IL-7. IL-7 is a critical T-cell trophic cytokine. Methods were as described previously[8,43]. Levels of T-cell exhaustion (dysfunction) were assessed as proportion of CD4[+] T-cells positive for programmed cell death protein-1 (PD1) (%CD4[+]PD1[+] T-cells) (proxy for exhaustion). Systemic inflammation was assessed by measuring plasma IL-6 levels using Luminex assays, employing methods described by the manufacturer. Further details are provided in Supplementary Information Section 6.

**RNA-seq in HIV+ persons and healthy controls.** RNA-seq analysis was performed in the designated groups (See Supplementary Information section 7.2 and Supplementary Fig. 1b). Total RNA was isolated from PBMCs using the RNeasy Mini Kit (Qiagen, Hilden, Germany) with DNase I digestion according to the manufacturer's instructions. RNA quantity and purity were determined using an Agilent 2100 Bioanalyzer with an RNA 6000 Nano assay (Agilent Technologies, Palo Alto, CA). Samples with integrity RIN ≥ 7.0 were selected for RNA sequencing.

## Library preparation and sequencing

A double-stranded cDNA library was prepared starting with 1 μg of total RNA input according to the TrueSeq RNA v2 sample preparation kit protocol (Illumina, San Diego, CA). Briefly, mRNA was selected using poly-T oligo-attached magnetic beads and then enzymatically fragmented. First and second cDNA strands were synthesized and end-repaired. Multiplexed adaptors were ligated after 3′-adenylation. The library with adaptors was enriched by PCR. Libraries were size checked using a DNA high-sensitivity assay on the Agilent 2100 Bioanalyzer (Agilent Technologies, Palo Alto, CA) and quantified by a Kapa Library quantification kit (Kapa Biosystems, Woburn, MA). Libraries were clustered using the Illumina cBot (Illumina, San Diego, CA) and then paired-end sequenced (2 × 101 bp) on an Illumina HiSeq 2000. Base

calling and quality filtering were performed using the CASAVA v1.8.2 (Illumina) pipeline. Sequences were aligned and mapped to the UCSC hg19 build of the Homo sapiens genome (from Illumina igenomes) using tophat v2.0.1[102]. Gene counts for 23,239 unique, well-curated genes were obtained using HTSeq framework v0.5.3P3. (https://htseq.readthedocs.io/en/master/history.html).

## Normalization and quality control for RNA-seq data

Gene counts were normalized, and dispersion values were estimated using the R package, DESeq v1.10.1[103]. The design matrix (row – samples; column – experimental variables) used in DESeq, along with gene-expression matrix (row – genes; column – gene counts in each sample), included the group variable (therapy-naïve, HIV–, IHG), CMV serostatus, and the personal identification number, all as factors, and other variables. Genes with a gene count of 0 across all samples were removed; the remaining zeros (0) were changed to ones (1) and these genes were used in the gene-expression matrix in DESeq. The size factors were estimated using the gene-expression matrix taking library sizes into account; these were used to normalize the gene counts. Genes with expression levels <25% of total expression from all samples were removed, leaving a total of 15,610 genes evaluated for differential expression. Of note, the filtered genes are expressed at low levels across all samples and would not be differentially expressed (FDR < 0.05) in comparisons. The dispersion factors were estimated using the options: method = blind and sharingMode = fit-only, as there were too many variables (due to personal identification numbers) to use the default. Cross-sectional differences between the groups were assessed. The correlation of genes with functional markers (T-cell responsiveness, T-cell dysfunction, and systemic inflammation) was assessed in a subset of this cohort and is detailed in Supplementary Information Section 7.5. The phenotype and processed gene expression data for the dataset are available in the Figshare database at the following accession link https://figshare.com/projects/Ahuja_Lab_HIV_dataset/158681.

## Transcriptomic signature scores

Details for deriving transcriptomic signature scores are in Supplementary Information Section 8.

**SAS-1 and MAS-1.** From our previous work on immunologic resilience in COVID-19[6], 3 survival-associated signatures (SAS) and 7 mortality-associated signatures (MAS) were derived from peripheral blood transcriptomes of 48 patients of the COVID-19 cohort. Of these, the topmost hits in each category (SAS-1 and MAS-1) were used in this study. Briefly, a generalized linear model based on the negative binomial distribution with the likelihood ratio test was used to examine the associations with outcomes: non-hospitalized [NH], hospitalized [H], nonhospitalized survivors [NH-S], hospitalized survivors [H-S], hospitalized-nonsurvivors [H-NS], and all nonsurvivors [NS] at 120 days. FDR < 0.05 cutoff was used to identify differentially expressed (DE) genes between the comparisons.

Genes that were DE (FDR < 0.05) between H vs. NH groups (genes associated with hospitalization status), and H-NS vs. H-S (genes associated with survival in hospitalized patients) were identified. Next, in peripheral blood transcriptomes, genes that were DE between H-S vs. NH-S, NS vs. H-S, and NS vs. NH-S groups were identified and the genes that overlapped in these comparisons with a concordant direction of expression were examined. This approach allowed us to identify genes that track from less to greater disease severity and vice versa (i.e., NH-S > H-S > NS vs. NS > H-S > NH-S). Note: NS in this analysis include both NH and H patients who died. DAVID v6.8[104,105] with default settings except for selection of biological process (BP) gene ontology (GO) terms (GO-BP terms) was used to identify GO-BP terms associated with differentially expressed genes that had a concordant direction of response at an FDR < 0.05.

Based on the differentially expressed genes identified in each comparison and their direction of expression (upregulated vs. downregulated) in the study group comparisons, a filtering process was applied to reduce the number of redundant GO-BP terms: a GO-BP term with a lower significant FDR (higher $P$ values) was filtered if at least 75% of the genes in them were represented in another GO-BP term with a more significant FDR (lower $P$ values). The filtering resulted in 51 GO-BP terms (51 sets of gene signatures) and 1 signature set of 28 genes, the top 52 gene signatures.

After adjusting for age and sex, as well as controlling for multiple comparisons (FDR correction), 29 signatures and 16 signatures out of the 52 signatures significantly associated (FDR < 0.05) with hazard of mortality in the COVID-19 cohort and FHS cohort, respectively (Supplementary Data 9a). Ten signatures overlapped between both cohorts and were further examined. Supplementary Data 9b describes the gene compositions of the 3 SAS and 7 MAS gene signatures. Of these 10 signatures, the 3 signatures that associated (after controlling for age/sex) with lower and the 7 signatures that associated with higher mortality hazards in both cohorts were termed as, SAS and MAS, respectively. SASs and MASs were numbered according to their prognostic capacity for predicting survival or mortality, respectively in the FHS [lowest to highest Akaike information criteria; SAS-1 to SAS-3 and MAS-1 to MAS-7] (Supplementary Data 9c–d). The top associated signature in each category (SAS-1 and MAS-1) were used in this study as $z$-scores. SAS-1 and MAS-1 correspond to the gene signature #32 (immune response) and #4 (defense response to gram-positive bacterium), respectively, as detailed in our recent report[6]. SAS-1 genes ($n = 21$): *CCL4L2, CCR4, CCR7, CD27, CD40LG, CXCL8, CXCR5, ETS1, GPR183, HLA-DQA1, HLA-DRB1, HLA-DRB5, ICOS, IL24, IL7R, MS4A2, PTGDR2, SUSD2, TCF7, TNFRSF25, VPREB3*. MAS-1 genes ($n = 22$): *ADAM17, ADM, ANG, C5AR1, CAMP, CD36, DEFA3, DEFA4, DEFB1, HAVCR2, HIST1H2BC, HIST1H2BD, HIST1H2BF, HIST1H2BG, HIST1H2BK, HIST2H2BE, HMGB2, MYD88, RNASE3, TBK1, TLR2, TNFSF8*.

To generate the $z$-scores, the normalized expression of each gene is $z$-transformed (mean centered then divided by standard deviation) across all samples and then averaged. Categorical score bins (high/low) of SAS-1 and MAS-1 were determined using the calculated median score values relative to each cohort.

**SAS-1/MAS-1 profiles.** The difference or change in proportions of the SAS-1/MAS-1 profiles was derived by combining the high/low expression of SAS-1 and MAS-1 scores based on median values in the entire dataset (which is dataset specific). High indicates expression of the score in the sample greater than the median expression of the score in the dataset, whereas low indicates expression of the score in the sample less than or equal to the median expression of the score in the dataset. The profiles detailed statistical methods per figure panel (Supplementary Information Sections 11.3 and 11.4).

**IMM-AGE transcriptomic signature score.** A list of 57 genes (Supplementary Information section 8.1) reported by Alpert et al.[11] as immune-aging transcriptomic signature (IMM-AGE) was used to derive this signature. The genes significantly and consistently correlated with both age and cell-based IMM-AGE score that predicted all-cause mortality in the FHS offspring cohort[11]. Note: the directionality of association of IMM-AGE (transcriptomic-based) with mortality reported by us in Fig. 2d (higher IMM-AGE score associated with lower mortality) is opposite from the association of IMM-AGE (cell-based) with mortality reported by Alpert et al.[11], as all 57 genes used in IMM-AGE (transcriptomic-based; reported by us in this study) are inversely correlated with the IMM-AGE (cell-based) score they derived. The IMM-AGE transcriptomic signature score was examined in different datasets to assess its association with survival. To generate the $z$-score, the $\log_2$ normalized expression of each gene is $z$-transformed (mean centered then divided by standard deviation) across all samples and then averaged.

**Publicly available expression datasets.** Details of the publicly available datasets are provided in Supplementary Information Section 8.3 and a summary of datasets studied is presented in Supplementary Data 13a.

**Statistics and reproducibility.** This study examines the metrics of immunologic resilience in a wide range of contexts, including acute/chronic infections, autoimmunity, aging, cancers, and vaccines. The broad principles used for the statistical approach are described in Supplementary Information Section 2.4. This section provides general information on the study design and how statistical analyses were conducted and are detailed in the statistics per panel section in the Supplementary information. In addition, each figure is linked with a source document for reproducibility. Furthermore, given the wide range of cohorts and conditions IHGs were examined under, we believe these results to be highly reproducible. Because secondary analyses were conducted, a priori sample size calculations were not conducted. No data were excluded from the analyses following cohort-specific inclusion/exclusion criteria, unless specifically stated in the detailed statistical methods in Supplementary Information Sections 11.2, 11.3, and 11.4. This was not an interventional study; therefore, no blinding or randomization was used.

### Data analysis
All analysis was conducted using R (https://cran.r-project.org/). Reported $P$ values are 2-sided and set at the 0.05 significance level. The models and $P$ values were not adjusted for multiple comparisons in the prespecified subgroup analyses, unless otherwise noted. All cutoffs and statistical tests were determined pre hoc. Logistic regression analyses were used to evaluate entry/baseline IR status and future HIV seroconversions; results are reported as odds ratios (ORs) with 95% confidence intervals (CIs). Kaplan-Meier plots were generated to depict rates of development of CSCC in HIV− RTRs and the rate of disease progression to AIDS (1993 CDC criteria) in HIV+ participants. The log-rank test was used to evaluate for overall significance. When comparing categorical data, the $\chi^2$ test was used when sample sizes were large (defined as when ≥80% of the values in the contingency table were ≥5 or none of the values in the contingency table was 0 or n's in the contingency table make Fisher's exact test computationally infeasible). In cases where the $\chi^2$ test was not applicable, the Fisher's exact test was used. For continuous variables, Welch's $t$-test, ANOVA, linear regression, Wilcoxon rank-sum test, Kruskal-Wallis test, and Pearson or Spearman correlation coefficient analyses were used where appropriate. Details of Pearson vs. Spearman correlation coefficient are provided in Supplementary Information Section 11.1. Follow-up times and analyses were prespecified.

### Plots
Boxplots (center line, median; box, the interquartile range (IQR); whiskers, rest of the data distribution ($\pm 1.5 \times$ IQR); and points, outliers greater than $\pm 1.5 \times$ IQR) were used to represent the median [IQR] of the indicated variables. Line plots with error bars were used to represent the mean $\pm$ standard error of the mean [SEM] of the indicated variable. Line plots with bands were used to represent either the odds with 95% confidence bands or mean $\pm$ [SEM] of the indicated variable. Line plots were used to represent proportions of indicated variables. Kaplan-Meier plots were used to represent proportion survived over time since score calculation (baseline) by indicated groups. Heatmaps were used to represent correlations of gene signature scores and continuous age. Stacked barplots or barplots were used to represent proportions or correlation coefficients of indicated variables. Forest plots were used to plot OR or HR [either unadjusted or adjusted] as

dots and 95% CI as error bars. Pie charts were used to represent proportions of indicated variables.

## Survival analysis

In the COVID-19 cohort, a Cox proportional hazards model, adjusted for sex and age as a continuous variable, was used to determine whether the gene scores associated with 90-day survival. In the FHS offspring cohort, a Cox proportional hazards model, adjusted for sex and age as a continuous variable, was used to determine whether the gene scores associated with survival. An FDR was used to correct the $P$ values from the Cox proportional hazards models for multiple comparisons and FDR < 0.05 was used to determine whether gene scores significantly associated with hazard of mortality. Kaplan-Meier survival plots of the FHS offspring cohort are accompanied by $P$ values determined by log-rank test.

## Predictors

Grades of antigenic stimulation and IR metrics were used as predictors. For determining the association between level of antigenic stimulation and IHG status in HIV− persons, proxies were used to grade this level and quantify host antigenic burden accumulated: (1) age was considered as a proxy for repetitive, low-grade antigenic experiences accrued during natural aging; (2) a BAS based on behavioral risk factors (condom use, number of clients, number of condoms used per client) and a total STI score based on direct [syphilis (rapid plasma reagin test) and gonorrhea] and indirect (vaginal discharge, abdominal pain, genital ulcer, dysuria, and vulvar itch) indicators of STI were used as proxies in HIV− FSWs for whom this information was available; and (3) *S. haematobium* egg count in the urine was a proxy in children with this infection. For HIV+ persons, plasma HIV VL was a proxy for level of HIV-associated antigenic stimulation.

## Key predictor-outcome dyads

(1) IHGs-age (proxy for accumulated antigenic experience); (2) IHGs at first episode of CSCC in RTRs with the outcome of second episode of CSCC; (3) IHG at baseline in HIV+ persons and AIDS development; (4) IHG at baseline in FSWs with future HIV seroconversion; (5) IHG at baseline with COVID-19 outcomes; (6) gene signatures and survival rates in either persons with or without acute COVID-19; (7) gene signatures and sepsis survival; and (8) gene signatures and influenza outcomes.

## Statistics for transcriptomic signature scores and SAS-1/MAS-1 profiles

For cross-sectional comparisons, Welch's $t$-test was used to evaluate the difference in scores between 2 groups. ANOVA-based linear regression model was used to evaluate the overall differences between 3 or more groups. For comparison of groups with multiple samples from the same individuals, we used a linear generalized estimating equation (GEE) model based on the normal distribution with an exchangeable correlation structure unless otherwise stated. Pearson's correlation coefficient was used to evaluate the correlation between transcriptomic signature scores. For the association of gene scores with outcomes, linear regression (linear model) was used to test them, instead of nonparametric tests as highlighted below in the panel-by-panel detailed statistical methods for each of the figures. For median-based SAS-1/MAS-1 profile distribution analysis, nonparametric tests (Fisher's exact test, $\chi^2$ test) were used as appropriate. For comparison of groups with multiple samples from the same individuals, we used a linear GEE model based on the normal distribution with an exchangeable correlation structure unless otherwise stated.

## Meta-analysis of gene expression datasets

For meta-analyses (e.g., data presented in Fig. 7d), the samples from 2 or more datasets combined for analyses were from the same source (tissue or cell type) and assayed on the same platform. All datasets were filtered for common probes. Then, an expression matrix of the probes and samples was created and concurrently normalized as stated in Supplementary Information Section 9.2 before scores for signatures were computed. Example: if dataset #1 provided $\log_2$ values and dataset #2 was quantile normalized, dataset #1 would be un-log transformed by exponentiation with the base 2 before combining with dataset #2 for concurrent normalization and computation of scores. The phenotype groups for plots were determined from the phenotype data deposited in the GEO or ArrayExpress along with the dataset. The phenotype groups were classified based on the hypothesis evaluated.

## Quality control of the dataset and interpretation

The transcriptomic signature score is a relative term within a dataset, and it is challenging to compare the score across different datasets. For the meta-analyses, we used a series of criteria as described in Supplementary Information Section 9.4.1 to make comparisons more equitable between the datasets. Different RNA microarray or RNA-seq platforms have differences in the availability of gene probes corresponding to the genes in a given transcriptomic signature score. Thus, we indicated the gene count range in each dataset (Supplementary Data 13b). As the overall median (IQR) percentage of available genes is high, 90.9% (85.7–100%), the chance of those unavailable genes impacting our interpretation are low. In addition, we stress that transcriptomic signature scores were defined in relative terms and caution is needed for cross-dataset comparisons.

## Reporting summary

Further information on research design is available in the Nature Portfolio Reporting Summary linked to this article.

# Data availability

The phenotype and processed gene expression data generated are available in Figshare database for COVID-19 cohort (https://figshare.com/projects/Ahuja_Lab_COVID-19_dataset/158732) and HIV dataset (https://figshare.com/projects/Ahuja_Lab_HIV_dataset/158681). Individual level raw data files of the VA COVID-19 cohort cannot be shared publicly due to data protection and confidentiality requirements. South Texas Veterans Health Care System (STVHCS) at San Antonio, Texas, is the data holder for the COVID-19 data used in this study. Data can be made available to approved researchers for analysis after securing relevant permissions via review by the IRB for use of the data collected under this protocol. Inquiries regarding data availability should be directed to the corresponding author. Accession links to all data generated or analyzed during this study are included in Supplementary Data 13a. Source data are provided with this paper. All other patient/individual-level raw data (including HIV dataset from EIC) underlying this article cannot be shared publicly due to data protection and confidentiality requirements. The data holders and contacts for inquiries to data access are in listed in Supplementary Data 13c for the following cohorts: SardiNIA, HIV− UCSD, schistosomiasis in Kenyan children, RTRs, FSWs from the Kenya MOCS cohort, EIC, PIC, SIV− and SIV+ sooty mangabeys, SIV− rhesus macaques, and CC-RIX mice. Database and sources of publicly available gene expression datasets analyzed in this study are provided Supplementary Data 13a for the following cohorts: SLE (GSE49454), HIV/TB Meta analysis with Finnish DILGOM cohort (GSE29429, GSE19439, GSE19442, GSE19444, E-TABM-1036), San Antonio Family Heart Study (E-TABM-305), FHS offspring cohort (phs000007.v30.p11, phs000363.v17.p11), Sepsis meta-analysis (E-MATB-4421, E-MATB-4451, E-MATB-5273, E-MATB-5274), sepsis cohort (E-MATB-1548), Vitality 90+ study cohort (GSE65218, GSE65219), HIV-1 Infection (GSE16363), influenza and other acute respiratory viral infections (GSE68310), influenza A H1N1 and H3N2 virus infection (GSE52428), symptomatic respiratory viral infection (GSE17156), burn injury (GSE182616), sepsis (GSE185263), severe

influenza (GSE111368), and influenza-infected pre-CC lines (GSE30506). Aggregate data presented for these cohorts in the current study are provided in the source data file. Immunophenotyping data from the SardiNiA cohort used in Fig. 10 are derived and sourced from Orrù et al. (doi: 10.1016/j.cell.2013.08.041) and were shared with us by the co-authors. Data from RTRs are derived and sourced from Bottomley et al. (doi: 10.1681/ASN.2015030250) and were shared with us by the co-author. The sources of the data for the literature survey (Fig. 5c) are summarized in Supplementary Table 2. Source data are provided with this paper.

## Code availability

GEO2R R script sourced from NCBI GEO [source: https://www.ncbi.nlm.nih.gov/geo/info/geo2r.html] was used for download and analyses of GEO datasets, and a script from vignette of ArrayExpress R package was used for download and analyses of ArrayExpress datasets. The scripts are available from the corresponding author on request.

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

## Acknowledgements

The two main sources of funding for the data presented herein are those awarded to S.K.A. and J.F.O. S.K.A. was supported by grants from the Veterans Affairs (VA) [VA Research Center for AIDS and HIV Infection, VA Center for Personalized Medicine (IP1 CX000875-01A1), and a VA MERIT award]; the National Institutes of Health (NIH) MERIT award (R37AI046326); the Doris Duke Distinguished Clinical Scientist Award; the Elizabeth Glaser Pediatric AIDS Foundation; the Burroughs Wellcome Clinical Scientist Award in Translational Research; and the Senior Scholar Award from the Max and Minnie Tomerlin Voelcker Fund. The work was also supported, in part, by an award jointly funded by NIAID/ NIH (#AAI20042-001) and the Veterans Affairs (COVID19-8100-01) awarded to S.K.A. and M.I.R. A portion of the material presented is based on research sponsored by the U.S. Air Force under agreement number FA8650-17-2-6816 (United States Air Force 59th Medical Wing Intramural Award to J.F.O.). This study was also supported by the Infectious Disease Clinical Research Program (IDCRP), a Department of Defense program executed by the Uniformed Services University of the Health Sciences through a cooperative agreement with The Henry M. Jackson Foundation for the Advancement of Military Medicine, Inc. (H.J.F). The IDCRP has been funded in whole, or in part, with federal funds from the National Institute of Allergy and Infectious Diseases, NIH, under Inter-Agency Agreement (Y1-AI-5072). The SardiNIA study was supported in part by the Intramural Research Program of the NIH, National Institute on Aging, with contracts N01-AG-1-2109 and HHSN271201100005C; by Italian grants FISM 2011/R/13, FaReBio2011, Funds MIUR/CNR for Rare Diseases and Molecular Screening, CNR/DSB flagship INTEROMICS, PNR/CNR Aging Program 2012-2014; European Union's Horizon 2020 Research and Innovation Programme under grant agreement 633964; Giovani Ricercatori 2007 (D.lgs 502/92); and Legge Regionale 30 giugno 2011 n.12, articolo 3, comma 3 (to F.C.). The Kenya Majengo Observational (female sex worker) Cohort Study was supported by grants from the NIH (R01 AI56980), the Canadian Institutes of Health Research (HOP-43135), the Bill and Melinda Gates Foundation (39673), and the CIHR through the Grand Challenges in Global Health Initiative (to F.A.P.). LRM was supported by a CIHR Biomedical/Clinical HIV/AIDS Research Fellowship and the International Infectious Diseases and Global Health Training Program. The HIV- UCSD cohort was supported by National Institute of Mental Health (NIMH) P30 grant (PI: R. Heaton, MH62512), MARC from National Institute on Drug Abuse P50 grant (PI: I. Grant DA26306), and ProM from NIMH R01 grant (PI: S. Woods, MH73419). S.L. was supported by K24 MH097673 from the National Institute of Mental Health. The renal transplant recipient cohort and MJB were supported by grants from the Wellcome Trust (Clinical Training Fellowship) and Oxford Hospitals Research Services Committee. M.J.B. acknowledges the support of the UK National Institute for Health Research through the Local Clinical Research Network. The HIV- Kenyan Schistosoma haematobium children cohort was supported by NIH grant AI064687 (C.L.K.). The primary HIV infection cohort and D.M.S., S.J.L., and D.D.R. were supported by NIH grants AI43638, AI074621, AI106039, and MH100974; Inter-Agency Agreement Y1-AI-5072; and the California HIV Research Program RN07-SD-702. The Sooty mangabey cohort and GS were supported by NIH grant A1 R3766998. The Chinese rhesus macaque study was supported by the National Basic Research Program of China (2012CBA01305), the Knowledge Innovation Program of CAS (KSCX2-EW-R-13), the National Natural Science Foundation of China (81172876, 81273251, U1202228), and the Key Scientific and Technological Program of China (2013ZX10001-002, 2012ZX10001-007). The CC mice study and J.M.L. were supported by NIH grants AI100625, AI096968, and AI087657. AMS was supported by the NIH T32DE014318 COSTAR institutional research training gigant. K.R.C. was supported by NIH grant T32GM113896/

STXMSTP. This work was also supported by NIH grant 1UL1 TR002645 (Clinical and Translational Science Award to RAC). G.C.L. was supported by the NIH K23-AG066933. We thank participants of the cohorts, other members of the Ahuja lab that contributed to the study, Dr. Kimberly Summers for help with study approvals, and Donna Thordsen for critical reading of the manuscript.

Framingham Heart Study dbGaP Acknowledgement Statement: The Framingham Heart Study is conducted and supported by the National Heart, Lung, and Blood Institute (NHLBI) in collaboration with Boston University (Contract No. N01-HC-25195 and HHSN268201500001I). This manuscript was not prepared in collaboration with investigators of the Framingham Heart Study and does not necessarily reflect the opinions or views of the Framingham Heart Study, Boston University, or NHLBI. Additional funding for SABRe was provided by Division of Intramural Research, NHLBI, and Center for Population Studies, NHLBI.

Disclaimer is noted in the Supplementary Information.

## Author contributions

S.K.A. conceived the idea; designed, supervised, and coordinated the study; interpreted the data; and wrote the manuscript. S.K.A. and J.F.O. obtained funding for the study. W.H., J.F.O., and R.A.C. provided key input since the inception of the study in 2015. W.H., M.S.M., G.C.L., N.H., J.F.O., J.A.M., R.R.-B., and R.A.C. provided perceptive conceptual contributions and scientific feedback. Access to cohorts and corresponding data was provided by: M.S., E.F., V.O., and F.C. (SardiNIA cohort); L.R.M., N.G., J.K., T.B.B., F.A.P., and K.R.F. (female sex worker cohort); J.F.O. and B.K.A. (early HIV infection cohort); S.J.L., D.D.R., S.G., and D.M.S. (acute HIV cohort); M.J.B., P.N.H., and K.J.W. (renal transplant cohort); C.L.K. (schistosomiasis cohort); S.L. (HIV-seronegative cohort from University of San Diego); G.S. (sooty mangabey monkeys cohort); H.-Y.Z. and Y.-T.Z. (Chinese rhesus macaques cohort); and J.M.L., M.T.F., and M.T.H. (CC mice cohort). M.I.R., S.S.-R., A.A., J.M.H., J.A.P., E.A.W., and J.A.C.Z. and additional members of the South Texas Veterans Health Care System COVID-19 Team provided access to, contributed to, and/or supervised data collection of the acute COVID-19 cohort. M.I.R., G.C.L., and S.K.A. directed accrual of the COVID-19 cohort. S.S.-R. and H.T. contributed to the assembly of the convalescent COVID-19 cohort. L.P., F.J., A.P.B., A.M.S., A.C., K.R.C., C.A.W., and L.A.W. provided experimental support. V.O. and E.F. performed immunophenotyping in the SardiNIA cohort. W.H., M.S.M., N.H., J.A.M., M.S., and G.C.L. performed bioinformatics or biostatistical analyses. A.P.B., A.M.S., M.S.M., and W.H. drafted the figures. Online supplementary information was prepared by W.H., J.A.M., M.S.M., N.H., G.C.L., A.P.B., and S.K.A. M.S.M., G.C.L., L.R.M., J.A.M., M.S., N.H., E.F., A.M.S., M.I.R., M.J.B., V.O., F.J., A.A.G., A.G.M., S.S.A.K., R.R.-B., R.A.C., and J.F.O. contributed to data interpretation and provided editorial suggestions.

## Competing interests

The authors declare no competing interests.

## Additional information

Sunil K. Ahuja [1,2,3,4] ✉, Muthu Saravanan Manoharan[1,4,30], Grace C. Lee[1,3,5,6,30], Lyle R. McKinnon[7,8,30], Justin A. Meunier[1,9,30], Maristella Steri [10,30], Nathan Harper[1,9,30], Edoardo Fiorillo[10,30], Alisha M. Smith[1,2,9,30], Marcos I. Restrepo[1,3,4,30], Anne P. Branum [1,9,30], Matthew J. Bottomley [11,12,30], Valeria Orrù[10], Fabio Jimenez[1,9], Andrew Carrillo[1,9], Lavanya Pandranki [1,4], Caitlyn A. Winter[1,4,9,13], Lauryn A. Winter[1,4,9,13], Alvaro A. Gaitan[1,9], Alvaro G. Moreira[1,13], Elizabeth A. Walter[1,3,4], Guido Silvestri [14], Christopher L. King [15], Yong-Tang Zheng [16,17], Hong-Yi Zheng[16,17], Joshua Kimani[8], T. Blake Ball[8], Francis A. Plummer[8], Keith R. Fowke[8], Paul N. Harden[12], Kathryn J. Wood[11], Martin T. Ferris[18], Jennifer M. Lund [19,20], Mark T. Heise[18], Nigel Garrett [7], Kristen R. Canady[1], Salim S. Abdool Karim[7,21], Susan J. Little [22,23], Sara Gianella[22,23], Davey M. Smith[22,23,24], Scott Letendre[22], Douglas D. Richman[23], Francesco Cucca [10,25], Hanh Trinh [3], Sandra Sanchez-Reilly[3,4], Joan M. Hecht[3,9], Jose A. Cadena Zuluaga[3,4], Antonio Anzueto[3,4], Jacqueline A. Pugh[1,3,4], South Texas Veterans Health Care System COVID-19 team*, Brian K. Agan[26,27], Robert Root-Bernstein [28], Robert A. Clark [1,2,3,4,9,30], Jason F. Okulicz[26,29,30] & Weijing He[1,9,30]

[1]VA Center for Personalized Medicine, South Texas Veterans Health Care System, San Antonio, TX 78229, USA. [2]Department of Microbiology, Immunology & Molecular Genetics, University of Texas Health Science Center at San Antonio, San Antonio, TX 78229, USA. [3]South Texas Veterans Health Care System, San Antonio, TX 78229, USA. [4]Department of Medicine, University of Texas Health Science Center at San Antonio, San Antonio, TX 78229, USA. [5]Pharmacotherapy Education and Research Center, School of Medicine, University of Texas Health Science Center at San Antonio, San Antonio, TX 78229, USA. [6]College of

Pharmacy, The University of Texas at Austin, Austin, TX 78712, USA. [7]Centre for the AIDS Programme of Research in South Africa (CAPRISA), University of KwaZulu-Natal, Durban 4001, South Africa. [8]Department of Medical Microbiology and Infectious Diseases, University of Manitoba, Winnipeg, MB R3T 2N2, Canada. [9]The Foundation for Advancing Veterans' Health Research, San Antonio, TX 78229, USA. [10]Istituto di Ricerca Genetica e Biomedica (IRGB), CNR, Monserrato 09042, Italy. [11]Transplantation Research Immunology Group, Nuffield Department of Surgical Sciences, University of Oxford, Oxford OX1 2JD, UK. [12]Oxford Kidney Unit, Churchill Hospital, Oxford University Hospitals NHS Foundation Trust, Oxford OX3 7LE, UK. [13]Department of Pediatrics, University of Texas Health Science Center at San Antonio, San Antonio, TX 78229, USA. [14]Department of Pathology, Emory University School of Medicine & Emory National Primate Research Center, Atlanta, GA 30322, USA. [15]Center for Global Health and Diseases, Case Western Reserve University, Cleveland, OH 44106, USA. [16]Key Laboratory of Animal Models and Human Disease Mechanisms of the Chinese Academy of Sciences, KIZ-CUHK Joint Laboratory of Bioresources and Molecular Research in Common Diseases, Kunming Institute of Zoology, Chinese Academy of Sciences, Kunming, Yunnan 650223, China. [17]National Resource Center for Non-Human Primates, Center for Biosafety Mega-Science, Kunming Institute of Zoology, Chinese Academy of Sciences, Kunming, Yunnan 650107, China. [18]Department of Genetics, University of North Carolina, Chapel Hill, NC 27599, USA. [19]Vaccine and Infectious Disease Division, Fred Hutchinson Cancer Research Center, Seattle, WA 98109, USA. [20]Department of Global Health, University of Washington, Seattle, WA 98195, USA. [21]Department of Epidemiology, Mailman School of Public Health, Columbia University, New York, NY 10032, USA. [22]Department of Medicine, University of California, La Jolla, CA 92093, USA. [23]San Diego Center for AIDS Research, University of California San Diego, La Jolla, CA 92093, USA. [24]Veterans Affairs San Diego Healthcare System, San Diego, CA 92161, USA. [25]Dipartimento di Scienze Biomediche, Università di Sassari, Sassari 07100, Italy. [26]Infectious Disease Clinical Research Program, Department of Preventive Medicine and Biostatistics, Uniformed Services University of the Health Sciences, Bethesda, MD 20814, USA. [27]The Henry M. Jackson Foundation for the Advancement of Military Medicine, Bethesda, MD 20817, USA. [28]Department of Physiology, Michigan State University, East Lansing, MI 48824, USA. [29]Department of Medicine, Infectious Diseases Service, Brooke Army Medical Center, San Antonio, TX 78234, USA. [30]These authors contributed equally: Muthu Saravanan Manoharan, Grace C. Lee, Lyle R. McKinnon, Justin A. Meunier, Maristella Steri, Nathan Harper, Edoardo Fiorillo, Alisha M. Smith, Marcos I. Restrepo, Anne P. Branum, Matthew J. Bottomley, Robert A. Clark, Jason F. Okulicz, Weijing He. *A list of authors and their affiliations appears at the end of the paper. ✉e-mail: ahujas@uthscsa.edu

## South Texas Veterans Health Care System COVID-19 team

Mohamed I. Abdalla[3], Sandra G. Adams[3], Yemi Adebayo[3], Joseph Agnew[3], Sunil K. Ahuja ®[1,2,3,4]✉, Saleem Ali[3], Gregory Anstead[3,4], Antonio Anzueto[3,4], Marichu Balmes[3], Jennifer Barker[3], Deborah Baruch-Bienen[3], Velma Bible[3], Angela Birdwell[3], Stacy Braddy[3], Stephen Bradford[3], Heather Briggs[3], Jose A. Cadena Zuluaga[3,4], Judith M. Corral[3], Jennifer J. Dacus[3], Patrick J. Danaher[3], Scott A. DePaul[3], Jill Dickerson[3], Jollynn Doanne[3], Aamir Ehsan[3], Samantha Elbel[3], Miguel Escalante[3], Corina Escamilla[3], Valerie Escamilla[3], Robert Farrar[3], David Feldman[3], Debra Flores[3], Julianne Flynn[3], Delvina Ford[3], Joanna D. Foy[3], Megan Freeman[3], Samantha Galley[3], Jessica Garcia[3], Maritza Garza[3], Sherraine Gilman[3], Melanie Goel[3], Jennifer Gomez[3], Varun K. Goyal[3], Sally Grassmuck[3], Susan Grigsby[3], Joshua Hanson[3], Brande Harris[3], Audrey Haywood[3], Joan M. Hecht[3,9], Cecilia Hinojosa[3], Tony T. Ho[3], Teri Hopkins[3], Lynn L. Horvath[3], Aneela N. Hussain[3], Ali Jabur[3], Pamela Jewell[3], Thomas B. Johnson[3], Austin C. Lawler[3], Grace C. Lee[1,3,5,6,30], Monica Lee[3], Chadwick S. Lester[3], Stephanie M. Levine[3], Haidee V. Lewis[3], Angel Louder[3], Charmaine Mainor[3], Rachel Maldonado[3], Celida Martinez[3], Yvette Martinez[3], Diego Maselli[3], Chloe Mata[3], Neil McElligott[3], Laura Medlin[3], Myra Mireles[3], Joanna Moreno[3], Kathleen Morneau[3], Julie Muetz[3], Samuel B. Munro[3], Charlotte Murray[3], Anoop Nambiar[3], Daniel Nassery[3], Robert Nathanson[3], Kimberly Oakman[3], Jane O'Rorke[3], Cheryl Padgett[3], Sergi Pascual-Guardia[3], Marisa Patterson[3], Graciela L. Perez[3], Rogelio Perez[3], Rogelio Perez III[3], Robert E. Phillips[3], Patrick B. Polk[3], Michael A. Pomager[3], Kristy J. Preston[3], Kevin C. Proud[3], Jacqueline A. Pugh[1,3,4], Michelle Rangel[3], Marcos I. Restrepo[1,3,4,30], Temple A. Ratcliffe[3], Renee L. Reichelderfer[3], Evan M. Renz[3], Jeanette Ross[3], Teresa Rudd[3], Maria E. Sanchez[3], Sandra Sanchez-Reilly[3,4], Tammy Sanders[3], Kevin C. Schindler[3], David Schmit[3], Raj T. Sehgal[3], Claudio Solorzano[3], Nilam Soni[3], Win S. Tam[3], Edward J. Tovar[3], Sadie A. Trammell Velasquez[3], Hanh Trinh ®[3], Anna R. Tyler[3], Anjuli Vasquez[3], Maria C. Veloso[3], Steven G. Venticinque[3], Jorge A. Villalpando[3], Melissa Villanueva[3], Lauren Villegas[3], Megan Walker[3], Andrew Wallace[3], Maria Wallace[3], Elizabeth A. Walter[1,3,4], Emily Wang[3], Stephanie Wickizer[3], Andreia Williamson[3], Andrea Yunes[3] & Katharine H. Zentner[3]

