## [Peer Review File · Nature Communications]

Immune resilience despite inflammatory stress promotes longevity and favorable health outcomes including resistance to infectionREVIEWER COMMENTS

Reviewer #1 (Remarks to the Author):

Lee et al. describe a remarkable survey in which they seek to align their measures of immune resilience with longevity and susceptibility to AIDS and severe COVID-19 essentially by establishing 4 categories of peripheral blood CD4:CD8 ratios in large international databases of over 40,000 people as well as 300 non-human primates and a number of genetically-heterogeneous mice. This is an enormous tour de force of diverse human populations with much to recommend it, but it is presented in a manner so densely-written, complicated, acronym-loaded and annotated that it is extremely difficult to read and to follow. The authors propose that no matter which populations are examined, be they HIV-positive or negative, young or old, organ transplant recipients, healthy or autoimmune-diseased, primate or not, the absolute numbers and ratios of CD4 T cells and CD8 T cells in the blood act as robust biomarkers of states that they label “antigenic stimulation-induced age-independent immunosenescence (AIIS)” subdivided into partial or complete, as opposed to age-dependent immunosenescence (ADIS). The 4 categories of CD4:8 ratio, labelled “Immune Health Grades” (IHGs) I thru IV are based on clinical AIDS data cutoffs for low CD4+ T cell numbers. On this foundation, Lee et al build an enormous edifice drawing some very interesting and provocative conclusions and attempting to infer causality rather than merely accepting their easy-to-use (but complicatedly-described) CD4:8 ratio indices as biomarkers. This may or may not be the case, but in any event the simple biomarkers deserve consideration by the community, as they are easily available for investigation in many other datasets not explicitly covered in the present study.

Notwithstanding the above considerations, I do feel that this paper is based on several questionable statements and assumptions that clearly mould the authors’ paradigm of “immunosenescence” and probably strongly reflect the HIV background of the senior author. The Abstract plunges straight in to distinguishing age-independent antigen-dependent immunosenescence and antigen stimulation-dependent immunosenescence as if these two could be clearly separated. I do not see how this can be achieved, and would ask the authors to simplify their manner of writing to clarify their meaning without using catch-all designations like “immunosenescence” which are by no means standardised. These mean different things to different people. Specifically:

1) What is the evidence that ADIS exists, as opposed to AIIS? All individuals studied here will have been challenged by many antigens. Only a comparison with gnotobiotic mice could truly be used to define ADIS, and maybe not even then due to autoantigens. I believe that it would be preferable to avoid these confusing invented acronyms and stick only to a description of what is actually measured. For example, the authors refer here to a paper in press as follows: “Supporting this inference, AIIS corresponds to age-independent loss of immunologic resilience, a trait that we recently showed associated with severe COVID-19 and shorter lifespan”. This paper is now out, but I cannot see that it refers to “AIIS” anywhere. So why in the present paper?

2) Coming from the HIV field, the beginning of the Intro makes the assumption that HIV infection models premature immune aging, but this is not as commonly accepted in the field as the authors seem to believe. Thus, it is stated that “both SLE and HIV infection are viewed as model systems of “premature aging” and two references are cited in support of this claim, neither of which represents anything like a consensus. They make statements focusing on changes that may be attributable to low-grade, HIV-associated antigenic

stimulation (although they may also be caused by the therapy and other effects of the virus) which “degrades immune status” – but in what way does Ag stimulation “degrade” immune status? It is the basis of adaptive immunity, not degradation. They then conclude that “In contrast, deterioration in immune status attributable to aging per se develops over time.” But what is “aging per se”? This is also a matter of intense debate and is not accepted as a condition that can be viewed as meaningful in this context without a definition of what is meant by it. And I have to say that I am puzzled by the subsequent statement that this state is one “representing physiologic involution of the immune system attributable to aging”. What do the authors mean by “physiologic involution”? It is also not the case that the presence of “CD4+ and CD8+ senescent T cells (CD28- CD4+ or CD8+ T cells), and T-cell receptor excision circles, and telomerase reverse transcriptase” do represent “4 conventional markers of immunosenescence”. T cells are not necessarily senescent because they are CD28-negative, TCR excision circles provide an indicator of antigen exposure and HERT marks activation.

3) In the Results, illustrating the assumption that the same mechanism lies behind the observed phenomena of CD4:8 ratios, the authors state that “CD4+ lymphopenia is a characteristic of aging, severe COVID-19, SLE, and HIV. While this is certainly true for HIV and may be for SLE, for different reasons, is it really true for aging?? For example, <https://pubmed.ncbi.nlm.nih.gov/27547234/> indicates very little CD4 lymphopenia with age. Disease states are completely different. Refs 31, 32 do not address this specifically. The authors then also state that “a higher CD4:CD8 ratio is viewed as a biomarker of immunocompetence, ref. 34” – yes, Ref. 34 is for HIV, but this is not necessarily also the case in aging, eg. see <https://pubmed.ncbi.nlm.nih.gov/27927759/>

4) The concept of CD4:8 equilibrium is introduced but what does this mean? This is not clear, also not from Note 2. The variable numbers and ratios of CD4:CD8 cells say nothing about an equilibrium.

5) The association of age with a disproportionately greater loss of CD8+ vs. CD4+ T-cells, representing IHG feature 3 may also be caused by CD8 clonal attrition due to chronic stimulation by persistent pathogens.

I would suggest that your very interesting survey is presented in a far more neutral fashion and made less difficult for readers to wade through by reducing the use of newly-invented acronyms. In the light of the above comments, I would suggest a more neutral descriptive and less interpretative title such as “Usefulness of modified CD4:8 ratio assessments as biomarkers of disease and longevity”

Reviewer #2 (Remarks to the Author):

This manuscript presents an analysis of clinical and immunologic parameters in varied cohorts in order to inform on the relationship between health status, antigenic stimulation-induced age-independent immunosenescence (AIIS) and age dependent immunosenescence (ADIS).

The authors developed metrics to differentiate among AIIS-free, partial-AIIS, and complete-AIIS status, as well as between AIIS and ADIS. These metrics are based on four immune health grades (IHGs) derived by co-indexing the CD4+T-cell count and CD4:CD8 T-cell ratio. When available, they also related IHGs to cell phenotype data and gene expression

signatures that associate with survival/mortality (SAS-1, MAS-1). They analyzed a large spectrum of cohorts of humans (n=43,286 including healthy children to centenarians, people living with HIV, lupus patients, and COVID-19 patients), non-human primates (n=279), and mice (n=332).

The authors report that AIIS and ADIS do not represent a single and same program, but are associated with distinct immunologic and clinical traits, that should not be confused with one another. Their findings indicate that the combination and interaction of ADIS, AIIS and sex influences lifespan.

This work focuses on interesting concepts like AIIS and ADIS, and important health contexts. The findings have a potential relevance with regards to the implementation of AIIS status monitoring using simple metrics like IHGs for prevention, prognostic, and clinical care.

Despite these interesting aspects and the size of the study, I see two important drawbacks.

1- The conclusion that combinatorial effects of AIIS and ADIS together with gender contribute to lifespan differences and longevity is sound but not particularly original, and already quite established in the field. It refers to the notions of biological and chronological age and their consequences on health. Moreover claiming that the authors' findings provide a refined mechanistic framework to understand the heterogeneity in aging phenotypes including lifespan is an overstatement. The study is not mechanistic.

2- The content of the manuscript is difficult to grasp, in part due the many cohorts presented in the study, and the way it is written. The original findings and conclusions of the work should be better highlighted, for instance using subheadings and adapted figures.

Overview

We very much appreciate the incisive comments of both reviewers. Both reviewers acknowledged the breadth, scope, and broad implications of our study. Both reviewers acknowledged that this is a very large study; reviewer 1 termed our study a “*remarkable survey*” and an “*enormous tour de force of diverse human populations with much to recommend it,*” and stated that the study builds “*an enormous edifice drawing some very interesting and provocative conclusions.*” Reviewer 1 also highlighted the broad translational utility of our work: “*in any event the simple biomarkers deserve consideration by the community, as they are easily available for investigation in many other datasets not explicitly covered in the present study.*” Reviewer 2 stated “*The findings have a potential relevance with regards to the implementation of AIIS status monitoring using simple metrics like IHGs for prevention, prognostic, and clinical care.*”

Reviewer 1 also acknowledged that we are attempting to address critical questions that are difficult to study in humans related to the identification of deficits in immune status attributable to host x environment (antigenic stimulation) interactions at any age vs. processes directly attributable to aging. We are very appreciative that reviewer 1 challenged us with some provocative and astute questions that are at the heart of the debate regarding immune deficits during aging and the development of interventions to improve healthspan and lifespan.

However, both reviewers found the paper dense and difficult to read. We agree. **While the overall concepts and fundamental observations of our study remain unchanged**, we have made extensive changes to the flow of information and revised terminologies used to improve readability. Below, we describe these changes and address the shared broad concerns of the reviewers. We then provide a detailed point-by-point response to each comment.

Key revisions and new datasets to address the reviewers’ comments are highlighted in yellow.

(1) Concepts:

Our overall hypothesis is that, *independent of age*, a novel antigen-activated immunologic program is associated with deficits in immune status that predict inferior immunity-dependent outcomes, including shorter lifespan, AIDS, and severe COVID-19 and influenza. We noted that females are more likely to resist induction of this program. Two metrics to monitor levels of immune status attributable to this program were developed: immune health grades (**IHG**s I to IV) and gene expression signatures that predict survival during acute COVID-19 and lifespan in the Framingham Heart Study participants (without COVID-19), *independent of age and sex*. These gene expression signatures were termed **SAS-1** (survival-associated signature-1) and **MAS-1** (mortality-associated signature-1).

We had framed our hypothesis in terms of age-independent immunosenescence (**AIIS**) and age-dependent immunosenescence (**ADIS**). Based on the abovementioned metrics, we assigned persons as having AIIS-free status vs. AIIS status, with AIIS-free status signifying the absence of AIIS. Through immunophenotyping of nearly 4,000 otherwise healthy individuals, we identified immunologic traits that distinguished those attributable to AIIS vs. ADIS. Through evaluation of large-scale epidemiologic studies, we noted that metrics signifying AIIS-free status were associated with superior immunity-dependent health outcomes, including a longevity advantage.

Reviewer concerns: concepts are described through unfamiliar, unclear, and confusing acronyms.

- We thank reviewer #1 for emphasizing that the acronyms AIIS and ADIS were prone to misinterpretation and confusion.
- Reviewer 1 also rightly alluded to the fact that having the term immunosenescence in both AIIS and ADIS is a source of confusion, especially since the definition of immunosenescence is not uniform and can vary depending on context (e.g., T-cell senescence vs. cellular senescence).

Key revision: Use of the acronyms AIIS and ADIS has been eliminated. However, the principles that prompted us to conceptualize AIIS as an age-independent process that is associated with lower immune status remains unchanged.

AIIS is the same as loss of immunologic resilience at any age. The term immunologic resilience is intuitively easier to understand and is in the public domain.

- As noted by reviewer 1, we recently introduced the term **immunologic resilience (IR)**, *defined as the capacity to preserve and/or rapidly restore immunocompetence and restrict or control inflammation during increased antigenic stimulation.*
- We showed that preservation of **optimal IR** during acute COVID-19 was associated with survival benefits irrespective of age.
- Our report, titled “Immunologic resilience and COVID-19 survival advantage,” was published in November 2021 in the *Journal of Allergy and Clinical Immunology (JACI)*¹.
- The above report was accompanied by an editorial that highlighted the use of IHGs as novel metrics of immune health: “Immune health grades: Finding resilience in the COVID-19 pandemic and beyond” by Marconi et al².
- Hence, the concept of IR is now in the public domain, cited in: *The Lancet Healthy Longevity* (“Biological ageing with HIV infection: evaluating the geroscience hypothesis”; March 2022)³, and *Nature Aging* (“Research and resource needs for understanding host immune responses to SARS-CoV-2 and COVID-19 vaccines during aging”; December 2021)⁴.

Parallels between IR status and AIIS status: reasons for framing our concepts from the perspective of IR rather than AIIS.

- AIIS indicates the *presence of inferior immunity* (lower immune status). AIIS-free indicates the *absence* of AIIS. In hindsight, we agree this terminology can be confusing.
- On the other hand, optimal IR refers to the *preservation* of superior immunity. Intuitively, it is simpler to understand optimal IR than AIIS-free status.
- Similarly, it is easier to understand loss of optimal IR than AIIS.
- Thus, in agreement with reviewer 1, based on our published work (*JACI*), we have framed our study from the perspective of an IR continuum (optimal, suboptimal, and nonoptimal) rather than AIIS-free vs. AIIS.
- Thus, optimal IR equates to AIIS-free status, whereas suboptimal or nonoptimal IR equates to partial or complete AIIS, respectively.
- *The metrics used for assignment of IR status and AIIS status are the same.*

- Based on our published work (*JACI*), IHG-I and gene expression signatures that are associated with survival/longevity were assigned as indicators of optimal IR.
- With new data, we have defined the precise IHGs associated with optimal, suboptimal, and nonoptimal IR (**new Figure 6, Fig. 7a**).
- Thus, indicators of optimal IR (previously AIIIS-free status) are associated with superior immunity-dependent health outcomes that include survival/longevity and resistance to severe COVID-19, AIDS, and symptomatic or severe influenza.

Acronyms/abbreviations: To assist the readers, the acronyms/abbreviations are now listed in **revised Figure 1**.

- 1) **IR:** immunologic resilience
- 2) **Immune health grade:** IHG, a metric to monitor level of IR
- 3) **IC:** immunocompetence
- 4) **IF:** inflammation
- 5) **FHS:** Framingham Heart Study
- 6) **SAS-1:** survival-associated signature-1.
 - SAS-1 expression served as a *transcriptomic proxy for immunocompetence (IC)*, as it comprised genes related to immunity (e.g., *CCR7, IL7R*) and higher expression levels of SAS-1 (SAS-1^{high}) were associated with **lower** mortality hazards during acute COVID-19 and during aging in participants in the Framingham Heart Study (FHS), after controlling for age and sex (**Fig. 2d**).
 - We are confident of our findings, as Mark Davis's group⁵ reported an IMM-AGE gene signature that is highly correlated with SAS-1 (**Supplementary Figure 11**). Akin to higher expression of SAS-1, higher expression of IMM-AGE also predicts survival in the FHS, independent of age and sex (**Fig. 2d**).
- 7) **MAS-1:** mortality-associated signature-1.

MAS-1 expression served as a *transcriptomic proxy for inflammation (IF)*, as it comprised genes related to immunity (e.g., *TLR2, MYD88*) and higher expression levels of MAS-1 (MAS-1^{high}) were associated with **higher** mortality hazards during acute COVID-19 and during natural aging in participants of the FHS, after controlling for age and sex (**Fig. 2d**). Expression levels of SAS-1 and IMM-AGE were negatively correlated with those of MAS-1 (**Supplementary Figure 11**).
- 8) **SAS-1/MAS-1 profiles:**

Based on higher vs. lower expression of SAS-1 and MAS-1, four SAS-1/MAS-1 profiles were generated: SAS-1^{high}-MAS-1^{low}, SAS-1^{high}-MAS-1^{high}, SAS-1^{low}-MAS-1^{low}, and SAS-1^{low}-MAS-1^{high} profiles.

 - *The hierarchy of lifespans in the FHS associated with these profiles was: longest with SAS-1^{high}-MAS-1^{low}, intermediate with SAS-1^{high}-MAS-1^{high} and SAS-1^{low}-MAS-1^{low}, and shortest with SAS-1^{low}-MAS-1^{high}*, after controlling for age and sex.
 - **This hierarchy provided the basis to assign**
 - the survival-associated SAS-1^{high}-MAS-1^{low} profile as a transcriptomic proxy for an IC^{high}-IF^{low} state.

- the mortality-associated SAS-1^{low}-MAS-1^{high} profile as a transcriptomic proxy for IC^{low}-IF^{high} state.
- Hence, the SAS-1^{high}-MAS-1^{low}, SAS-1^{high}-MAS-1^{high}, SAS-1^{low}-MAS-1^{low}, and SAS-1^{low}-MAS-1^{high} profiles are representative of IC^{high}-IF^{low}, IC^{high}-IF^{high}, IC^{low}-IF^{low}, and IC^{low}-IF^{high} states, respectively.
- **To validate that SAS-1/MAS-1 profiles are proxies for IC-IF states, we evaluated younger adults intranasally challenged with common respiratory viruses (e.g., influenza). Preservation or development of a survival-associated SAS-1^{high}-MAS-1^{low} was associated with asymptomatic infection (new Fig. 8f-g). In contrast, symptomatic infection was associated with development of the mortality-associated SAS-1^{low}-MAS-1^{high} profile.**

Please note: because of these indicator or proxy functions of SAS-1/MAS-1 profiles for IC-IF states, for ease of explaining our findings, below we refer mainly to the IC-IF states.

Mechanisms: The association of optimal IR with superior immunity-dependent health outcomes is attributable to its linkage to an IC^{high}-IF^{low} state.

- Both reviewers inquired about potential mechanisms.
- To link IHGs to mechanisms, we performed new studies. The sum of our studies shows that:
 - IHG-I is in nearly universal linkage with the survival-associated SAS-1^{high}-MAS-1^{low} profile, a transcriptomic proxy for a IC^{high}-IF^{low} state (**new Fig. 7e-g**)
 - Non-IHG-I grades indicative of nonoptimal IR are in nearly universal linkage with the mortality-associated SAS-1^{low}-MAS-1^{high}, a transcriptomic proxy for IC^{low}-IF^{high} states (**new Fig. 7e-g**).
- We provide **new data** showing that the superior outcomes associated with optimal IR are attributable to its linkage to an IC^{high}-IF^{low} state that predicts and/or is associated with:
 - survival during COVID-19, after controlling for age and sex (**new Fig. 7f,h,i**);
 - longevity in the Framingham Heart Study, after controlling for age and sex (**Fig. 8a; new Fig. 8b**);
 - survival during sepsis (**new Fig. 8d**);
 - less-severe or asymptomatic influenza (**new Fig. 8f-h**); and
 - a further survival advantage in nonagenarians (**new Supplementary Figure 13e**).
- Reviewer 1 posed a fundamental question: *How does antigenic stimulation erode immune status?*
 - To address this question directly, we examined prospective cohorts of natural or experimental infection with common respiratory viruses (e.g., influenza). We show that these viral infections induce a **rapid switch** from an IC^{high}-IF^{low} state to an IC^{low}-IF^{high} state (**new Fig. 8f,g,h**; i.e., switch from SAS-1^{high}-MAS-1^{low} to SAS-1^{low}-MAS-1^{high}). **However, persons who resist a switch from an IC^{high}-IF^{low} to IC^{low}-IF^{high} state also resist symptomatic or severe influenza (new Fig. 8g,h).**
 - We also show that delayed restoration of the IC^{high}-IF^{low} state was associated with severe influenza (**new Fig. 8i**).

- We found that, during natural influenza infection, while persons with an $IC^{high}-IF^{low}$ state before infection rapidly restored the survival-associated $IC^{high}-IF^{low}$ state, nearly 25% failed to restore this protective state post-infection (**new Fig. 8e**).
- Notably, the transcriptomic proxy for $IC^{high}-IF^{low}$ is associated with survival, whereas the transcriptomic proxy for $IC^{low}-IF^{high}$ is associated with mortality. Thus, we suggest that antigenic stimulation erodes immune status by inducing an $IC^{low}-IF^{high}$ state. Hence, our new data (**new Fig. 7d-i**) provide additional supportive evidence that, because of repetitive exposures during aging, some persons lose the capacity to preserve the $IC^{high}-IF^{low}$ status across their lifespan, resulting in a progressive switch from an $IC^{high}-IF^{low}$ to continuous $IC^{low}-IF^{high}$ state.
- To support our viewpoint that optimal IR is an indicator of superior immunity, we now provide mechanistic data linking IR to the trifecta of enhanced T-cell responsiveness, lower T-cell dysfunction, and lower levels of plasma IL-6 (**new Fig. 9c**).
- In our original submission, we reported the results of a large immunophenotyping study ($n \sim 4000$). We showed that immunologic traits classified into four groups: IHG status, regardless of age; age in persons preserving IHG-I or IHG-II (the two most prevalent grades in humans); IHG status and age; and neither IHG status nor age. Immunophenotyping studies in nonhuman primates supported this classification. We have now extended this examination to show the longitudinal analysis of two immune traits (CD28- and CD127- CD8+ T cells) that are associated with IHG status in humans and nonhuman primates (**new Fig. 10c**). These results highlight our hypothesis that levels of immunologic traits may differ by IHG status independent of age alone or by IHG status and age.

IR erosion-resistant and erosion-susceptible phenotypes as a parsimonious basis for inter-individual variation in immunity-dependent health outcomes at any age.

- In our original submission, we suggested that persons classify into those with a capacity to preserve AIIIS-free status vs. a susceptibility to develop AIIIS in clinical settings associated with increased antigenic stimulation. That same concept framed within the context of IR refers to persons with the capacity to preserve optimal IR vs. susceptibility to erode IR during antigenic stimulation. *That is, people classify into the IR erosion-resistant vs. IR erosion-susceptible phenotypes.*
- We show that, across all age ranges, some persons preserve the capacity to preserve metrics of optimal IR linked to a survival-associated $IC^{high}-IF^{low}$ state even in settings of increased antigenic stimulation (**e.g., new Fig. 7b-i**).
- Conversely, we show that, across all age ranges, some persons have a predilection to switch from an $IC^{high}-IF^{low}$ to $IC^{low}-IF^{high}$ state (switch from optimal IR to suboptimal or nonoptimal IR) (**e.g., new Fig. 7b-i**). This predilection signifies the *IR erosion-susceptible phenotype*.
- Metrics of the IR erosion-resistant phenotype were associated with superior immunity-dependent health outcomes, including survival/longevity. New superior outcomes include *survival during sepsis* (**new Fig. 8d**) and *resistance to symptomatic or severe infection with common respiratory viral infections, such as influenza, rhinovirus, and respiratory syncytial virus* (**new Fig. 8g-h**).

- Hence, the IR erosion-resistant and erosion-susceptible phenotypes are a parsimonious basis to explain the wide inter-individual differences in immunity-dependent health outcomes.
- IHG-I and the survival-associated SAS-1^{high}-MAS-1^{low} profile tracking an IC^{high}-IF^{low} state are indicators of the IR erosion-resistant phenotype (**new Fig. 7a**).
- We identified a core set of non-IHG-I grades that are associated with inferior immunity-dependent outcomes; these grades are closely linked to the SAS-1^{low}-MAS-1^{high} profile tracking an IC^{low}-IF^{high} state (**new Fig. 6,7a,7e-g**).

What are some mechanisms by which persons manifest the IR erosion-resistant phenotype?

- Evaluation of natural and experimental viral infection studies highlight two mechanisms by which some persons resist erosion of IR (**new Fig. 8e-i, Fig. 9b**). Our findings suggest that optimal IR relates to resistance to erode IR in response to antigenic challenge and/or rapidly reconstitute optimal IR.

(2) Clarity:

To improve readability and clarity, we have made the following changes.

Key revisions in text:

The paper has been completely rewritten for clarity and focus.

- We have streamlined the paper. We rearranged the presentation of the datasets to allow for a stepwise and systematic flow of information and logic.
- To facilitate this flow, we have divided the study into **four study phases schematized in new Fig. 2e**. To orient the reader, the study phase is noted in each header.
- We connect each of the study phases to show the stepwise progression and linkages in the datasets.
- At the end of each results section, we provide a concise summary of the findings.
- To place our results into context and mitigate repetition in the discussion section, we provide a combined results and discussion.
- A summary is provided as a conclusion. **This summary is new in alignment with the revised text.**
- The original text was written to ascribe causality; we agree that writing the paper in that manner was a source of confusion. In the current version, we have removed this source of confusion. We have added a limitations section and now state: “Thus, while we cannot ascribe cause-and-effect relationships (eroded IR → inferior immunity-dependent healthy outcomes), our findings satisfy the nine Bradford-Hill criteria, the most frequently cited framework for causal inference in epidemiologic studies (**Supplementary Note 11**).”

Revisions in figures for improving clarity:

All figures have been revised for clarity. To help the reader navigate the study, the revised figures provide a step-by-step logical flow of information.

- **Revised Figure 1** provides a broad overview of the unifying framework, outcomes, and cohorts as well as abbreviations/acronyms.
- **New Figure 2e** schematizes the study phases. In *study phases 1 and 2*, we report on the distribution and associations of IHG status across varied cohorts; in *study phase 3*, we report on the distributions and association of transcriptomic metrics of IC-IF status; and in *phase 4*, we report on the mechanisms associated with IR status.

- We have provided schematics to illustrate the key inferences of the study.
- New and revised figures are provided to address the reviewer's comments. The figure numbers and panels have changed.
- Thus, there are revisions to Figures 1-5; Figure 10 is the revised version original Figure 4.
- **Figure 6, Figure 7c-i, Figure 8d-i, and Figure 9a-c** are new.

(3) Summary of key new datasets and additions:

1. As noted, Reviewer 1 asked some very pertinent questions. To address those, we have included new datasets that further supports our viewpoint that optimal IR (previously AIIS-free status) is a determinant of superior immunity-dependent health outcomes.
2. We have now examined metrics of IR in healthy younger adults during natural and experimental infection with common respiratory viruses (e.g. influenza), and *show that resistance to develop symptoms or have less-severe symptoms relates to resistance to erode IR or rapidly restore IR (new Fig. 8e-i)*. However, we also found that some persons who had optimal IR before infection fail to restore this state post-infection, possibly accounting for the waning of IR over time. This addresses the question of reviewer 1: how does antigenic stimulation erode IR?
3. In our original submission, we had shown the association between optimal IR with survival in the Framingham Heart Study. We have now extended this viewpoint, with analysis of survival in sepsis cohorts (**new Fig. 8d**).
4. With new data, we have defined the precise IHGs associated with optimal, suboptimal, and nonoptimal IR (**new Figures 6, 7**).
5. We have included more mechanistic studies, linking the transcriptomic metrics of IR to indicators of immunocompetence and inflammation (**new Fig. 9c**).
6. To complement our findings regarding the immunologic traits that are associated with IHG status, we have included new longitudinal data (**new Fig. 10c**).
7. To provide clearer context, some datasets that were previously shown in the supplement, have been moved to the main figures (**e. g. new Fig. 5c**).
8. To address the comment of reviewer 1 regarding the CD4+ lymphopenia of age and the CD4:CD8 ratio, we have included new preliminary data that are now presented in **new Supplementary note 6**.
9. We have revised all the supplementary notes for clarity and other improvements.

(4) Summary:

We have

- eliminated confusing terms/descriptors/acronyms with a simple-to-understand concept, i.e., immunologic resilience.
- completely revised the paper, flow of information, and figures.
- added figures to address the comments of both reviewers.
- added mechanistic data to support the idea that optimal IR is associated with superior immunologic attributes.
- **placed our study hypothesis in the context of a simple unifying framework (revised Fig. 1 and new Fig. 2E)**. This framework predicts the following algorithm with respect to the association of optimal IR with longevity and other superior immunity-dependent health outcomes.

1. Immunologic resilience (IR) signifies the capacity to preserve and/or restore immunocompetence coupled with control of inflammation during antigenic stimulation.
2. IR metrics indicate an IR continuum: optimal, suboptimal, and nonoptimal.
3. Optimal IR tracks an IC^{high}-IF^{low} state.
4. Some persons have a proclivity to preserve optimal IR even during high-grade antigenic stimulation. This proclivity signifies the IR erosion-resistant phenotype.
5. Conversely, some persons have a predilection to erode IR even in settings of low-grade antigenic stimulation. This predilection signifies the IR erosion-susceptible phenotype.
6. The IR erosion-resistant phenotype is associated with superior immunity-dependent health outcomes, including survival/longevity.
7. The associations of optimal IR with superior outcomes are independent of age and sex.
8. The IR erosion-resistant phenotype is more common in females than males.
9. Females have a longevity advantage independent of IR status.
10. Age, sex, and IR status are independent determinants of lifespan.

Taken together, we suggest that IR tracks a fundamental, antigen-activated program that is intrinsically linked to IC and IF status. The IR erosion-resistant and erosion-susceptible phenotypes provide a parsimonious basis to explain the wide inter-individual variations in immune status among individuals of similar ages. We hope these concepts and changes address the major concerns of the reviewers.

The title of the study has been revised to align with the changes made

“Immunologic resilience forecasts superior immunity, linking sexually dimorphic longevity-promoting mechanisms with resistance to COVID-19, HIV/AIDS, and influenza”

Please note: because the manuscript has been extensively revised, we have not provided a version with the changes tracked. A point-by-point response follows.

Point-by-point response

Reviewer #1:

Comment #1:

Lee et al. describe a remarkable survey in which they seek to align their measures of immune resilience with longevity and susceptibility to AIDS and severe COVID-19 essentially by establishing 4 categories of peripheral blood CD4:CD8 ratios in large international databases of over 40,000 people as well as 300 non-human primates and a number of genetically-heterogeneous mice. This is an enormous tour de force of diverse human populations with much to recommend it.

Response #1.

We very much appreciate your comment regarding the size and scope of our studies. Understandably, given that the CD4:CD8 T-cell ratio has been used extensively by the field, it is natural to consider the immune health grades (IHGs) that we generated as simply a reflection of the CD4:CD8 categories. **However, we stress that the IHGs were not derived by making CD4:CD8 ratio categories.** The IHGs represent four categories of CD8-CD4 equilibrium/disequilibrium states. We regret that our description of the IHGs was unclear on this point. Our description of the IHGs is now in the public domain (*JACI*¹).

As noted in our original manuscript, we use the CD4:CD8 ratio *as a proxy for the level of CD8+ T-cells relative to CD4+ counts, as ratio values <1.0 are simply a mathematical representation of or proxy for higher levels of CD8+ T-cells that are uncompensated by higher levels of CD4+ counts.* To generate the IHGs, we capitalized on this proxy function of ratio values <1.0 and the average CD4+ T-cell count found in otherwise healthy persons (800 cells/mm³) (**Figure 2b**). Hence, the grades are not representative of 4 categories of ratio values; instead, the IHGs track two CD8-CD4 equilibrium states represented by IHG-I and IHG-II and two CD8-CD4 disequilibrium states represented by IHG-III and IHG-IV.

The IHGs **cannot** be intuited by individual inspection of the CD4+ count or ratio values.

- The IHGs track distinct CD8-CD4 profiles as follows: IHG-I (CD8^{lower}-CD4^{highest}), IHG-II (CD8^{lowest}-CD4^{lower}), IHG-III (CD8^{highest}-CD4^{higher}), and IHG-IV (CD8^{higher}-CD4^{lowest}) (**Supplementary Table 4**).
- Hence, CD8-CD4 equilibrium grades IHG-I and IHG-II indicate restrained CD8+ expansion, with higher and lower CD4+ counts, respectively; CD8-CD4 disequilibrium grades IHG-III and IHG-IV indicate unrestrained CD8+ expansion, with higher and lower CD4+ counts, respectively (hence, the terms *equilibrium vs. disequilibrium*).
- The importance of these distinctions is underscored by the fact that persons may have higher CD4+ counts with relatively lower (IHG-I) vs. higher (IHG-III) CD8+ counts; conversely, persons may have lower CD4+ counts with relatively lower (IHG-II) vs. higher (IHG-IV) CD8+ counts (**Supplementary Table 4**).
- Similarly, some persons with IHG-I or IHG-II may have identical higher ratio values, despite having contrasting CD8-CD4 profiles; similarly, some persons with IHG-III or IHG-IV may have identical lower ratio values, despite contrasting CD8-CD4 profiles (**Supplementary Table 4**).

Summary and revisions: The IHGs do not represent four CD4:CD8 ratio categories; rather, they represent the relative equilibrium between peripheral blood levels of CD8+ and CD4+ T-cells. While the CD4+ count and the ratio are the conventional metrics of immune status, we show that their use has likely resulted in

a confounded assessment of immune status. We have clarified these points in the revised text and **revised Supplementary Note 1**.

Comment #2.

“but it is presented in a manner so densely-written, complicated, acronym-loaded and annotated that it is extremely difficult to read and to follow.”

Response #2.

We agree. As noted in the overview, we have completely revised the paper, eliminating difficult-to-understand acronyms (AIIS, ADIS). All acronyms, abbreviations, and concepts are now depicted in Figure 1.

Comment #3.

The authors propose that no matter which populations are examined, be they HIV-positive or negative, young or old, organ transplant recipients, healthy or autoimmune-diseased, primate or not, the absolute numbers and ratios of CD4 T cells and CD8 T cells in the blood act as robust biomarkers of states that they label “antigenic stimulation-induced age-independent immunosenescence (AIIS)” subdivided into partial or complete, as opposed to age-dependent immunosenescence (ADIS). The 4 categories of CD4:8 ratio, labelled “Immune Health Grades” (IHGs) I thru IV are based on clinical AIDS data cutoffs for low CD4+ T cell numbers. On this foundation, Lee et al build an enormous edifice drawing some very interesting and provocative conclusions and attempting to infer causality rather than merely accepting their easy-to-use (but complicatedly-described) CD4:8 ratio indices as biomarkers. This may or may not be the case, but in any event the simple biomarkers deserve consideration by the community, as they are easily available for investigation in many other datasets not explicitly covered in the present study.

Response #4.

- Biomarker function of IHGs. We thank the reviewer for acknowledging that the IHGs are distinctive biomarkers of immune status for consideration by the community. As acknowledged by the reviewer, the advantage of these biomarkers is that they provide prognostication independent of age or other host characteristics, such as HIV or cytomegalovirus serostatus or disease (e.g., systemic lupus erythematosus [SLE], kidney transplant).
- CD4+ cutoffs. We stress that we did not use AIDS data cutoffs for CD4+ T-cell numbers to generate the IHGs.
 - The cutoff of the CD4+ T-cell count used to derive the IHGs was 800 cells/mm³ (**Fig. 2b**), which approximates the median CD4+ T-cell count in 16,126 otherwise healthy HIV-seronegative (HIV-) persons (**Supplementary Table 1**).
 - This CD4+ count threshold is in the public domain: in our studies published in the *New England Journal of Medicine* and *JAMA Internal Medicine*, we used 800 CD4 cells/mm³ as an outcome to signify normalization of CD4+ counts. Since then, we have realized that the findings were possibly confounded because CD4+ counts >800 cells/mm³ may occur in the context of IHG-I (equilibrium) or IHG-III (disequilibrium) (**Fig. 2b,4i**), and that near-normalization of immune functions in HIV+ persons occurs with achievement of IHG-I but not IHG-III during antiretroviral therapy (data not shown).
 - **Revisions:** We have simplified the description for how and why the IHGs were derived.
- Causality: We agree with the reviewer that it is an error to ascribe causality.
 - **Revisions:** We have eliminated any reference to causality as a basis for our study.

- As noted, we have revised the discussion to state: Thus, while we cannot ascribe cause-and-effect relationships (eroded IR → inferior immunity-dependent healthy outcomes), our findings satisfy the nine Bradford-Hill criteria⁶, the most frequently cited framework for causal inference in epidemiologic studies (**Supplementary Note 11**).

Comment #4.

Notwithstanding the above considerations, I do feel that this paper is based on several questionable statements and assumptions that clearly mould the authors' paradigm of "immunosenescence" and probably strongly reflect the HIV background of the senior author. The Abstract plunges straight in to distinguishing age-independent antigen-dependent immunosenescence and antigen stimulation-dependent immunosenescence as if these two could be clearly separated. I do not see how this can be achieved and would ask the authors to simplify their manner of writing to clarify their meaning without using catch-all designations like "immunosenescence" which are by no means standardised. These mean different things to different people. Specifically:

What is the evidence that ADIS exists, as opposed to AIIS? All individuals studied here will have been challenged by many antigens. Only a comparison with gnotobiotic mice could truly be used to define ADIS, and maybe not even then due to autoantigens. I believe that it would be preferable to avoid these confusing invented acronyms and stick only to a description of what is actually measured. For example, the authors refer here to a paper in press as follows: "Supporting this inference, AIIS corresponds to age-independent loss of immunologic resilience, a trait that we recently showed associated with severe COVID-19 and shorter lifespan". This paper is now out, but I cannot see that it refers to "AIIS" anywhere. So why in the present paper?

Response #4.

- We agree with the reviewer that use of *immunosenescence* in two different contexts (AIIS and ADIS) is confusing. As noted in the overview, AIIS corresponds to loss of immunologic resilience (IR). To avoid confusion and remain aligned with our recent publication in *JACI*, we only refer to IR in the revised manuscript.
- In agreement with the reviewer, the revised results section only describes the distributions and associations of the IR metrics. Inferences are limited to what we actually measured.
- For clarity, based on our recent publication in *JACI*, we assign IR metrics that signify optimal IR upfront allowing for a clearer picture of our study design (**new Fig. 2a,e**).
- The reviewer is spot-on with the fundamental question he/she articulated (which we have been trying to address): **What is the evidence that ADIS exists, as opposed to AIIS?** This is a critical question to address, given the enormous efforts to develop interventions to improve healthspan and lifespan. We agree with the reviewer that human systems perhaps are inadequate to differentiate between these two processes, if at all. Acknowledging the difficulty in making this distinction in humans, we have removed all reference to use of the acronym ADIS.
- We remain open to the possibility that there is distinction between what we previously designated as AIIS (now termed erosion of IR) vs. an age-dependent process.
 1. We believe that the distinction resides in the fact that erosion of IR (AIIS) is the "byproduct of aging" vs. "due to aging", i.e., the longer one lives the greater the exposure to antigens. In turn, greater exposure increases the chances that an individual will erode IR (i.e., develop AIIS).

2. However, younger individuals also manifest metrics suggestive of suboptimal or nonoptimal IR. In our surveys of HIV-seronegative individuals shown in our paper and in our unpublished data, we found that about 5% of otherwise healthy younger adults have metrics suggestive of “unsuspected immunosuppression.”
3. Additionally, we show that, in certain contexts of increased antigenic stimulation, the level of IR eroded in younger individuals can be comparable to that observed in older persons not experiencing acute antigenic stimulation.
4. **In four different contexts, we also show that optimal IR can be restored with mitigation of antigenic stimulation (COVID-19, female sex workers, HIV+ persons on ART, and acute influenza or other viral infections).** We believe that it is unlikely that processes that drive ADIS are reversible.
5. We show that transcriptomic metrics of IR predict survival in the COVID-19 cohort and lifespan in the Framingham Heart Study, independent of both age and sex. These associations of IR metrics with longevity independent of age provide support for the idea that processes independent of age influence lifespan.
6. In a large-scale immunophenotyping study of nearly 4000 humans, we showed that immunologic traits classify into four categories whose levels differ by (i) IHG status irrespective of age; (ii) age in persons with IHG-I or IHG-II (the two most prevalent IHGs in human populations); (iii) both IHG status and age; and (iv) neither IHG status nor age (**revised Fig. 10a-b; new Fig. 10c**). Similar patterns were observed in a more-limited profiling of nonhuman primates (**new Fig. 9d**). These observations speak to the possibility that deficits in immune status may be attributable to two processes: eroded IR (AIIS) and age. However, because IR levels decline with age (due to repeated antigenic stimulation), the attributes of suboptimal or nonoptimal IR may be misattributed to age.

Revisions made: Based on these lines of evidence, it remains conceivable that eroded IR (AIIS) attributable to antigenic stimulation is a process that is a “byproduct of aging” rather than “due to aging”. Hence, we remain open to the possibility that other processes directly due to aging (e.g., cellular senescence) are operative that contribute to immune deficits. We realize that these are theoretical viewpoints that have broad implications for development of “anti-aging” strategies that need additional testing. In the revised manuscript, **we suggest a distinction between deficits attributable to erosion of IR vs. aging as a point of possibilities in the introduction and discussion.**

Comment #5. Coming from the HIV field, the beginning of the Introduction makes the assumption that HIV infection models premature immune aging, but this is not as commonly accepted in the field as the authors seem to believe. Thus, it is stated that “both SLE and HIV infection are viewed as model systems of “premature aging” and two references are cited in support of this claim, neither of which represents anything like a consensus. They make statements focusing on changes that may be attributable to low-grade, HIV-associated antigenic stimulation (although they may also be caused by the therapy and other effects of the virus) which “degrades immune status” – but in what way does Ag stimulation “degrade” immune status? It is the basis of adaptive immunity, not degradation. They then conclude that “In contrast, deterioration in immune status attributable to aging per se develops over time.” But what is “aging per se”? This is also a matter of intense debate and is not accepted as a condition that can be viewed as meaningful in this context without a definition of what is meant by it.

Response #5.

- We agree that the concept of premature aging lacks definitions and consensus. What we were alluding to through these statements was that, at any age, a subset of individuals experiencing higher levels of antigenic stimulation may manifest loss of IR to an extent that is also observed in older persons.
- **Revisions:** To avoid confusion, we have revised the introduction without reference to HIV as an experimental model for premature aging.
- The reviewer asked a fundamental question: “...but in what way does Ag stimulation “degrade” immune status?”

Our data suggest the following algorithm to propose a mechanism by which antigenic stimulation erodes immune status. We show that:

1. Metrics indicative of optimal IR track an $IC^{high}-IF^{low}$ state.
2. Settings associated with increased antigenic stimulation are associated with a **switch** from an optimal IR status to suboptimal or nonoptimal IR status.
3. This switch corresponds to a switch from an $IC^{high}-IF^{low}$ to $IC^{low}-IF^{high}$ state.
4. This switch was observed in the settings of low-grade (e.g., aging), moderate-grade (COVID-19, risk factor-associated antigenic stimulation, influenza), and severe-grade (sepsis, HIV) antigenic stimulation.
5. Mitigation of antigenic stimulation was associated with reconstitution of optimal IR status (i.e., switch from an $IC^{low}-IF^{high}$ to $IC^{high}-IF^{low}$ state).
6. However, despite mitigation of antigenic stimulation, some persons manifest a persistent $IC^{low}-IF^{high}$ state.
7. Metrics tracking an $IC^{low}-IF^{high}$ state show an association of this state with mortality and other inferior immunity-dependent health outcomes.

Revisions made: The points discussed above are now integrated into the text. We have also provided a revised supplementary discussion of these points.

Comment #6.

And I have to say that I am puzzled by the subsequent statement that this state is one “representing physiologic involution of the immune system attributable to aging”. What do the authors mean by “physiologic involution”? It is also not the case that the presence of “CD4+ and CD8+ senescent T cells (CD28- CD4+ or CD8+ T cells), and T-cell receptor excision circles, and telomerase reverse transcriptase” do represent “4 conventional markers of immunosenescence”. T cells are not necessarily senescent because they are CD28-negative, TCR excision circles provide an indicator of antigen exposure and HTERT marks activation.

Response #6.

This comment is similar to comment #4, referring to the issue of immune traits attributable to the IR process vs. age vs. both. We were trying to allude to the possibility that processes distinct from an antigen-driven process influence immune trait levels during aging.

Revisions:

- To avoid confusion, these statements have been removed.
- Since levels of CD28- CD4+ and CD8+ T-cells increase with age, they have been viewed as a proxy for the immune aging process. We have now plotted the levels of CD28- CD8+ T-cells over time by IHG status and age. These data, shown in **new Figure 10c**, suggest that both IHG status and age

influence levels of these traits. However, we agree that inclusion of these marker sets is a source of confusion. We have therefore removed all analysis of this the gene signature (c-AIIS signature) that correlated with these immune traits.

Comment #7.

In the Results, illustrating the assumption that the same mechanism lies behind the observed phenomena of CD4:8 ratios, the authors state that “CD4+ lymphopenia is a characteristic of aging, severe COVID-19, SLE, and HIV. While this is certainly true for HIV and may be for SLE, for different reasons, is it really true for aging?? For example, <https://pubmed.ncbi.nlm.nih.gov/27547234/> indicates very little CD4 lymphopenia with age. Disease states are completely different. Refs 31, 32 do not address this specifically. The authors then also state that “a higher CD4:CD8 ratio is viewed as a biomarker of immunocompetence, ref. 34” – yes, Ref. 34 is for HIV, but this is not necessarily also the case in aging, eg. see <https://pubmed.ncbi.nlm.nih.gov/27927759/>

Comment 7a: In the Results, illustrating the assumption that the same mechanism lies behind the observed phenomena of CD4:8 ratios,.....

Response #7a: CD4+ mechanisms

- Foremost, we wish to emphasize that we are **agnostic to mechanisms**. Thus, we are not ascribing the same mechanism behind the ratios, CD4+ or CD8+ counts, or IHGs.
- Our focus is twofold:
 - understanding the extent to which different diseases or conditions are associated with IHGs, regardless of mechanism.
 - defining whether the IHGs in different disease contexts have similar biomarker functions of predicting/associating with immunity-dependent outcomes.
- Being **mechanism agnostic** is important, as it is common in the scientific field to compare groups of individuals with differing host characteristics (e.g., age, HIV serostatus). However, such comparisons are *prone to confounding* as they do not account for the underlying heterogeneity in immune status. For example, a comparison of younger vs. older persons is a conflated assessment of populations with differing IHG distributions: IHG-I is more common in younger persons and non-IHG-I grades associated with inferior immunity-dependent outcomes are more common in older persons (**new Fig. 6**). Mitigation of this confounding may be an important step, as we found that some trait levels are similar in younger and older persons with IHG-I or IHG-II (**revised Fig. 10a,b; new Fig. 10c**).
- What we are alluding to is that persons with the same IHGs but arising in the context of different diseases or conditions is associated with similar levels of immunosuppression regardless of age.

Revisions: We have created a **new supplementary note 6** discussing these points.

Comment 7b: the authors state that “CD4+ lymphopenia is a characteristic of aging, severe COVID-19, SLE, and HIV. While this is certainly true for HIV and may be for SLE, for different reasons, is it really true for aging??

Response #7b: CD4+ lymphopenia of age

- CD4+ lymphopenia of age is an aspect of aging after one accounts for confounding factors, which is what we had done in our original submission and is discussed further below.

- Confounding occurs because of a) small numbers and b) a comparison is made between younger vs. older persons without regard to the fact that there are 4 IHG states that may be present in both groups and the CD8-CD4 profile of the IHGs differs.
- This confounding is illustrated in the study cited by the reviewer: <https://pubmed.ncbi.nlm.nih.gov/27547234/>. In this study, Lin et al evaluated 135 individuals, and inspection of Figs 1-4 of this report reveals that most participants were older. A slope cannot be adequately generated in this setting, as the reference population (younger individuals) is underrepresented. Our analysis comprised a nearly 4000-person cohort spanning ages 18 to >90 years, with good representation across age (**see new Supplementary Figure 9**).
- More importantly, a comparison of younger vs. older persons without regard to the IHGs is confounded because it masks the immense complexity in CD8-CD4 profile changes that occur with age.
 - IHG-I is the most common grade in human populations, as well as in younger nonhuman primates (**revised Fig. 2f, 4d**). With age, there is a switch from IHG-I to a non-IHG-I grade (**revised Fig. 2; new Fig. 6b**). Thus, a comparison between younger and older persons does not account for the fact that the prevalence of two IHGs is showing opposing frequencies: the prevalence of IHG-I is *decreasing* whereas the prevalence of IHG-III is *increasing*. That is, by failing to preserve IHG-I, some persons are experiencing a CD4+ loss, whereas, by developing IHG-III, others are preserving higher CD4+ counts with age. Immune traits differ by equilibrium status (**Fig. 10a,b**). Thus, depending on the representation of the IHGs in a study population, the CD4+ count decline may not be discernable.
 - Similarly, during aging, persons may switch from IHG-I to IHG-II or IHG-IV (**Fig. 2f**); a switch to IHG-II signifies CD4+ lymphopenia but with restrained expansion of CD8+ T-cell counts, preserving CD8-CD4 equilibrium; in contrast, a switch to IHG-IV signifies CD4+ lymphopenia but with relatively higher CD8+ counts.
 - The less-confounded way to examine CD4+ lymphopenia seen with age is to focus on CD4+ counts among younger and older persons by IHG status. These data categorized by IHG-I and IHG-II status were shown in our original submission Fig. 2D.
 - The **response letter Figure 1a** (*shown below*) depicts the differences in CD4+ and CD8+ counts and ratios in younger and older persons by IHG status.
 - No substantial differences in the values were observed in younger or older persons with IHG-III.
 - CD4+ counts were lower by 9.9%, 9.9%, and 15.9% in older persons with IHG-I, IHG-II, and IHG-III, respectively, vs. younger persons with the same grades.
 - CD8+ counts were lower by 28.7%, 23.5%, and 16.4% in older persons with IHG-I, IHG-II and IHG-IV, respectively, vs. younger persons with the same grades. There were no significant differences in CD8+ levels in younger or older persons with IHG-IV.
 - Thus, there was a disproportionately greater decline in CD8+ T-cells than CD4+ T-cells in older persons with IHG-I or IHG-II. Because of this disproportionality, ratio values were higher by 22.2% and 21.7% in older persons with IHG-I and IHG-II, respectively, compared to younger persons with these same grades.
 - In contrast, ratio values were lower in older persons with IHG-IV compared with younger persons with the same grade.

- However, these nuances are obscured if these variables had been examined by age alone, which is plotted in **response letter Figure 1b,c** (shown below). ***These data are a conflated assessment, as they show a progressive decline in CD4+ and CD8+ T-cell counts by age and an increase in the CD4:CD8 ratio followed by a decrease beginning at approximately age 70 years (response letter Figure 1c).***
- The question arises: why do ratio values increase initially with age and then decline in the older age population?
 - This has to do with the changes in distributions of the IHGs that occur with age. The initial rise in ratio values is attributable to increases in ratio values in older persons with IHG-I and IHG-II.
 - However, with age there is a concomitant increase in IHG-III and IHG-IV tracking an inverted ratio (6.2% individuals in the 70- to 79-yr and 12.3% of those in the 80- to 103-yr age stratum have IHG-III or IHG-IV vs. 3.3% in individuals <40 yr) (**revised Fig. 2f**).
 - Thus, the conflation of ratio values of IHG-I, IHG-II, IHG-III, and IHG-IV are lower in older persons attributable to the increased prevalence of IHG-III and IHG-IV in this group (**response letter Figure 1c**).

Revisions: We have created a new supplementary note 6 discussing these points.

Comment 7c: The authors then also state that “a higher CD4:CD8 ratio is viewed as a biomarker of immunocompetence, ref. 34” – yes, Ref. 34 is for HIV, but this is not necessarily also the case in aging, eg. see <https://pubmed.ncbi.nlm.nih.gov/27927759/>

Comment #7c: CD4:CD8 ratio

We had discussed this point in the original submission. Our data showed that a higher ratio value must be considered in the context of age. In our original text and the accompanying Supplementary materials, we provided many details about this point. In the main text we noted: “Additional analysis showed that among persons older than 40 years preserving IHG-I or IHG-II, a CD4:CD8 ratio ≥ 2.5 may serve as a metric of an “aged” CD8-CD4 equilibrium or ADIS (**Supplementary Note 3**).” We have closely evaluated the data in an attempt to understand the basis of higher ratio values in older persons preserving IHG-I or IHG-II and their possible clinical implications.

- As described above, age is associated with an increase in ratio values in persons with CD8-CD4 equilibrium grades IHG-I or IHG-II. An increase in ratio values is not possible in persons with IHG-III or IHG-IV because, by definition, these IHGs have ratio values less than unity.
- We stated in our original submission that the singular feature of ADIS was a ratio value that exceeds 2.5.
- To illustrate these points in the original supplementary note 3, we showed the kernel density plots of ratios by IHG status.
- These kernel plots are shown below in **response letter Figure 1d,e**. They describe the prevalence (density) of ratio values in younger (<40 yrs) and older (≥ 40 yrs) individuals from the HIV- UCSD cohort (panel d) and the Sardinia cohort (panel e). The median ratio values in 13,703 otherwise healthy individuals approximates 1.75 (Supplementary Table 1).
- These findings illustrate that:

- (i) Younger and older individuals may have ratio values <1.0 , signifying CD8-CD4 disequilibrium grades of IHG-III or IHG-IV.
- (ii) Younger compared with older persons with IHG-I or IHG-II are more likely to have ratio values that approximate the median ratio values in healthy persons (~ 1.75).
- (iii) Older compared with younger persons with IHG-I or IHG-II are more likely to have ratio values that exceed the point where the kernel density plots by age intersect, i.e., 2.5 in the UCSD cohort and 3.0 in the Sardinia cohort.
- (iv) Thus, older persons (red line) preserving IHG-I or IHG-II are more likely to manifest higher ratio values. Higher ratio values were observed in a subset of older persons with CD8-CD4 equilibrium grades IHG-I or IHG-II.
- Therefore, regardless of age, ratio values that approximate the median ratio values (~ 1.75) in persons with IHG-I or IHG-II are a sign of greater immunocompetence. Ratio values >2.5 in older persons with IHG-I or IHG-II are a sign of an “aged CD8-CD4 equilibrium” and may be associated with negative outcomes.

Revisions: We have created a new supplementary note 6 discussing these points.

a

Variables	15-39 yrs (younger)			≥70 yrs (older)			older vs. younger		
	n	median	IQR	n	median	IQR	% difference	Wilcoxon- P	
IHG-I	CD4+	986	1199	1005 - 1472	308	1081	931 - 1238	-9.9	< 0.001
	CD8+	986	570	454 - 725	308	406	289 - 574	-28.7	< 0.001
	CD4+:CD8+ ratio	986	2.18	1.74 - 2.72	308	2.66	2.00 - 3.69	+22.2	< 0.001
IHG-II	CD4+	216	689	610 - 752	189	621	526 - 709	-9.9	< 0.001
	CD8+	216	377	295 - 470	189	288	191 - 389	-23.5	< 0.001
	CD4+:CD8+ ratio	216	1.75	1.37 - 2.25	189	2.13	1.51 - 2.93	+21.7	< 0.001
IHG-III	CD4+	23	1046	885-1238	17	996	912-1140	-4.8	0.626
	CD8+	23	1281	1027-1410	17	1217	1070-1392	-5.0	0.787
	CD4+:CD8+ ratio	23	0.9	0.81-0.96	17	0.91	0.78-0.94	1.1	0.584
IHG-IV	CD4+	18	658	594-713	26	553	430-670	-15.9	0.034
	CD8+	18	752	674-857	26	807	589-998	7.3	0.456
	CD4+:CD8+ ratio	18	0.86	0.77-0.93	26	0.72	0.60-0.82	-16.4	0.005

Response letter Figure 1 legend (a) The number of participants and median and IQR of CD4+ T-cell counts, CD8+ T-cell counts, and CD4:CD8 ratio by age strata and IHG in the SardiNIA cohort. The %difference refers to difference in median values between older (≥70 years old) vs.

younger (<40 years old) participants. **(b)** Line plots of median CD4+ and CD8+ T-cell counts as well as CD4:CD8 T-cell ratio by age and sex in the SardiNIA cohort. **(c)** Line plots of overall median CD4:CD8 ratio by age in the SardiNIA cohort depicting the basis for confounding. **(d,e)** CD8-CD4 equilibrium states in the HIV- UCSD **(d)** and SardiNIA cohorts **(e)**. Kernel density estimates plots of the CD4:CD8 T-cell ratio and the corresponding CD8-CD4 equilibrium/disequilibrium states noted on the x-axis by ages: <40 vs. ≥40 years old. *P* values for differences in distribution. Vertical line at 1.0 represents the cut-off to demarcate CD8-CD4 disequilibrium status, i.e., persons with IHG-III or IHG-IV (indicated by red shaded area). Of note, the intersection of the plots at ratio of 2.5 in panel **d** and at ratio of 3 in panel **e** is indicated by vertical lines. suggesting that older persons with IHGs I or II with ratio ≥2.5 or ≥3 may show sign of aged CD8-CD4 equilibrium (i.e., age dependent changes in immune profiles; indicated by blue shaded area). In our literature survey of 13,703 HIV- persons worldwide, the median CD4:CD8 ratio was 1.76 (IQR: 1.57-2.04; **Supplementary Table 1**) and is indicated by a vertical dashed line.

Comment 8: The concept of CD4:8 equilibrium is introduced but what does this mean? This is not clear, also not from Note 2. The variable numbers and ratios of CD4:CD8 cells say nothing about an equilibrium.

Response 8:

- By definition, equilibrium implies “a state in which opposing forces or influences are balanced.” In the case of IHGs, the opposing forces are the CD4+ vs. CD8+ counts.
- As mentioned, we use the CD4:CD8 ratio as a proxy for level of CD8+ T-cells relative to CD4+ T-cells. As indicated, ratio values <1.0 are simply a mathematical representation of or proxy for higher CD8+ levels uncompensated for by higher CD4+ counts. People could have CD4+ counts higher or lower than the average CD4+ count found in healthy populations (800 cells/mm³).
- Hence, by using cutoff ratio values <1.0 vs higher ratio values and CD4+ cell count of higher vs. lower than average, we derived the relative equilibrium vs. disequilibrium between CD8+ and CD4+ T cells.
- Disequilibrium implies unrestrained CD8+ expansion with higher (IHG-III) or lower (IHG-IV) CD4+ cell counts. Equilibrium implies restrained CD8+ expansion with higher (IHG-I) or lower (IHG-II) CD4+ T-cell counts.
- If we had used ratios or CD4+ counts alone vs. CD8-CD4 profiles, we would not have been able to assign a primordial immune state that is associated with optimal IR. The primordial nature of IHG-I was supported in nonhuman primate studies (**Fig. 4d,e**).

Revisions: We have created a new supplementary note 6 discussing these points.

Comment 9: The association of age with a disproportionately greater loss of CD8+ vs. CD4+ T-cells, representing IHG feature 3, may also be caused by CD8 clonal attrition due to chronic stimulation by persistent pathogens.

Response 9: We agree with this viewpoint. There are three groups in which we observe this disproportionality: IHG-I, IHG-II, and IHG-IV (**response letter figure 1**).

Comment 10: I would suggest that your very interesting survey is presented in a far more neutral fashion and made less difficult for readers to wade through by reducing the use of newly-invented acronyms. In the light of the above comments, I would suggest a more neutral descriptive and less interpretative title such as “Usefulness of modified CD4:8 ratio assessments as biomarkers of disease and longevity”

Response 10:

- As noted in the overview, we have made extensive changes to the manuscript, removed all acronyms.
- The manuscript has been rewritten in a neutral tone.
- We very much appreciate your providing a suggestion for a title. Given that we have (i) identified the biomarkers of a primordial immunologic state suggestive of optimal IR and (ii) expanded our mechanistic datasets, we have revised the title to reflect this.

Reviewer #2 (Remarks to the Author):

Comment 1. This manuscript presents an analysis of clinical and immunologic parameters in varied cohorts in order to inform on the relationship between health status, antigenic stimulation-induced age-independent immunosenescence (AIIS) and age dependent immunosenescence (ADIS).

The authors developed metrics to differentiate among AIIS-free, partial-AIIS, and complete-AIIS status, as well as between AIIS and ADIS. These metrics are based on four immune health grades (IHGs) derived by co-indexing the CD4+T-cell count and CD4:CD8 T-cell ratio. When available, they also related IHGs to cell phenotype data and gene expression signatures that associate with survival/mortality (SAS-1, MAS-1). They analyzed a large spectrum of cohorts of humans (n=43,286 including healthy children to centenarians, people living with HIV, lupus patients, and COVID-19 patients), non-human primates (n=279), and mice (n=332).

The authors report that AIIS and ADIS do not represent a single and same program, but are associated with distinct immunologic and clinical traits, that should not be confused with one another. Their findings indicate that the combination and interaction of ADIS, AIIS and sex influences lifespan.

This work focuses on interesting concepts like AIIS and ADIS, and important health contexts. The findings have a potential relevance with regards to the implementation of AIIS status monitoring using simple metrics like IHGs for prevention, prognostic, and clinical care.

Despite these interesting aspects and the size of the study, I see two important drawbacks.

Response 1. We thank the reviewer for the careful read of our study and acknowledging the strengths of the study.

Comment 2. The conclusion that combinatorial effects of AIIS and ADIS together with gender contribute to lifespan differences and longevity is sound but not particularly original, and already quite established in the field. It refers to the notions of biological and chronological age and their consequences on health.

Response 2. We respect these views, and this is an important point. However, we believe that our concepts related to AIIS are not referring to biological age, as currently defined in the literature. Both biological and chronologic age anchor findings to the same reference point, namely age. Thus, terms such as biological age are referring to changes relative to age. We are defining a process that *occurs with but is not dependent on age. None of our metrics is anchored to age as a reference point.* Within each age stratum, we identify individuals who have features of optimal, suboptimal, or nonoptimal IR in younger

and older persons. We are referring to a loss of a primordial immunologic state that is associated with superior immunocompetence and lower inflammation (optimal IR). With IR metrics, we are simply quantifying the degree to which a deviation from this primordial optimal IR state has occurred regardless of age. In all our models of survival, we adjusted or controlled for age. Biological age, as defined, has no “normative” value for comparison to except age.

Comment 2. Moreover, claiming that the authors’ findings provide a refined mechanistic framework to understand the heterogeneity in aging phenotypes including lifespan is an overstatement. The study is not mechanistic.

Response 3. We have removed all discussion of aging phenotypes. However, as noted in the overview, we have now provided substantial new mechanistic data linking IR status to gene expression signatures that predict survival/mortality independent of age and serve as proxies for IC-IF states. These new mechanistic data support our mechanistic framework.

Comment 3. The content of the manuscript is difficult to grasp, in part due the many cohorts presented in the study, and the way it is written. The original findings and conclusions of the work should be better highlighted, for instance using subheadings and adapted figures.

Response 3. As noted in the overview, we have made extensive changes to the manuscript to streamline content and interpretation.

References

1. Lee, G.C., *et al.* Immunologic resilience and COVID-19 survival advantage. *J Allergy Clin Immunol* **148**, 1176-1191 (2021).
2. Marconi, V.C., Krishnan, V., Ely, E.W. & Montano, M. Immune Health Grades: Finding Resilience in the COVID-19 Pandemic and Beyond. *J Allergy Clin Immunol* (2021).
3. Montano, M., Oursler, K.K., Xu, K., Sun, Y.V. & Marconi, V.C. Biological ageing with HIV infection: evaluating the geroscience hypothesis. *The Lancet Healthy Longevity* (2022).
4. PrabhuDas, M., *et al.* Research and resource needs for understanding host immune responses to SARS-CoV-2 and COVID-19 vaccines during aging. (Nature Publishing Group, 2021).
5. Alpert, A., *et al.* A clinically meaningful metric of immune age derived from high-dimensional longitudinal monitoring. *Nat Med* **25**, 487-495 (2019).
6. Fedak, K.M., Bernal, A., Capshaw, Z.A. & Gross, S. Applying the Bradford Hill criteria in the 21st century: how data integration has changed causal inference in molecular epidemiology. *Emerg Themes Epidemiol* **12**, 14 (2015).

REVIEWER COMMENTS

Reviewer #1 (Remarks to the Author):

I greatly appreciate the authors' re-write of their paper following my comments, and their extensive detailed rebuttal and clarifications. I did realize that the IHGs were slightly more sophisticated than simple CD4:8 ratios, but I think that the current version of the paper does make this clearer, and the impressive set of correlations in widely variable human and non-human cohorts based on this simple metric is quite remarkable. I am still not sure about the use of the term "equilibrium", but OK, I am sure that this paper will generate much discussion and will represent an important resource for comparing these different datasets. The trillion dollar question remains what are the reasons for the differences between "resilient" and "non-resilient"? In humans, the authors present one SNP that may be related to intrinsic control of CD8 cell levels, and presume that the differences must be genetic. More discussion of this would be welcome, and looking at human twin datasets and different mouse strains would be interesting, as well as familial longevity (eg. the Leiden Longevity Study). But perhaps in a subsequent paper.....I think this one is big enough already!

Reviewer #2 (Remarks to the Author):

The authors have provided answers to my comments.

Reviewer #3 (Remarks to the Author):

Ahuja et al describe an impressive study on the use of CD8+ and CD4+ T-cell levels as markers of immunocompetence and inflammation. The major strength of the study is the diversity of the human cohorts studied, as well as both primate and mouse models. While the basic concept at the heart of the study is a relatively simple one, im afraid its impact will be lessened by a very dense and acronym heavy manuscript that makes it challenging to read. This would be a shame as this simple measure could be assessed in many different studies and contexts with strong potential to help clinical strategies and patient stratification approaches. In addition I have some other concerns listed below:

A big assumption is made that aging equals increased antigenic stimulation. This is a massive oversimplification of aging, which for effects on T cells alone will be driven by multiple factors such as thymic involution, decline in stem cell production, and tissue migration and residency of memory cells, etc. This assumption should be corrected and modified.

The previously reported studies that CD8 T cells decline at faster rates than CD4 T cells with increasing age could simply explain the age effects seen in this study (see for example Vrisekoop, N. et al. Sparse production but preferential incorporation of recently produced naive T cells in the human peripheral pool. Proc. Natl Acad. Sci. USA 105, 6115–6120 2008 or Patin E et al, Natural variation in the parameters of innate immune cells is preferentially driven by genetic factors. Nat Immunol. 2018 Mar;19(3):302-314. doi: 10.1038/s41590-018-0049-7.)

What was the justification for not correcting p values for the multiple tests performed? Given

the amount of comparisons and tests performed there is a high risk of false discovery and these should be controlled for, especially as many of the tests are essentially validation of the initial hypotheses.

Why were Pearson correlations used which is more suited to parametric data sets, when the rest of the tests applied are more suited for non-parametric tests?

The SAS and IR markers are likely to be highly dependent on each other, for example IL7R and CCR7 are expressed CD4 & CD8 T cells. How did the authors account for this?

Given how much this study has benefited from other published studies, the newly generated data sets should either be deposited in repositories or made available as supplemental data files, so that other research teams can re-use them.

Alternatively I'm satisfied with the reviewers response, they have clearly outlined their analytical strategy which makes sense and is justified.

This manuscript is a huge amount of work and I think will contribute to the field.

RESPONSE TO REVIEWER COMMENTS

Green- laudatory comments for which we are very grateful.

Red- comments/questions

Blue - response provided

Reviewer #1 (Remarks to the Author):

I greatly appreciate the authors' re-write of their paper following my comments, and their extensive detailed rebuttal and clarifications. I did realize that the IHGs were slightly more sophisticated than simple CD4:8 ratios, but I think that the current version of the paper does make this clearer, and the impressive set of correlations in widely variable human and non-human cohorts based on this simple metric is quite remarkable. I am still not sure about the use of the term "equilibrium", but OK, I am sure that this paper will generate much discussion and will represent an important resource for comparing these different datasets. The trillion dollar question remains what are the reasons for the differences between "resilient" and "non-resilient"? In humans, the authors present one SNP that may be related to intrinsic control of CD8 cell levels, and presume that the differences must be genetic. More discussion of this would be welcome, and looking at human twin datasets and different mouse strains would be interesting, as well as familial longevity (e.g. the Leiden Longevity Study). But perhaps in a subsequent paper.....I think this one is big enough already!

Comment #1

I am still not sure about the use of the term "equilibrium", but OK, I am sure that this paper will generate much discussion and will represent an important resource for comparing these different datasets.

Response #1

Our concept of CD8-CD4 equilibrium vs. disequilibrium was published in the *Journal of Allergy and Clinical Immunology (JACI)* last year. Additionally, our report was accompanied in *JACI* by an editorial entitled: "Immune health grades: Finding resilience in the COVID-19 pandemic and beyond". Additional studies have quoted our work, and our ongoing studies also refer to the concepts of CD8-CD4 equilibrium vs. disequilibrium. Thus, these concepts are embedded in the literature. This point was noted in our original response letter.

As highlighted in our main text and supplement, the terms equilibrium and disequilibrium are important paradigms that distinguish our work. Since these terms refer to the relative proportions of CD8+ vs. CD4+ counts, with or without higher CD4+ counts, we are not aware of a viable alternative term. Additionally, the equilibrium vs. disequilibrium framework places a spotlight on CD8+ T-cells.

Moreover, these concepts have clinical relevance, as extensive CD4+ lymphopenia occurs in the context of CD8-CD4 equilibrium linked to IHG-II as in COVID-19 (with ratio >1.0) vs. in the context of CD8-CD4 disequilibrium linked to IHG-IV as in untreated HIV infection (with an inverted CD4:CD8 ratio). We demonstrate that CD4+ lymphopenic

COVID-19 patients with CD8-CD4 equilibrium are younger and have fewer comorbid conditions than those with disequilibrium. CD8-CD4 disequilibrium linked to IHG-III (signifying CD8-CD4 disequilibrium with high CD4+ counts) is a distinctive feature of SIV+ sooty mangabey monkeys, as well as some HIV+ patients on antiretroviral therapy. Additionally, in the supplement we have extensively discussed the relationship between CD8-CD4 equilibrium/disequilibrium and CMV serostatus.

We have provided extensive justification for this nomenclature in the supplement. We agree with the reviewer that our studies will generate discussion. For these reasons, we request continued use of the terms CD8-CD4 *equilibrium* and *disequilibrium*.

Comment #2

The trillion dollar question remains what are the reasons for the differences between “resilient” and “non-resilient”? More discussion of this would be welcome, and looking at human twin datasets and different mouse strains would be interesting, as well as familial longevity (e.g. the Leiden Longevity Study). But perhaps in a subsequent paper. I think this one is big enough already!

Response #2

We are currently working on the questions that the reviewer poses. We are still in the early stages of our analysis. However, please note, we did provide murine data from the Collaborative Cross-RIX mice in the main text **Fig. 4f** and pre-CC mice in **Supplementary Fig. 13i**, suggesting a genetic basis for these phenotypes. For purposes of brevity, the discussion related to host genetic features in humans was restricted to the **Supplementary Note 5**. We are open to any collaborations with the Leiden Longevity Study and will reach out to them. We agree, the study is big enough already without inclusion of additional datasets.

Reviewer #2 (Remarks to the Author):

The authors have provided answers to my comments.

Reviewer #3 (Remarks to the Author):

Comment 1

Ahuja et al describe an impressive study on the use of CD8+ and CD4+ T-cell levels as markers of immunocompetence and inflammation. The major strength of the study is the diversity of the human cohorts studied, as well as both primate and mouse models. While the basic concept at the heart of the study is a relatively simple one, I'm afraid its impact will be lessened by a very dense and acronym heavy manuscript that makes it challenging to read. This would be a shame as this simple measure could be assessed in many different studies and contexts with strong potential to help clinical strategies and patient stratification approaches.

Response #1

We have addressed the concern of the reviewer and made great efforts to reduce the number of acronyms in the manuscript and have kept them to the minimum. In **Fig. 1b**

we have listed the main acronyms which we hope this will serve as a guidepost for the reader.

Comment #2

A big assumption is made that aging equals increased antigenic stimulation. This is a massive oversimplification of aging, which for effects on T cells alone will be driven by multiple factors such as thymic involution, decline in stem cell production, and tissue migration and residency of memory cells, etc. This assumption should be corrected and modified.

Response #2

We completely agree with the reviewer that thymic involution of T-cells and other factors could contribute to the IHG distribution pattern we observe with age. We think the confusion came from our use of the term “independent of age”. We used that term from a statistical perspective and not from a mechanistic perspective. Thus, when we observed an association in multivariate models corrected for age, we used the statistical terminology – “independent of age”. This clearly can be a potential source of confusion. For this reason, we have changed the term from “independent of age” to “controlled for age”.

However, we would like to respectfully point out that the distribution of IHG-II/III/IV we observe in younger people could be due to antigenic stimulation experienced and the distribution of IHG-II we see in older people could be a combination of thymic involution (as explained in the next comment) and antigenic stimulation, whereas the distribution of IHG-III/IV in older people could be mainly due to antigenic stimulation experienced. Extensive discussion on the role and difference in distribution of IHG-II (CD8-CD4 equilibrium) and IHG-III/IV (CD8-CD4 disequilibrium) with age and varied grades of antigenic stimulation is presented in **Supplementary Notes 1, 3, and 6**.

We have made the following changes to clarify these points.

1. We have **modified the introduction** to place our work in a broader evolutionary context apart from age. Please see the first paragraph of the revised introduction.
2. We have clarified what we mean by immunologic resilience (IR) in the second paragraph of the introduction along with a new Fig. **1a**. In this figure panel, we **clearly describe how multiple cycles of antigenic stimulation may degrade IR during the aging process**. In the discussion section, we allude to how viral infections may **lead to residual deficits in IR**.
3. We have modified the introduction to state:
“Hence, at any cross-sectional moment in a person’s life, impairments in immune status may be attributable to the combination of two distinct processes. The first relates to antecedent time (age)-dependent processes. Thus, age is a proxy, albeit imperfect, for the extent of antecedent immune deficits. The second relates to immune allostasis, a time-independent process, defined as the instantaneous or ongoing response to antigenic exposures.”
4. We have revised the limitations section to state:

We acknowledge that, in addition to antigenic experience, the changes in IHG distributions observed during aging (**Fig. 2f, 4d-e**) might be driven by multiple factors such as thymic involution, decline in stem cell production, as well as tissue migration and residency of memory cells. Possible confounders regarding the generation of the IHGs and their distribution patterns in varied settings of increased antigenic stimulation are discussed (**Supplementary Notes 1, 2, 3 and 6**). While we focused on the association between antigenic stimulation and shifts in IHGs during aging, it is possible that psychosocial stressors may also contribute, as social stressors may associate with age-related T lymphocyte percentages in older adults.

Comment #3

The previously reported studies that CD8 T cells decline at faster rates than CD4 T cells with increasing age could simply explain the age effects seen in this study (see for example Vrisekoop, N. et al. Sparse production but preferential incorporation of recently produced naive T cells in the human peripheral pool. Proc. Natl Acad. Sci. USA 105, 6115–6120 2008 or Patin E et al., Natural variation in the parameters of innate immune cells is preferentially driven by genetic factors. Nat Immunol. 2018 Mar;19(3):302-314. doi: 10.1038/s41590-018-0049-7.)

Response #4

Thank you for these comments. This comment was made by another reviewer as well in the original review. We had responded with extensive details that are provided in **Supplementary Note 6**. In **Supplementary Note 6** we show that there is a proportionally greater decline in CD8+ T-cells than CD4+ T-cells in younger and older persons with IHG-I and IHG-II; however, the level of decline is similar in older persons with IHG-I and IHG-II (-28.7% and -23.5%, respectively; Fig. in **Supplementary Note 6**, panel a). Thus, the proportionate decline in CD8+ T-cells in older persons with IHG-I and IHG-II cannot explain the fact that with age there is an increase in the prevalence of IHG-II, with a reciprocal decline in the prevalence of IHG-I. Also, please note that we did not see such a decline in CD8+ T-cells in older individuals with IHG-III and IHG-IV. Thus, we believe that the changes in distributions in IHGs are not attributable to the faster decline in CD8+ T-cells. A shift from IHG-I to IHG-II reflects CD4+ lymphopenia while CD8+ T-cell counts are restrained, whereas a shift from IHG-I to IHG-IV reflects CD4+ lymphopenia with unrestrained expansion of CD8+ T-cells. In addition, the median values for IHG-II and IHG-IV grades for different HIV- cohorts reported in **Supplementary Table 4** support the above statement. We refer the reviewer to **Supplementary Note 6** for additional details. We refer the reviewer to **Supplementary Note 6** for additional details.

Comment #4

What was the justification for not correcting p values for the multiple tests performed? Given the amount of comparisons and tests performed there is a high risk of false discovery and these should be controlled for, especially as many of the tests are essentially validation of the initial hypotheses.

Response #4

This comment was provided to us by the editor in August, and we provided an extensive response which is enclosed below at the end of this document.

Changes made:

We incorporated the response to this comment in Section 2.2-2.4 of the revised Supplementary Methods and the panel-by-panel statistical explanation. We have further clarified the statistical tests performed and indicated whether there was a correction performed or correcting for multiple testing (using the Bonferroni method for <20 comparisons) does not alter the results. These points were noted in my email response on Aug. 9th.

Comment #5

Why were Pearson correlations used which is more suited to parametric data sets, when the rest of the tests applied are more suited for non-parametric tests?

Response #5

This comment had been provided to us by the editor on August 5th, and we had responded on August 8th as follows. These points are included in Section 11.1 of the Supplementary Methods as well.

1. Pearson's and Spearman's correlation coefficient were used appropriately depending on the assumption of normality in the data evaluated.
2. For correlations of BAS vs. STI scores, Spearman's correlation coefficient was used.
3. For correlation of genes with immunologic correlates, since log₂ transformed gene expression and log₂ transformed functional (immunologic) markers were used and as they all fit normal distribution, Pearson's method was used to estimate the correlation coefficients.
4. The gene scores that are linear combination of Z-transformed log₂ normalized gene expression are well approximated by normal distribution and therefore Pearson's correlation coefficient was used.

Of note, for the association of gene scores with outcomes, linear regression (linear model) was used to test them, instead of non-parametric tests as highlighted below in the panel-by-panel detailed statistical methods for each of the figures. For median-based SAS-1/MAS-1 profile distribution analysis, non-parametric tests were used as appropriate.

Comment #6

The SAS and IR markers are likely to be highly dependent on each other, for example IL7R and CCR7 are expressed CD4 & CD8 T cells. How did the authors account for this?

Response #6

Thank you for this comment. Please note, we arrived at these markers through completely independent approaches. We did not have any preconceived bias of dependency or associations. The positive association of SAS-1 with IHG-I and negative association of MAS-1 with IHG-I were made after their independent discoveries as noted in the main

text. These associations further support our approach of creating the IHGs and the transcriptomic proxies. These associations between the IR metrics explain the parallels of IHG distributions with age/sex in **Fig. 2f** and SAS-1/MAS-1 profiles distribution with age/sex in **Fig. 7b-c**. Thus, IHG-I may be enriched for SAS-1 and therefore tracking similar mechanistic pathways linked to IL-7R and CCR7, as suggested by our findings presented in **Figure 9c,d**.

Comment #7

Given how much this study has benefited from other published studies, the newly generated data sets should either be deposited in repositories or made available as supplemental data files, so that other research teams can re-use them.

Response #7

Yes, data and code would be made available for each access as mentioned in the data and code availability statement of the manuscript.

Data availability

The datasets generated during and/or analyzed in the current study are available from the corresponding author on reasonable request.

Code availability

Modified version of GEO2R script sourced from NCBI GEO was used for analyses of GEO datasets and custom script based on ArrayExpress R package was used for analyses of ArrayExpress datasets. These scripts are available from the corresponding author on reasonable request.

Alternatively I'm satisfied with the reviewers response, they have clearly outlined their analytical strategy which makes sense and is justified. This manuscript is a huge amount of work and I think will contribute to the field.

Response to statistical comments of reviewer #3 that we had submitted on August 8th – these comments have been incorporated into the supplement

Overview

1. We thank the reviewer for providing us the opportunity to further clarify the rigor of our statistical approach/design and findings.
2. Our statistical approach adhered to the best practices for observational studies and reported according to the Strengthening the Reporting of Observational studies in Epidemiology (STROBE) guidelines.
3. *In each of our two submissions, we had provided a detailed statistical section in the Supplementary methods in which we had outlined in detail the statistical tests applied to each of the panels in figures presented in the main text and supplement. The supplementary methods included a description where corrections for multiple comparisons (FDR and Bonferroni) had been made and the basis of the P values reported.*

4. To address the reviewer's comments, we have modified the detailed statistical section in the Supplementary methods by further expanding on the statistical approach and testing applied to each panel of the main and supplementary figures, including multiple comparisons. **All our results/findings remain unchanged.**
5. For full transparency, if permitted by the editors, we will include the points noted below and the revised supplementary document as part of our manuscript.

Concern #1: What was the justification for not correcting p values for the multiple tests performed? Given the amount of comparisons and tests performed there is a high risk of false discovery and these should be controlled for, especially as many of the tests are essentially validation of the initial hypotheses.

Response. Below are the principles we applied to our statistical design that addresses this concern of the reviewer.

A1. Overarching hypothesis, cohorts, and comparisons.

Overarching hypothesis

Our hypothesis is based on the concept that in response to environmental stressors, including infections, individuals may manifest "resilience" (ability to recover after deviation from an optimal level of immunity) and/or "robustness" (ability to resist such deviation). Without intensive prospective monitoring, it is challenging to distinguish between resilience vs. robustness. We therefore combined these two concepts under the umbrella of the definition of immunologic resilience (**IR**). We previously defined optimal IR as the capacity to preserve and/or restore immunocompetence and suppress inflammation during acute, chronic, or repetitive antigenic exposures experienced across the lifespan, including SARS-CoV-2 infection. Using laboratory (immune health grades; IHGs) and transcriptomic metrics (SAS-1, MAS-1), we established a benchmark of optimal IR. We tested the overarching hypothesis that deviations from optimal IR predicted inferior immunity-dependent health outcomes in settings of acute (SARS-CoV-2) and chronic (HIV) infections, as well as settings of repetitive antigenic stimulation, such as sex work and aging.

Rationale for studying distinct cohorts and the comparisons made

1. While we articulated an omnibus or overarching hypothesis, this hypothesis has distinct facets depending on the clinical or biological context.
2. We tested specific hypotheses relevant to IR within distinct clinical or biological contexts. Each of these contexts has no association with any context other than environmental/antigenic stress.
3. It was not possible to study a single cohort or group of individuals to test whether deviations from optimal IR influence each of the outcomes (e.g., HIV acquisition, COVID-19 severity, lifespan, AIDS).

4. For these reasons, we examined IR metrics in distinct cohorts that permitted us to address specific IR-related questions that are aligned with the overarching hypothesis.
5. The impetus to evaluate multiple cohorts depicting the same issue (e.g. erosion of IR during aging; resistance to IR erosion associates with influenza resistance) was for purposes of replication and to define inter-cohort variability.
6. Although there may be conceptual overlaps, each cohort provided a unique biological context. While the broad principles of IR may apply to sex workers, HIV, influenza, and COVID-19, the biological underpinnings by which deviations from optimal IR may associate with inferior immunity-dependent outcomes are distinct. For example,
 - erosion of IR attributable to HIV risk factor-associated antigenic stimulation results in an increase in the number of sex workers with IHG-III and IHG-IV. This may be attributable to the high rate of CMV seropositivity in the cohort.
 - erosion of IR during COVID-19 results an increase in the proportion of patients with IHG-II vs. IHG-IV. IHG-III is infrequent in COVID-19 or aging.
 - during COVID-19, IHG-II emerges more frequently in patients with CMV seronegativity, whereas IHG-IV emerges more frequently in patients with CMV seropositivity.
 - for these reasons, to mitigate the confounding effects of CMV serostatus, where appropriate we performed comparisons in CMV seropositive vs. seronegative persons.
7. We performed comparisons to account for the effects of age and sex, as antigenic stimulation during aging is associated with erosion of IR, and females preserve optimal IR to a greater extent than men. In our analyses we (i) adjusted for age and/or sex, or (ii) performed overall comparisons, followed by stratification by age strata and sex.
8. Additionally, we could not apply the same IR metric or strata of a metric across each of the cohorts. For some analyses, we used IHGs and in others we used transcriptomic metrics of IR (SAS-1/MAS-1 profiles). An important aspect of this study was to validate SAS-1/MAS-1 profiles as transcriptomic proxies for IHGs. This was important, as while gene expression studies are commonplace, assessments of CD4+ and CD8+ T-cell counts required for derivations of IHGs are uncommon in most cohort-based studies.
9. Additionally, we define mechanisms by which IR status influences immunity-dependent outcomes. Each mechanistic cohort lends itself to distinct questions and analyses.
10. **SUMMARY:** For the above-noted reasons, it is not possible to apply an omnibus statistical test *across* these varied cohorts. Each cohort was distinct, and each comparison addressed a specific question using the best statistical practices and accounted for multiple comparisons and relevant variables that may influence results (e.g., age, sex, CMV serostatus).

A2. Biological plausibility

A key goal of our study was to examine the biological plausibility of our overarching hypothesis wherein the general principles of IR are applicable to distinct

biological/clinical contexts studied in distinct cohorts. Thus, it was to be anticipated that IR metrics will show statistically significant differences in varied cohorts.

Statistical approach:

1. The focus of our statistical design was to determine *whether the directions and magnitudes of differences (perhaps including some trending with $P > 0.05$ and < 0.10) fit a biologically coherent pattern vs. chance findings.*
2. If only one or very few measures reached statistical significance and their directions and/or magnitudes did not coherently fit with our hypothesis that IR was associated with superior immunity-dependent outcomes, our plan was to note that result(s) significant at $P < 0.05$ lacked biological plausibility and could be due to chance despite meeting the conventional cutoff for statistical significance.
3. **With recognition of points 1 and 2, our approach to multiple comparisons adheres to best practices. Depending on the question asked and nature of the dataset, we provide** (i) an overall omnibus test (e.g., ANOVA), (ii) FDR or Bonferroni corrections, and (iii) nominal P values, without adjustment for multiple testing. Such adjustment would be focused on avoidance of one or more results with $P < 0.05$ in the case where all differences are truly zero¹⁻³, which is an extremely unrealistic possibility about the association between IR status and immunity-dependent outcomes. In addition, adjustment would require that each result detract from the others, but there are clear biological relationships among many of the issues that we examine, and these permit coherent sets of findings to reinforce each other rather than detract from one another. Thus, multiple comparison adjustment would be an incorrect approach in such cases⁴. Hence, when reporting nominal P values, we relied on scientific judgment regarding the biological plausibility of our findings in the context of our overall findings, rather than formal adjustment methods to indicate where caution is warranted despite findings with significance at $P < 0.05$. Most importantly, in all cases where we reported nominal P values, it is self-evident from the smaller P values ($P < 0.001$) that formal adjustments for multiple comparisons would not change the interpretation of our findings.
4. **Thus, we individualized the analysis of each dataset, adhering to these best practices:**
 - a) We based interpretations on a synthesis of statistic results with scientific considerations.
 - b) We relied on scientific considerations to guard against overinterpretation of findings with $P < 0.05$.
 - c) We acknowledged the desirability of independent replications, particularly for unexpected findings (e.g., IR response was associated with symptom responsiveness to viral infections).
 - d) Where appropriate we determined group level differences followed by post-hoc testing.
 - e) The assumptions of each statistical test were evaluated to ensure that associated proposed analyses were appropriate (e.g., normality, homogeneity, linearity, and independence).
 - f) We provide estimates with confidence intervals

- g) **We chose accuracy, scientific judgement, and biological plausibility balanced by appropriate statistical testing for multiple comparisons.** We are mindful that conventions for statistical analysis and interpretation have emerged from a formal statistical hypothesis testing paradigm, guarding against chance false-positive results by application of multiple comparisons adjustments. However, these paradigms are inherently focused only on *P* values, promoting use of the *P* value fallacy. These adjustments also have the unfortunate property that the results of each analysis are automatically assumed to detract from all the others, with no consideration of how well the different results fit together conceptually or scientifically^{3,4}. This general approach has been criticized as unreliable and contrary to the original statistical theories that supposedly support it⁵⁻⁹, but it remains engrained in research culture^{10,11}.
- h) **Taken together, our overall statistical approach balances the considerations in points a) to g).**

Reviewer concern #2: Why were Pearson correlations used which is more suited to parametric data sets, when the rest of the tests applied are more suited for non-parametric tests

Response to concern #2: Thank you for allowing us to clarify the tests we used.

5. We use Pearson's and Spearman's correlation coefficient, depending on the assumption of normality.
6. For correlations of BAS vs. STI scores, Spearman's correlation coefficient is used.
7. For correlation of genes with immunologic correlates, we used log₂ transformed gene expression and log₂ transformed functional (immunologic) markers and they all fit normal distribution.
8. The gene scores that are linear combination of z-transformed log₂-normalized gene expression are well approximated by normal distribution and therefore use of the Pearson's correlation coefficient is appropriate. In addition, for these types of gene scores, we used linear regression (linear model) to test them, instead of non-parametric tests as highlighted below in the panel-by-panel detailed statistical methods for each of the figures. Note: for median-based SAS-1/MAS-1 profile distribution analysis, we used non-parametric tests as appropriate.

Revision of our original detailed statistical method in the supplementary document that had provided a per-panel description of each statistical test.

We have revised the supplementary document in which we had outlined the statistical treatment of every panel of the main and supplementary figures. **All major revisions are in a yellow highlighted** where we further clarify the statistical tests performed and indicate if there was a correction performed or correcting for multiple testing (using the Bonferroni method for <20 comparisons) does not alter the results.

MAIN FIGURES

- Figure 1
 - Schema. No statistical analysis was performed.
- Figure 2
 - *Panels a-c:*
 - Schema. No statistical analysis was performed.
 - *Panel d:*
 - Data derived from COVID-19 cohort and FHS offspring cohort
 - Table of results from Cox proportional hazards models evaluating the association of 90-day mortality in COVID-19 cohort (left) or all-cause mortality (9-year) in FHS offspring cohort (right) (outcomes) with selected transcriptomic signatures (SAS-1, MAS-1, and IMM-AGE scores; predictors) adjusted by age and sex. FHS follow-up time median: 8.163 years, IQR: 7.483 – 8.841, min: 0.178, max: 9.815. Adjusted hazards ratios (aHR), 95% confidence intervals (CI), and nominal *P* values are shown for each model.
 - SAS-1 and MAS-1 were identified by examining 52 gene signature scores in the acute COVID-19 and FHS cohorts using an FDR correction in both cohorts. In this report, we focused on the 10 signatures (3 SASs and 7 MASs) that associated with survival in both cohorts, after correcting for FDR, as reported in Table 11. Based on the AIC values, SAS-1 and MAS-1 were selected as being most prognostic. The associations of the IMM-AGE score with lifespan in the FHS cohort has been reported previously and used herein as a control and hence no additional corrections for multiple comparisons were warranted.
 - *Panel e:*
 - Schema. No statistical analysis was performed.
 - *Panel f:*
 - Data derived from SardiNIA cohort
 - *Left:* Distribution of IHGs by age strata. *P* value determined by χ^2 test.
 - *Right:* Distribution of IHGs by sex and age strata. *P* values determined by χ^2 test.
 - Two *P* values presented in this panel. We did not correct for multiple comparisons, as the right panel is an expansion of the left panel by sex. Additionally, since both *P* values were <0.001 , a Bonferroni corrected *P* values would remain significant.
 - *Panel g:*
 - Data derived from SardiNIA cohort
 - Odds with 95% CI bands of indicated IHG comparisons by sex across age. *P* values determined by logistic regression.
 - There are four separate hypotheses (different sample groups) being tested, each with two *P* values. We did not apply a correction for multiple comparisons for the two *P* values within each hypothesis. We also did not apply a correction for multiple comparisons across the four plots, as they depict results for separate hypotheses. Moreover, since seven of the eight significant *P* values were <0.001 , a Bonferroni corrected *P* value would remain significant.
 - *Panel h:*

- Schema. No statistical analysis was performed.
- Figure 3
 - *Panel a*:
 - Data derived from cohort of Kenyan HIV- children with Schistosomiasis
 - Distribution of IHGs (outcome) by egg counts (predictor). *P* value determined by Fisher's exact test.
 - One *P* value; no correction required.
 - *Panel b*:
 - Data derived from COVID-19 cohort.
 - Paired Distribution of IHGs (outcome) at baseline and during convalescence (predictors) among 220 COVID-19 patients in the overall group (left) and younger (middle; < 60 years) patients by overall and CMV serostatus, and older (right; ≥ 60 years) patients by overall and CMV serostatus. *P* values determined using a logsitic GEE model with an exchangeable correlation structure for evaluating if the odds of IHG-I, II, or IV differ by time (baseline vs. convalescence).
 - We examined the changes in IHG distributions in the same patients evaluated at baseline vs. convalescence. Since the overall changes between baseline and convalescence were highly significant ($P < 0.001$), we performed post-hoc analyses to determine the changes according to IHG status (IHG-I, IHG-II, and IHG-IV), reporting three *P* values in the overall group, the younger age group, and the older age group. Since the *P* values within each analysis were highly significant, a Bonferroni corrected *P* value would remain significant. We did not compare whether the changes in IHGs between younger vs. older were different. For these reasons, additional tests for multiple comparisons were not warranted.
 - *Panel c*:
 - Schema. No statistical analysis was performed.
 - *Panel d*:
 - Data derived from COVID-19 cohort.
 - Distribution of IHGs (outcome) by CMV serostatus (predictor) in overall, younger (Y, age < 60 years), and older (O, age ≥ 60 years). *P* values determined by Fisher's exact test for overall differences or IHG-II vs. rest or IHG-IV vs. rest in the indicated comparisons
 - Here, we first determined whether there were differences in IHG distributions by CMV serostatus in the overall cohort (All). Since the differences by CMV serostatus were significant ($P < 0.001$) in the overall group, we performed a post-hoc analysis to determine whether similar differences were observed in the younger and older group, and whether these differences were attributable to changes in IHG-II and/or IHG-IV. Since the *P* values within each analysis were highly significant, a Bonferroni corrected *P* value would remain significant. Additionally, we did not compare whether the changes between younger vs. older were different. For these reasons, additional tests for multiple comparisons were not warranted.
 - *Panel e*:
 - Data derived from COVID-19 cohort.

- Distribution of IHG (outcome) by age strata (predictor). $P_{\text{IHG-I}}$ and $P_{\text{IHG-IV}}$ is for the change in proportion of IHG-I vs. rest and IHG-IV vs. rest across age strata. P values determined by Fisher's exact test.
- There are two P values presented in this panel, each addressing a specific hypothesis, i.e., whether akin to what is observed in individuals without COVID-19, the prevalence of IHG-I decreases whereas the prevalence of IHG-IV increases with age in patients with COVID-19. Since, we are testing two separate hypotheses, a test for multiple comparisons were not warranted. Moreover, since comparisons between IHG-I and IHG-IV were not made, additional testing for multiple comparisons was not warranted.
- *Panel f:*
 - Data derived from COVID-19 cohort.
 - Distribution of IHGs (outcome) at presentation in the overall COVID-19 cohort, by sex, as well as hospitalization, survivor status, and age (predictors). P values determined by Fisher's exact test.
 - Three separate hypotheses (differences in IHGs by sex, hospitalization status, survivorship), were tested with one P value for each comparison. All results are reported, and additional testing for multiple comparisons was not warranted.
- *Panel g:*
 - Data derived from RTR cohort and SLE cohort
 - *Left:* Distribution of IHGs (outcome) in 114 renal transplant recipients (RTR) by sex (predictor). P values determined by χ^2 test.
 - *Right:* Distribution of IHGs (outcome) in 53 patients with SLE (source: GSE49454). P values determined by χ^2 test.
 - One P value per cohort; no correction required.
- *Panel h:*
 - Data derived from HIV+ PIC cohort
 - *Left:* Distribution of IHGs (outcome) by entry viral load (VL) strata (predictor) in 592 participants from the PIC cohort. P values determined by χ^2 test.
 - *Right:* Distribution of IHGs (outcome) at baseline and during ART treatment course (predictors) for a subset of PIC cohort ($n=75$). P values determined by linear GEE model based on a binomial distribution with an AR1 correlation structure comparing the change in IHG-I vs. rest and the change in IHG-IV vs. rest over time.
 - Analysis was performed in two separate subsets of the same cohort testing different hypotheses. Left, examines the overall differences in IHG distributions in therapy-naïve patients categorized according to HIV VL strata; an overall difference is reported, and additional testing for multiple comparison was not warranted. Right, tests two separate hypotheses, i.e., whether suppression of HIV-VL was associated with reconstitution of IHG-I and a decrease in the prevalence of IHG-I. Additional testing for multiple comparisons was not warranted.
- *Panel i:*
 - Schema. No statistical analysis was performed.
- *Panel j:*
 - Data derived from subset of HIV+ EIC participants who are therapy-naïve
 - Distribution of IHGs (outcome) at entry and years since entry (predictor) who remained therapy-naïve. P values determined by χ^2 test.

- One *P* value; no correction required.
 - *Panel k*:
 - Data derived from HIV- FSW cohort.
 - *Top*: Distribution of IHGs (outcome) by the indicated factors (predictor). *P* values determined by χ^2 test.
 - *Bottom*: Percentage of FSW who had later HIV seroconversion (outcome) according to study groups (predictor) in the top panel. *P* values determined by logistic regression.
 - Groupings in top and bottom panels are by following predictors: duration of sex work (in years) strata, condom use, clients per week strata, clients per week minus condoms per week strata, behavioral activity score (BAS), direct STI score, indirect STI score, and total STI score.
 - Distinct hypotheses were tested, i.e., whether the extent of distinct correlates of sex work associated with changes in IHG status. Each comparison has one *P* value. All results are reported, and additional testing for multiple comparisons was not warranted.
- Figure 4
 - *Panel a*:
 - Data derived from HIV- FSW cohort using the first available CD4+ and CD8+ T-cell counts, CD4:CD8 ratio, and BAS during each 2 years intervals.
 - *Left*: Distribution of IHGs (outcomes) within each 2-year intervals (predictor) in 27 FSWs who remained HIV- for at least 10 years. *P* value determined by Fisher's exact test comparing window $>0-\leq 2$ vs. $>8-\leq 10$.
 - *Right*: Boxplots of BAS (outcome) in each 2-year intervals (predictor). *P* value determined by linear GEE model with an AR1 correlation structure evaluating the change in BAS over indicated time windows.
 - Two different hypotheses were tested; left and right, depict whether a decrease in sex work associated with a reconstitution in IHG-I and BAS, respectively. One *P* value per hypothesis is reported; additional testing for multiple comparisons was not warranted.
 - *Panel b*:
 - Data derived from HIV- FSW cohort
 - *Left*: Distribution of IHGs (outcome) at baseline and first available IHG at or after year 4 of follow-up (predictors) in FSW who remained HIV-seronegative for at least 4 years ($n=101$); IHG distribution (outcome) is depicted in the overall group of FSWs and by baseline IHG (predictor). The outcomes are the IHG distributions with the predictors being baseline and first available measurement at year 4. *P* values determined by χ^2 test.
 - *Right*: Boxplots of BAS (outcomes) in 101 FSWs who remained HIV-seronegative for at least 4 years (predictor); data are for values at baseline and first available measurement on or after the fourth anniversary (predictors). *P* values determined by Wilcoxon signed-rank test.
 - Two different hypotheses similar those shown in *panel a* are tested. One *P* value per hypothesis is reported; additional testing for multiple comparisons was not warranted.

- *Panel c:*
 - Data derived from HIV- FSW cohort
 - *Left:* Distribution of IHGs (outcome) pre- and post- HIV seroconversion (predictor) in 43 FSWs (paired samples; predictor). *P* value determined by Fisher's exact test.
 - *Right:* Distribution of IHGs (outcome) in Sooty mangabeys stratified by SIV serostatus (non-paired samples; predictor). *P* values determined by χ^2 test.
 - Two different hypotheses are tested; overall differences in humans (left) and Sooty mangabeys (right) by lentiviral serostatus are tested. One *P* value per hypothesis is reported; additional testing for multiple comparisons was not warranted.
- *Panel d:*
 - Data derived from SIV-/SIV+ sooty mangabeys
 - Distribution of IHGs (outcome) in Sooty mangabeys stratified by SIV serostatus, sex and age strata (predictors). *P* values determined by Fisher's exact test.
 - Two different hypotheses were evaluated, examining differences in IHG distributions in SIV- and SIV+ animals by sex and age. One *P* value per hypothesis is reported. Differences between SIV- and SIV+ animals were not examined. Additional testing for multiple comparisons was not warranted.
- *Panel e:*
 - Data derived from SIV- Chinese rhesus macaques
 - Distribution of IHGs (outcome) in SIV- Chinese Rhesus macaques overall, by sex, and age strata (predictors). *P* values determined by Fisher's exact test.
 - Two different hypotheses akin to *panel d* were tested. Additional testing for multiple comparisons was not warranted.
- *Panel f:*
 - Data derived from CC-RIX mice cohort
 - Distribution of IHGs (outcome) by indicated infectious outcome groups (predictor) in uninfected counterparts of CC-RIX mice infected with Ebola. *P* value determined by Fisher's exact test comparing mice groups lethal vs. resistant to Ebola infection.
 - One *P* value; no correction required.
- *Panel g:*
 - Data derived from COVID-19 cohort.
 - *Top:* Schema. No statistical analysis was performed.
 - *Middle:* Forest plots depicting the odds ratio (OR) with 95% CI of hospitalization (outcome) from two separate logistic regression models both adjusted by age strata: Hospitalization by IHG and hospitalization by CMV serostatus (predictors).
 - *Bottom:* Plots depicting the hazard ratio (HR) with 95% CI of 30-day mortality (outcome) from two separate Cox proportional hazards models both adjusted by age strata: Mortality by IHG and mortality by CMV serostatus (predictors).
 - Point estimates and confidence intervals are reported, with no comparisons between the study groups. Additional testing for multiple comparisons was not warranted.
- *Panel h:*
 - Data derived from RTRs cohort.
 - KM plot depicting time to second cutaneous squamous cell carcinoma (CSCC; outcome) in a group of 40 RTRs with CSCC in the preceding year (as detailed

previously¹²) and was re-analyzed for CSCC-free (i.e. time to second CSCC during study follow-up) by IHG (predictor). *P* value determined by log-rank test.

▪ One *P* value; no correction required.

○ *Panel i*:

▪ Data derived from HIV+ EIC participants

▪ KM plot depicting time to AIDS (CDC 1993 criteria; outcome) and the median entry CD4+ T-cell counts, CD8+ T-cell counts, CD4:CD8 ratios, and log₁₀ VL (outcomes) by entry IHG (predictors). *P* value for KM plot were determined by log-rank test. *P* values for difference in HIV-VL from IHG-I to other IHGs were determined by linear regression.

▪ Two separate hypotheses are tested, reporting the first with one *P* value, and the second with three *P* values from a single model testing whether the VL in patients presenting with a non-IHG-I grade differs from that of IHG-I. All *P* values in the latter model were <0.001 and remain significant after Bonferroni correction.

○ *Panel j*:

▪ Data derived from HIV+ PIC cohort.

▪ Entry log₁₀ VL levels (outcome) by entry IHG status (predictor). *P* values determined by Wilcoxon rank-sum test.

▪ Three *P* values are reported; the two significant *P* values were <0.001 and remain significant after Bonferroni correction.

○ *Panel k*:

▪ Data derived from HIV+ EIC cohort.

▪ Log₁₀ VL levels (outcome) by entry IHG-I vs. IHG-II/III/IV (rest) of the cohort, and among participants with entry IHG-I stratified by those preserving IHG-I vs. switching from IHG-I to another IHG during each year therapy-naïve follow-up (predictors) The data illustrate the first IHG with an available coincident VL measurement during the indicated time window. *P* value determined by Wilcoxon rank-sum test.

▪ A single hypothesis across six different timepoints was tested to determine whether VL differed between individuals who preserved IHG-I vs. had the other grades. A consistent pattern was observed across timepoints, and additional testing for multiple comparisons was not warranted.

• Figure 5

○ *Panel a*:

▪ Data derived from HIV- FSW cohort

▪ Distribution of IHGs (outcome) in 449 FSWs by their behavioral activity score (BAS) categorized as less than 0, equal to 0, or greater than 0 (predictor) and according to whether they did vs. did not become HIV+ later (predictor). *P* value determined by χ^2 test.

▪ Two different hypotheses with one *P* value each were examined; additional testing for multiple comparisons was not warranted.

○ *Panel b*:

▪ Data derived from HIV- FSW cohort

- Logistic regression was used to determine the OR with 95% CI of having IHG-III or IHG-IV at baseline and/or future HIV seroconversion (outcomes) by BAS, total STI scores, and IHGs (predictors). Three separate models were used.
 - Coefficients with 95% CI separate logistic regression models are depicted. No *P* values are shown. Additional testing for multiple comparisons was not warranted.
 - *Panel c*:
 - Data derived from (i) Literature review for the proportion of CD8-CD4 disequilibrium (i.e., CD4:CD8 ratio <1.0 as a proxy for IHG-III or IHG-IV; outcome) as listed in **Supplementary Table 9**, and (ii) participants from HIV– SardiNIA, HIV– UCSD and HIV– FSWs as listed in Figure 1. Percentage of participants with IHG-III/IV (outcome) by indicated groups (predictor). All *P* values were determined by χ^2 test (except for Tetanus vaccine data where Fisher’s exact test was used) for comparing difference in proportion of IHG-III or IHG-IV within the specified study groups.
 - This was an exploratory survey performed to determine whether the indicated factors (e.g., age, sex, varied inducers, or outcomes) associated with differences in the proportions in individuals with vs. without IHG-III or IHG-IV. All results are reported. We did not compare for differences across different study groups. Additional testing for multiple comparisons was not warranted.
 - Schema. No statistical analysis was performed.
- Figure 6
 - *Panel a*:
 - Schema. No statistical analysis was performed.
 - *Panel b*:
 - Data derived from HIV– SardiNIA cohort
 - Distribution of IHG subgrades (outcome) by age strata (predictor). *P* value determined by Fisher’s exact test.
 - One *P* value; no correction required.
 - *Panel c*:
 - Data derived from COVID-19 cohort
 - Distribution of IHG subgrades (outcome) in overall (*n*=541) and at baseline and at convalescence in 220 COVID-19 patients (predictor). *P* value determined by Fisher’s exact test.
 - One *P* value; no correction required.
 - *Panel d*:
 - Data derived from multiple cohorts.
 - *Left-most*. Distribution of IHG subgrades (outcome) in SLE cohort (source: GSE49454). No statistical analysis was performed.
 - *Left*. Distribution of IHG subgrades (outcome) in 114 RTRs.
 - *Right*. Distribution of IHG subgrades (outcome) in HIV+ PIC cohort pre-ART and during ART (predictor). *P* value determined by Fisher’s exact test.
 - *Right-most*. Distribution of IHG subgrades (outcome) at entry by BAS<0 vs. ≥0 (predictor) in participants from HIV– FSW cohort. *P* value determined by Fisher’s exact test.

- One P value per comparison of IHG distributions across two cohorts, and no comparisons made between cohorts. Additional testing for multiple comparisons was not warranted.
- *Panel e*:
 - Schema. No statistical analysis was performed.
- Figure 7
 - *Panel a*:
 - Schema. No statistical analysis was performed.
 - *Panel b*:
 - All analyses were performed in the FHS offspring cohort (Source: dbGaP: phs000007.v30.p11; median age [IQR] years = 66 [60-73]; females: 54%) and all P values were determined by χ^2 test.
 - *Left*: Distribution of SAS-1/MAS-1 profiles (outcome) in participants stratified by sex (predictor). P values were determined for differences in proportions of the SAS-1/MAS-1 profiles and the differences in proportion of SAS-1^{high}-MAS-1^{low} or SAS-1^{low}-MAS-1^{high} vs. rest between males vs. females.
 - *Right*: Distribution of SAS-1/MAS-1 profiles (outcome) in participants stratified by indicated age bins (predictor). P values were determined for differences in proportions of the SAS-1/MAS-1 profiles and the differences in proportion of SAS-1^{high}-MAS-1^{low} or SAS-1^{low}-MAS-1^{high} vs. rest overall across the indicated age bins.
 - There are four distinct hypotheses being tested: differences in SAS-1/MAS-1 profiles by sex and by age, and whether age is associated with an increase in the SAS-1/MAS-1 profiles that associated with mortality and a decrease in the profile that is associated with survival. Additional testing for multiple comparisons was not warranted.
 - *Panel c*:
 - All analyses were performed in the San Antonio Family Heart Study (Source: E-TABM-305) and all P values were determined by χ^2 test.
 - *Left*: Distribution of SAS-1/MAS-1 profiles (outcome) in participants stratified by sex (predictor).
 - *Middle*: Distribution of SAS-1/MAS-1 profiles (outcome) in participants stratified by indicated age bins (predictor). P values were determined for differences in proportions of the SAS-1/MAS-1 profiles and the differences in proportion of SAS-1^{high}-MAS-1^{low} or SAS-1^{low}-MAS-1^{high} vs. rest overall across the indicated age bins.
 - *Right*: Distribution of SAS-1/MAS-1 profiles (outcome) in participants stratified by indicated age bins and sex (predictor).
 - This series of analyses were hypothesis driven and performed to corroborate findings shown in *panel a* in a larger dataset across a wider age range. All P values are reported, and the overall group differences by sex are highly significant. The far-right panel depicts whether group level differences in sex are evident across age strata. Additional testing for multiple comparisons was not warranted.
 - *Panel d*:
 - All analyses were performed in the Alzheimer's disease (AD) and other dementia disorders cohort

- Distribution of SAS-1/MAS-1 profiles (outcome) in first four groups: Pooled sets of control groups [set 1 ($n= 249$; median age [IQR] years = 73 [69 -78]; females: 56%; source: GSE140829) and set 2 ($n= 281$; median age [IQR] years = 72 [68-77]; females: 56%; source: GSE140830)] stratified by indicated age bins (predictor).
- Distribution of SAS-1/MAS-1 profiles (outcome) in last four groups: Pooled sets of disease groups [set 1 includes patients with AD and mild cognitive impairment ($n= 338$; median age [IQR] years = 72 [68-79]; females: 51%; source: GSE140829) and set 2 includes patients one or multiple disorders within the frontotemporal dementia ($n= 261$; median age [IQR] years = 66 [60-72]; females: 51%; source: GSE140830) were pooled together and stratified by indicated age bins (predictor).
- P values were determined using GEE model with an exchangeable correlation structure, binomial family, and ANOVA for comparing differences in proportions of SAS-1^{high}-MAS-1^{low} vs. rest across indicated all age bins within control or disease groups. P values determined using GEE model with an exchangeable correlation structure, and binomial family for comparing differences in proportions of SAS-1^{high}-MAS-1^{low} vs. rest between indicated age bins of control vs. disease groups.
- Specific hypotheses are tested within and across two distinct datasets differing by disease status. **Additional testing for multiple comparisons was not warranted.**
- *Panel e:*
 - All analyses were performed in the SLE dataset (GSE49454).
 - *Left to right:* Distribution of SAS-1/MAS-1 profiles (outcome) in healthy controls stratified by median age of the overall cohort (median age [IQR] years = 39 [30-51]; females: 85%) and SLE patients stratified by IHG subgrades (predictor). Note: Healthy controls and SLE patients in IHG-I were further stratified by median age of the overall cohort (Younger/Older). For this analysis, only first available laboratory (CD4+ count and CD4:CD8 ratio) measurements and corresponding gene expression measurement from patients with SLE was used.
 - P values were determined using Fisher's exact test for comparing differences in proportions of SAS-1/MAS-1 profiles between IHG-I (pooling younger and older) vs. IHG-II (pooling IHG-IIa/b/c subgrades) and IHG-I (pooling younger and older) vs. IHG-IV (pooling IHG-IVa/b/c subgrades).
 - **Two P values are shown to depict differences in SAS-1/MAS-1 profiles in patients with IHG-II or IHG-IV vs. IHG-I.**
- *Panel f:*
 - Data derived from COVID-19 cohort.
 - Distribution of SAS-1/MAS-1 profiles (outcome) in COVID-19 patients stratified by baseline IHG subgrades (predictor).
 - P values were determined using Fisher's exact test for comparing differences in proportions of SAS-1/MAS-1 profiles across IHG subgrades as well as between IHG-I vs. IHG-II (pooling IHG-IIa/b/c subgrades) and IHG-I vs. IHG-IV (pooling IHG-IVa/b/c subgrades).
 - **Group level differences in IHG status were significant, and two post hoc comparisons were performed to corroborate findings in *panel e* using an acute COVID-19 cohort. Additional testing for multiple comparisons was not warranted.**
- *Panel g:*
 - Data derived from HIV+ EIC cohort

- Distribution of SAS-1/MAS-1 profiles in participants with HIV on ART (EIC subset) stratified by IHG subgrades at the time of experiment and spontaneous virologic controllers (SVC) not on ART. Note: For this analysis, HIV– participants ($n=3$) and HIV+ participants not on ART ($n=3$) in the dataset were excluded from the plot.
 - P values were determined using Fisher's exact test for comparing differences in proportions of SAS-1/MAS-1 profiles between IHG-I vs. IHG-II (pooling IHG-IIa/b subgrades), IHG-I vs. IHG-IV (pooling IHG-Iva/b subgrades), and IHG-I vs. SVC.
 - Two comparisons performed to corroborate findings in *panel e* using an HIV cohort; additional testing for multiple comparisons was not warranted. A single comparison between IHG-I and SVCs was performed.
 - *Panel h*:
 - Data derived from COVID-19 cohort.
 - Distribution of SAS-1/MAS-1 profiles (outcome) in COVID-19 patients stratified by age, hospitalization and 30-day survival categories (predictor). P values determined by Fisher's exact test for indicated comparisons
 - Two separate hypotheses tested comparing differences in profiles by IHG status and COVID-19 outcomes. Single P value was reported for each comparison. Additional testing for multiple comparisons was not warranted.
 - *Panel i*:
 - Data derived from multiple cohorts.
 - *Left*, COVID-19: Distribution of SAS-1/MAS-1 profiles (outcome) in COVID-19 patients (source: COVID-19 cohort) stratified by baseline IHG status (predictor).
 - *Middle*, SLE: Distribution of SAS-1/MAS-1 profiles (outcome) in SLE patients (source: GSE49454) stratified by first IHG measurement (predictor).
 - *Right*, Therapy-naïve: Distribution of SAS-1/MAS-1 profiles (outcome) in 5 uninfected controls ($n=10$ samples), and HIV+ patients classified into three groups (predictors): 9 asymptomatic HIV+ patients ($n=18$ samples); 9 acute HIV+ patients ($n=16$ samples); and 4 AIDS patients ($n=8$ samples) (Source: GSE16363, median age [IQR] years = 39 [32-45]; females: 15%). Note: Two samples from each donor or patient were available in GEO and all samples were used in the analysis.
 - P values for first two panels were determined using Fisher's exact test for comparing proportions of SAS-1^{high}-MAS-1^{low} vs. rest between IHG-I vs. rest. P value for right panel was determined using GEE model with an exchangeable correlation structure, binomial family, and ANOVA for comparing differences in proportions of SAS-1^{high}-MAS-1^{low} vs. rest across indicated groups.
 - Single P value per comparison in three separate cohorts. Left and middle comparisons are corroborative analyses; comparison for the right panel is for change in IHG-I distributions. Additional testing for multiple comparisons was not warranted.
 - *Panel j*:
 - Schema. No statistical analysis was performed.
- Figure 8:
 - *Panel a*:
 - Data derived from FHS offspring cohort (Source: dbGaP: phs000007.v30.p11; median age [IQR] years = 66 [60-73]; females: 54%).

- KM plot depicting proportion survived (outcome) over 9-year follow-up time in FHS participants stratified by SAS-1/MAS-1 profiles (predictors). *P* value determined by log rank test.
- One *P* value; no correction required.
- *Panel b:*
 - Data derived from FHS offspring cohort (Source: dbGaP: phs000007.v30.p11; median age [IQR] years = 66 [60-73]; females: 54%)
 - Distribution of SAS-1/MAS-1 profiles (outcome) in FHS participants stratified by survival status (predictor). *P* values were determined using χ^2 test for differences in proportions of the SAS-1/MAS-1 profiles and the differences in proportion of SAS-1^{high}-MAS-1^{low} or SAS-1^{low}-MAS-1^{high} vs. rest between survivors vs. non-survivors.
 - Group level differences by survivorship in SAS-1/MAS-1 profiles were significant, followed by specific post-hoc testing whether there were differences in the extremes of the SAS-1/MAS-1 profiles by survivorship status.
- *Panel c:*
 - Schema. No statistical analysis was performed.
- *Panel d:*
 - Data derived from sepsis cohorts.
 - *Left:* Distribution of SAS-1/MAS-1 profiles [outcome] in the meta-analysis of patients with severe sepsis due to community acquired pneumonia (CAP) or fecal peritonitis (FP) admitted to the intensive care unit with either sepsis response signature 1 (SRS1, G1) or sepsis response signature 2 (SRS2, G2) [predictor] (Sources: E-MTAB-4421, median age [IQR] years = 64 [52-75], females: 46%; E-MTAB-4451, median age [IQR] years = 73.5 [63.5-80.0], females: 26%; E-MTAB-5273, n=147, median age [IQR] years = 66 [52.5-76.0], females: 50%; and E-MTAB-5274, n=106, median age [IQR] years = 71 [62.2-77.0], females: 36%) and healthy controls. Note: G1 (SRS1) is associated with higher mortality than G2 (SRS2); the SRS1 and SRS2 were defined using baseline samples at admission into the intensive care unit. All the four datasets used for this analysis were from same microarray platform.
 - *P* values were determined using Fisher's exact test comparing differences in proportions of SAS-1/MAS-1 profiles between G1 vs. G2 within patients with CAP and FP.
 - *Right:* Distribution of SAS-1/MAS-1 profiles [outcome] in patients with sepsis (n=43), post-surgical patients with septic shock who are non-survivors (septic shock NS; n=17), post-surgical patients with septic shock who are survivors (septic shock S; n=22), patients with systemic inflammatory response syndrome (SIRS; n=58) and healthy controls (normal; n=15) [predictor] (Source: E-MTAB-1548). Median age [IQR] years of patients (excluding the healthy controls) in the study = 72 [62-79]; females among patients (excluding the healthy controls): 28%.
 - *P* values were determined using χ^2 test for comparing differences in proportions of SAS-1/MAS-1 profiles across all groups as well as within disease groups and GEE model with an exchangeable correlation structure, binomial family, and ANOVA for comparing differences in proportions of SAS-1^{high}-MAS-1^{low} vs. rest or SAS-1^{low}-MAS-1^{high} vs. rest across all groups. Note: For determination of overall or within

disease groups P value using χ^2 test, the repeated measurements from the same individual were considered as unique. Since the proportion of SAS-1^{low}-MAS-1^{high} in controls was zero, the factor levels were reversed and NS within septic shock survivors group was used as reference to calculate the P value for SAS-1^{low}-MAS-1^{high} vs. rest.

- Sepsis #1. Analysis was performed in two separate cohorts with a single comparison per cohort; comparisons across cohorts were not performed. Additional testing for multiple comparisons was not warranted. Sepsis #2. In this cohort, an overall difference was observed in SAS-1/MAS-1 profiles; post-hoc testing with three comparisons within the sepsis group was performed to illustrate changes across the sepsis spectrum. Additional testing for multiple comparisons was not warranted.
- Panel e:
 - Data derived from natural viral infection cohort (GSE68310; age range: 18-49 years)
 - *Left*: Distribution of SAS-1/MAS-1 profiles [outcomes] in 133 adults at enrollment (baseline: B; Group 1) and followed longitudinally during a seasonal acute respiratory illness (days 0, 2, 4, 6, 21; Groups 2-6) and the following spring (SP; Group 7) the next year [predictor] (Source: GSE68310, females: 52%). P values were determined using GEE model with an exchangeable correlation structure and binomial family for comparing differences in proportions of SAS-1^{low}/MAS-1^{high} vs. rest between baseline vs. Day 0, and Day 0 vs. spring timepoints.
 - *Middle to Right*: Distribution of SAS-1/MAS-1 profiles (outcome) in participants presenting with a SAS-1^{high}/MAS-1^{low} (*middle*) and SAS-1^{low}/MAS-1^{high} (*right*) and followed longitudinally. P values were determined by Fisher's exact test for comparing distributions between the two SAS-1/MAS-1 profile groups (SAS-1^{high}/MAS-1^{low} vs. SAS-1^{low}/MAS-1^{high}) at the indicated timepoints.
 - *Left*, two hypotheses were tested: ARI causes a shift in SAS-1/MAS-1 profiles, and SAS-1/MAS-1 profiles are restored during convalescence. One P value per comparison, and additional testing for multiple comparisons was not warranted. *Middle and right*: Based on findings in the left panel, post-hoc we examined whether the changes in SAS-1/MAS-1 profiles during convalescence differed by the SAS-1/MAS-1 profile that antedated ARI. Four of the five significant P values from six comparisons remain significant after Bonferroni correction and do not meaningfully impact on interpretation (Table below).

Comparison at time	P value	P_{adj} (Bonferroni)
Day 0	0.4332 (ns)	1 (ns)
Day 2	0.0482 (*)	0.2893 (ns)
Day 4	0.0049 (**)	0.0296 (*)
Day 6	<0.001 (***)	<0.001 (***)
Day 21	<0.001 (***)	<0.001 (***)
Spring	0.0025 (**)	0.0151 (*)

- Panel f:
 - Schema. No statistical analysis was performed.
- Panel g:

- Data derived from viral challenge study (GSE17156).
- Distribution of SAS-1/MAS-1 profiles [outcome] in healthy volunteers (18-49 year old) inoculated experimentally with H3N2, RSV, or rhinovirus (infections pooled for analysis) at baseline (T1) and when peak symptoms (T2) developed after inoculation stratified by symptom status [predictor] (Source: GSE17156). *P* values were determined using GEE model with an exchangeable correlation structure and binomial family for comparing differences in proportions of SAS-1^{low}/MAS-1^{high} vs. rest between asymptomatics vs. symptomatics at baseline as well as peak time point.
- Comparisons are made in two separate groups with distinct phenotypes. Single comparison per group. Additional testing for multiple comparisons was not warranted.
- *Panel h*:
 - Data derived from viral challenge study (GSE52428).
 - Distribution of SAS-1/MAS-1 profiles [outcome] evaluated following intranasal inoculation with influenza A (H1N1 and H3N2, pooled for analysis) in healthy persons who became symptomatic at baseline (T1; -24 hr. and 0 hr. pooled) and peak symptoms (T2; 60 hr. and 69.5 hr. pooled) [predictor] (Source: GSE52428, median age [IQR] years = 25 [23-28]; females: 37%). Note: Timepoint 60 and 69.5 hours were selected as peak timepoint based on IFN response (data not shown) in symptomatic participants after which the IFN response achieved a plateau, and these two time points was the midpoint of the study. *P* values were determined using GEE model with an exchangeable correlation structure and binomial family for comparing differences in proportions of SAS-1^{low}/MAS-1^{high} vs. rest between asymptomatics vs. symptomatics at baseline as well as peak time point.
 - Comparisons are made in two separate groups with distinct phenotypes. Single comparison per group. Additional testing for multiple comparisons was not warranted.
- *Panel i*:
 - Data derived from hospitalized influenza study (source: GSE111368)
 - *Left*: Distribution of SAS-1/MAS-1 profiles [outcome] in healthy controls and patients hospitalized with severe influenza collected at T1 (recruitment), T2 (approx. 48hr after T1), and T3 (at least 4 weeks after T1) [predictor] (source: GSE111368; median age [IQR] years = 72 [67-76]; females: 54%). *P* values were determined by GEE model with an exchangeable correlation structure and binomial family for comparing differences in proportions of SAS-1^{high}/MAS-1^{low} vs. rest between HC and T1, and between T1 and T3.
 - *Middle to Right*: Distribution of SAS-1/MAS-1 profiles [outcome] in patients stratified by severity, time, and median age of 39: 18-39 year old (*middle*) and 40-71 year old (*right*) [predictor]. Descriptive analysis. No statistical analysis was performed.
 - Two separate hypotheses are tested requiring comparisons in profile distributions between health controls vs. acute infection, and changes in profiles at baseline (T1) vs. recovery (T3). Additional testing for multiple comparisons was not warranted.

- Figure 9:
 - *Panels a-b*:
 - Schema. No statistical analysis was performed.
 - *Panel c*:
 - Data derived from subset of HIV+ EIC cohort
 - Correlation (r ; Pearson correlation coefficient) [outcomes] between expression levels of representative genes within the SAS-1 or MAS-1 scores with levels of T-cell responsiveness (orange), T-cell dysfunction (green), and plasma IL-6 (blue) [predictors]. Correlation coefficient (r), and P values determined by correlation test using Pearson's method; and thresholds of r values at false discovery rate (FDR; Benjamini-Hochberg) <0.05 are shown in **Supplementary Table 12**. Measures of T-cell responsiveness, T-cell dysfunction and plasma IL-6 were from 55, 56 and 50 HIV+ persons, respectively.
 - A false discovery rate was used and is shown in Supplementary Table 12.
 - *Panel d*:
 - Data derived from SIV+ sooty mangabeys
 - Boxplots of the indicated immune traits (outcome) by IHG (predictor). P values determined by Wilcoxon rank-sum test.
 - Three different hypotheses are being tested here. The first is the %cell type differs by IHG-I and II vs. IHG-III and IV. The second is the %cell type differs by IHG-I vs. IHG-III. The third is the %cell type differs by IHG-II vs. IHG-IV. These comparisons are made for specific cell types and not across cell types. Additional testing for multiple comparisons was not warranted.
 - *Panel e*:
 - Data derived from SIV– Chinese Rhesus macaques
 - Boxplots of percent PD1+CD8+ cells (outcome) by age and IHG status (predictor). P values determined by Kruskal-Wallis test across the indicated groups.
 - Two different hypotheses are tested reporting one P value per comparison. Additional testing for multiple comparisons was not warranted.
- Figure 10:
 - *Panel a*:
 - Data derived from SardiNIA cohort
 - 75 Immune traits sub-grouped into 19 signatures that classify into four groups, as described in Supplementary Section 11.2.
 - *Panel b*:
 - Data derived from SardiNIA cohort
 - Boxplots of the indicated immune traits (outcome) by age (<40 and ≥ 70) within IHG-I and IHG-II and by IHGs (predictor). The trait levels are presented as inverse-normalized residuals after adjusting for sex and/or age using linear regression. P values determined by Wilcoxon rank-sum test and corrected for multiple comparisons using Bonferroni method. Detailed methods are as described in Supplementary Section 11.2.
 - *Panel c*:
 - Data derived from SardiNIA cohort

- Log₂ Levels of CD127-CD8^{bright} T-cells and CD28-CD8^{bright}T-cells (outcome) across age stratified by IHGs (predictors). Analysis was performed using linear regression with log₂ cell counts as the outcome and age and IHG as predictors. Fitted lines, 95% confidence band, and *P* values determined using linear regression and are corrected for multiple comparisons using FDR.
- *Panel d*:
 - Schema. No statistical analysis was performed.

SUPPLEMENTARY FIGURES

- Supplementary Fig. 1:
 - *Panels a-d*:
 - Schemas of Study cohorts. No statistical analysis was performed.
- Supplementary Fig. 2:
 - *Panel a*:
 - Data derived from COVID-19 cohort
 - Schema of gene score derivation and analysis. The methods are detailed in Supplementary Section 8.2. No statistical analysis was performed.
 - *Panel b*:
 - Data derived from FHS offspring cohort (Source: dbGaP: phs000007.v30.p11; median age [IQR] years = 66 [60-73]; females: 54%)
 - KM plots depicting proportion survived (outcome) over 9-year follow-up time in the FHS participants stratified by quartiles (first through fourth, Q1-Q4) of indicated gene signature scores (predictors). *P* values determined by log-rank test.
 - Three different hypotheses are tested with one *P* value per comparison. Additional testing for multiple comparisons was not warranted.
 - *Panel c*:
 - Data derived from cohort of Kenyan HIV- children with Schistosomiasis
 - Boxplots of percent CD25+CD127-CD4+ (outcome) by *S. haematobium* egg counts (predictor). *P* value determined by Kruskal-Wallis test.
 - One *P* value; no correction required.
- Supplementary Fig. 3:
 - *Panels a-d*:
 - All data derived from HIV- FSWs cohort
 - *Panel a*:
 - Schema of participant subsets. No statistical analysis was performed.
 - *Panel b*:
 - Similar to **Figure 3k**
 - *Top*: Distribution of IHGs (outcome) by the indicated factors (predictor). *P* values determined by χ^2 test.

- *Bottom*: Percentage of FSW who had later HIV seroconversion (outcome) according to study groups (predictor) in the *top* panel. *P* values determined by logistic regression.
 - Groupings in *top* and *bottom* panels are by following predictors: duration of sex work (in years) strata, condom use, clients per week strata, clients per week minus condoms per week strata, behavioral activity score (BAS), direct STI score, indirect STI score, and total STI score.
 - These analyses mirror those in main Fig. 3k but in a larger group of FSWs. Separate hypotheses were tested, i.e., whether the extent of distinct correlates of sex work associated with changes in IHG status. Each comparison has one *P* value. All results are reported, and additional testing for multiple comparisons was not warranted.
 - *Panel c*:
 - Barplots depicting the proportion of each value of the indirect (*top left*), direct (*top right*), and total (*bottom*) STI score (outcomes) by the BAS (predictor). Spearman's correlation coefficient is shown at the top. *P* values determined by a test statistic based on Spearman's product moment correlation coefficient
 - Three separate correlations between distinct correlates of sex work are shown. Additional testing for multiple comparisons was not warranted.
 - *Panel d*:
 - KM plot depicting proportion free of IHG-III or IHG- (outcome) over time by baseline BAS (predictor) in FSW that were IHG-I or IHG-II at baseline. *P* value determined by log-rank test.
 - One *P* value; no correction required.
- Supplementary Fig. 4:
 - *Panels a-d*:
 - All data derived from HIV- FSWs cohort
 - *Panel a*: Similar to **Figure 4a**, except that all data for indicated follow-up windows are shown in boxplots using the first available CD4+ and CD8+ T-cell counts, CD4:CD8 ratio and accompanying BAS data during each 2-year intervals.
 - *Left*: Distribution of IHGs (outcome) within each 2-year intervals (predictor) in 27 FSWs who remained HIV- for at least 10 years. *P* value determined by Fisher's exact test comparing window >0-≤2 vs. >8-≤10.
 - *Middle*: Boxplots of BAS (outcome) in each 2-year intervals (predictor). *P* value determined by linear GEE model with an AR1 correlation structure evaluating the change in BAS over indicated time windows.
 - *Right*: Boxplots of CD4+, CD8+ T-cell counts, and CD4:CD8 ratio (outcomes) in each 2-year intervals (predictors). *P* value determined by linear GEE model with an AR1 correlation structure evaluating the change in log₂ transformed data over indicated time windows.
 - This panel examines three hypotheses and parallels the findings shown in main Fig. 4A: IHGs over time, BAS over time, and clinical metrics over time. A single *P* value per comparison is shown. Additional testing for multiple comparisons was not warranted.
 - *Panel b*:

- Same as **Figure 4b**, left
 - Distribution of IHGs (outcome) at baseline and first available IHG at or after year 4 of follow-up (predictors) in FSW who remained HIV-seronegative for at least 4 years ($n=101$); IHG distribution (outcome) is depicted in the overall group of FSWs and by baseline IHG (predictor). The outcomes are the IHG distributions with the predictors being baseline and first available measurement at year 4. P values determined by χ^2 test.
 - **One P value; no correction required.**
 - *Panel c:*
 - Similar to **Figure 4b**, right, except that all data for indicated timepoints are shown in boxplots
 - Boxplots of BAS, CD4+, CD8+ T-cell counts, and CD4:CD8 ratio (outcomes) in 101 FSWs who remained HIV-seronegative for at least 4 years (predictor); data are for values at baseline and first available measurement on or after the fourth anniversary (predictors). P values determined by Wilcoxon signed-rank test.
 - **This panel examines two hypotheses: BAS over time and clinical metrics over time. Single P value per comparison is shown. Additional testing for multiple comparisons was not warranted.**
 - *Panel d:*
 - *Top:* Median values of CD8+ cell counts (outcomes) by comparison groups indicated in bottom panel (predictor). P values determined by Wilcoxon signed-rank test.
 - *Bottom:* similar to Supplementary **Fig. 4b**, except that the data is shown for baseline and year 6. Distribution of IHGs (outcome) at baseline and first available IHG at year 6 of follow-up (predictor) in FSW who remained HIV-seronegative for at least 6 years ($n=73$); IHG distribution (outcome) is depicted in the overall group of FSWs and by baseline IHG (predictor). The outcomes are the IHG distributions with the predictors being baseline and first available measurement at year 6. P value for overall was determined by χ^2 test. P value for baseline IHG-III comparison group was determined by Fisher's exact test.
 - **This panel examines two hypotheses: comparing CD8+ counts overall and by IHG status, and reconstitution of IHGs according to baseline IHG. Additional testing for multiple comparisons was not warranted.**
- Supplementary Fig. 5:
 - *Panels a-c:*
 - All data derived from subset ($n=43$) of HIV+ FSW
 - *Panel a:*
 - Distribution of post-infection IHGs (outcome) by IHG-III/IV (CD8-CD4 disequilibrium) vs. IHG-I/II (CD8-CD4 equilibrium) at baseline (predictor) in 43 FSWs who have longitudinal data while HIV- and at least one CD4+ and CD8+ T-cell counts measurement within 1 year of HIV seroconversion. No statistical analysis was performed.
 - *Panel b:*
 - Distribution of post-infection IHGs (outcome) by pre-infection IHGs (predictor) in 43 FSW who acquired HIV. Pre-infection IHGs were calculated using the first

available CD4+ T-cell counts and CD4:CD8 ratio before HIV seroconversion for each subject. Post-infection IHGs were calculated using the first available CD4+ T-cell counts and CD4:CD8 ratio after HIV seroconversion for each subject. No statistical analysis was performed.

- *Panel c:*
 - Pre- vs. post-HIV seroconversion changes in IHG status (predictor) within 43 FSWs who acquired HIV with accompanied percent change in CD4+, CD8+ T-cell counts, and ratio (outcomes) between first available measurements before and after HIV seroconversion. CD8+ expansion was classified as suppressed, restrained, and “yes” based on % change in CD8+ T-cell counts of <-20, -20 to 20, and >20, respectively. No statistical analysis was performed.
- Supplementary Fig. 6:
 - Data derived from sooty mangabeys
 - Distribution of IHGs (outcome) in sooty mangabeys stratified by SIV serostatus and age strata (predictors). *P* values determined by Fisher’s exact test.
 - Two separate hypotheses tested; single *P* value per comparison. Additional testing for multiple comparisons was not warranted.
- Supplementary Fig. 7:
 - Data derived from COVID-19 cohort. Note: COVID-19 patients who are IHG-III at baseline were excluded from both analysis.
 - *Left:* IHG-adjusted OR of hospitalization (outcome) stratified by categorical age (predictor) from logistic regression model adjusted by IHG status.
 - *Right:* IHG-adjusted HR of 30-day mortality (outcome) stratified by categorical age (predictor) from Cox proportional hazards model adjusted by IHG status.
 - Two separate hypotheses tested. Point estimates with CI reported. Additional testing for multiple comparisons was not warranted.
- Supplementary Fig. 8:
 - Data derived from RTR cohort
 - Distribution of CD57+CD8+ high/low groups (outcome) by IHG (predictor). Each participant was dichotomized into high/low CD57+CD8+ group on the basis of a majority (i.e., >50%, referred to as CD57+ high) or minority (CD57+ low) of CD57+ cells within the CD8+ population. No statistical analysis was performed.
- Supplementary Fig. 9:
 - Data derived from HIV- SardiNIA cohort
 - The fitted line and 95% confidence band derived from linear regression for (*left to right*) log₂ transformed values of CD4+, CD8+ T-cell counts, and CD4:CD8 ratio (outcome) versus age (predictor) stratified overall (top row) and by sex (middle and bottom rows). *R*² and *P* values for log₂ transformed data versus age determined by linear regression. To evaluate the differences between change in CD4+ T-cell count

across age vs. change in CD8+ T-cell count across age, a linear GEE model with an exchangeable correlation structure of the log₂ cell counts (both CD4 and CD8) as outcomes, and age, cell type (CD4 or CD8), and an interaction term between age and cell type was used. The *P* value for the interaction term shown if the change in CD8+ T-cell counts across age differed from the change in CD4+ T-cell counts across age.

- We examined trajectories of different clinical metrics in overall, males, and females with age. These constitutes three different hypotheses. Additional testing for multiple comparisons was not warranted.
- Supplementary Fig. 10:
 - All data derived from HIV- FSWs cohort.
 - *Panel a*:
 - Distribution of future HIV seroconversion status (outcome) by BAS strata, STI score strata, and IHGs with subgrades (predictors). *P* values determined by χ^2 test.
 - Three separate hypotheses are tested: whether BAS, STI and IHG status associated with HIV serostatus. For BAS and STI analysis single comparisons are performed. For IHG analysis, the hypothesis tested was whether increasing deviations from IHG-I associated with HIV+ serostatus. All significant *P* values remain significant after Bonferroni correction. Additional testing for multiple comparisons was not warranted.
 - *Panel b*:
 - aOR with 95% CI of future HIV seroconversion (outcome) determined using logistic regression by BAS strata, STI score strata, and IHGs with subgrades (predictors). BAS strata are adjusted by STI score strata and IHGs with subgrades. STI score strata is adjusted by BAS strata and IHGs with subgrades. IHGs with subgrades is adjusted by BAS strata and STI score strata.
 - Statistical plan same as panel a but examining the adjusted odds ratio of incident HIV. Three separate hypotheses are tested: whether BAS, STI and IHG status associated with HIV serostatus. For BAS and STI analysis single comparisons are performed. For IHG analysis, the hypothesis tested was whether increasing deviations from IHG-I associated with HIV+ serostatus. The odds ratios were derived from a single model. Additional testing for multiple comparisons was not warranted.
- Supplementary Fig. 11:
 - Data derived from multiple cohorts.
 - Unsupervised hierarchical clustering of SAS-1, MAS-1, IMM-AGE, and age in the FHS-offspring cohort (*left*), San Antonio Family Heart Study (*middle*), COVID-19 cohort (*right*) performed using 1 – Pearson's correlation coefficient as the distance with Ward's D linkage. *P* value determined by correlation test using Pearson's method.
 - All comparisons are within three separate cohorts.
- Supplementary Fig. 12:
 - *Panel a*:

- All data derived from FHS-offspring cohort (Source: dbGaP: phs000007.v30.p11; median age [IQR] years = 66 [60-73]; females: 54%)
 - *Top*: Line plots of SAS-1 and MAS-1 scores (mean \pm SEM) [outcomes] in the FHS cohort stratified by categorical age and sex [predictors]. *P* values determined by linear model comparing males vs. females adjusting for categorical age.
 - *Bottom*: Line plots of SAS-1 and MAS-1 scores (mean \pm SEM) [outcomes] in the San Antonio Family Heart Study stratified by categorical age and sex [predictors]. *P* values determined by linear model comparing males vs. females adjusting for categorical age.
 - Three separate associations are described with a single *P* value per gene signature score in two separate cohorts. There are no inter-cohort comparisons. Additional testing for multiple comparisons was not warranted.
- *Panel b*:
- Boxplots of indicated gene signature scores (outcomes) in meta-analysis of Alzheimer's disease (AD) and other dementia disorders cohorts
 - *Left*: Two sets of control groups [set 1 ($n=249$; median age [IQR] years = 73 [69 - 78]; females: 56%; source: GSE140829) and set 2 ($n= 281$; median age [IQR] years = 72 [68-77]; females: 56%; source: GSE140830)] were pooled together and stratified by indicated age bins (predictors).
 - *Right*: Two sets of disease groups [set 1 includes patients with Alzheimer's disease and mild cognitive impairment ($n=338$; median age [IQR] years = 72 [68-79]; females: 51%; source: GSE140829) and set 2 includes patients one or multiple disorders within the frontotemporal dementia ($n=261$; median age [IQR] years = 66 [60-72]; females: 51%; source: GSE140830) were pooled together and stratified by indicated age bins (predictors).
 - *P* values determined by GEE model with an exchangeable correlation structure for overall comparison of the groups.
 - Three separate gene signature scores were examined and a single *P* value per signature is shown. Additional testing for multiple comparisons was not warranted.
- *Panel c*:
- Boxplots of indicated gene signature scores (outcomes) in meta-analysis of whole blood samples from/of the following groups:
 - *Left*: 518 HIV- participants from Finnish DILGOM cohort stratified into indicated age bins (predictor). (Source: E-TABM-1036). *P* value determined by Kruskal-Wallis test for overall comparison of the age bins.
 - *Middle-Left*: Participants from USA who are HIV-, HIV+ without ART, or HIV+ with ART (predictors). (Source: GSE29429). *P* values determined by GEE model with an exchangeable correlation structure comparing between indicated groups.
 - *Middle-Right*: Participants from UK who are HIV- healthy control, and patients with LTBI or PTB (predictors). (Source: GSE19439, GSE19444). *P* value determined by Wilcoxon rank-sum test between indicated groups.
 - *Right*: Participants from Africa who are HIV- healthy controls, and patients with LTBI or PTB, and HIV+ without ART (predictors). (Source: GSE29429). *P* values between control and HIV+ groups determined by GEE model with an exchangeable correlation structure.

- Four different cohorts are evaluated for levels of three gene signature scores. In each cohort the association of distinct variables with score expression was determined. The differences in the significance values relate to the underlying biology (e.g., age is associated with a decrease in SAS-1, or patients with PTB and HIV have higher antigenic loads). Additional testing for multiple comparisons was not warranted.
- Supplementary Fig. 13:
 - *Panel a:*
 - All data derived from FHS-offspring cohort (Source: dbGaP: phs000007.v30.p11; median age [IQR] years = 66 [60-73]; females: 54%) and all *P* values were determined using χ^2 test for indicated overall comparisons.
 - *Left:* Distribution of SAS-1/MAS-1 profiles (outcome) in participants stratified by sex (predictor).
 - *Middle:* Distribution of SAS-1/MAS-1 profiles (outcome) in participants stratified by indicated age bins (predictor). *P* values were determined for overall differences in proportions of the SAS-1/MAS-1 profiles and the differences in proportion of SAS-1^{high}-MAS-1^{low} or SAS-1^{low}-MAS-1^{high} vs rest across the indicated age bins.
 - *Right:* Distribution of SAS-1/MAS-1 profiles (outcome) in participants stratified by indicated age bins and sex (predictors). *P* values were determined for differences in proportions of the SAS-1/MAS-1 profiles between males vs females within each of the indicated age bins.
 - This series of analysis was hypothesis driven and performed to corroborate findings in Fig. 7c (main text). Overall differences in SAS-1/MAS-1 profiles by sex and age examined individually were highly significant. The stratified analysis was performed to determine the pattern of sex and age combinations remained consistent. Differences between age/sex strata were not examined. Additional testing for multiple comparisons was not warranted.
 - *Panel b:*
 - All data derived from Finnish DILGOM cohort (Source: E-TABM-1036) and *P* values were determined using χ^2 test for indicated overall comparisons. Note: Data presented in this panel is an excerpt from the meta-analysis of datasets described in Supplementary Section 8.3.12.
 - *Left:* Distribution of SAS-1/MAS-1 profiles (outcome) in participants stratified by sex (predictor).
 - *Middle:* Distribution of SAS-1/MAS-1 profiles (outcome) in participants stratified by age strata (predictor). *P* values were determined for overall differences in proportions of the SAS-1/MAS-1 profiles and the differences in proportion of SAS-1^{high}-MAS-1^{low} or SAS-1^{low}-MAS-1^{high} vs. rest across the indicated age strata.
 - *Right:* Distribution of SAS-1/MAS-1 profiles (outcome) in participants stratified by indicated age strata and sex (predictors). *P* values were determined for differences in proportions of the SAS-1/MAS-1 profiles between males vs. females within each of the indicated age strata.
 - This series of analysis was hypothesis driven and performed to corroborate findings in Fig. 7c (main text). Overall differences in SAS-1/MAS-1 profiles by sex and age examined individually were highly significant. The stratified analysis was performed to determine the pattern of sex and age combinations remained consistent.

Differences between age/sex strata were not examined. Additional testing for multiple comparisons was not warranted.

- *Panel c:*
 - Data derived from multiple cohorts
 - Boxplots of indicated gene signature scores (outcomes) in the following groups (predictor):
 - *Left:* CMV+/HIV+ EIC subjects on ART by IHG-I vs. rest.
 - *Middle-left:* HIV- SLE patients by IHG-I vs. rest.
 - *Middle-right:* COVID-19 patients by IHG-I vs. rest.
 - *Right:* COVID-19 patients by IHG status.
 - *P* values for the data shown in first three columns were determined by Welch's t-test for comparison of the IHG-I vs. rest.
 - *P* values for the data shown in last column was determined by ANOVA for overall comparison of the IHG groups.
 - This panel tests three hypothesis in three different cohorts: differences in SAS-1, MAS-1 and IMM-AGE in patients classified as IHG-I vs. rest. Single *P* value per comparison is shown. Since the overall difference in the COVID-19 cohort was significant, a post-hoc analysis was performed comparing IHG-I vs. IHG-II or IHG-II. Additional testing for multiple comparisons was not warranted.
- *Panel d:*
 - All data derived from FHS-offspring cohort (Source: dbGaP: phs000007.v30.p11; median age [IQR] years = 66 [60-73]; females: 54%)
 - KM plots depicting proportion survived (outcome) over 9-year follow-up time in the FHS participants stratified by sex with SAS-1 and MAS-1 profiles by age (*top*, younger; *bottom*, older). *P* values determined by log-rank test.
 - Single *P* value are reported per comparison for SAS-1 and MAS-1. The scores were associated with time to death as a continuous variable after application of an FDR (**Supplementary Table 11a**). Additional testing for multiple comparisons was not warranted.
- *Panel e:*
 - Data derived from 90+ Vitality cohort (source: GSE65218)
 - *Left:* Distribution of SAS-1/MAS-1 profiles [outcome] in nonagenarian ($n=146$, ages: ≥ 90 years; females: 71%) and young ($n=30$, median age [IQR] years = 22.5 [20.2-24.0]; females: 70%) participants stratified by sex [predictor] (Source: GSE65219, $n=176$). *P* value determined by Fisher's exact test for comparing overall distributions of SAS-1/MAS-1 profiles between nonagenarians vs. young.
 - *Right:* KM plots depicting the proportion survived [outcome] over 4 years follow-up in nonagenarian ($n=151$) participants stratified by median based low/high categorical groups of (*left*) SAS-1 and (*right*) MAS-1 scores [predictors]. (Source: GSE65218, ages: ≥ 90 years; females: 70%). *P* values determined by log-rank test.
 - There are two different hypotheses being tested and single *P* value per comparison are reported. Additional testing for multiple comparisons was not warranted.
- *Panel f:*
 - Data derived from multiple sepsis cohorts
 - Distribution of SAS-1/MAS-1 profiles [outcome] in the meta-analysis of patients with severe sepsis due to community acquired pneumonia (CAP) or fecal

peritonitis (FP) admitted to the intensive care unit with either sepsis response signature 1 (SRS1, G1) or sepsis response signature 2 (SRS2, G2) [predictor] (Sources: E-MTAB-4421, median age [IQR] years = 64 [52-75]; females: 46%; E-MTAB-4451, median age [IQR] years = 73.5 [63.5-80.0]; females: 26%; E-MTAB-5273, $n=147$, median age [IQR] years = 66 [52.5-76.0]; females: 50%; and E-MTAB-5274, $n=106$, median age [IQR] years = 71 [62.2-77.0]; females: 36%) and healthy controls. Note: G1 (SRS1) is associated with higher mortality than G2 (SRS2); the SRS1 and SRS2 were defined using baseline samples at admission into the intensive care unit. All the four datasets used for this analysis were from same microarray platform.

- P values were determined using Fisher's exact test comparing differences in proportions of SAS-1/MAS-1 profiles between G1 vs G2 within patients with CAP and FP.
- There are four cohorts being examined. The results of the grouped analysis according to condition (CAP or FP) are reported in Fig. 8d. These analyses were performed to show the results in the individual cohorts. We did not perform cross-cohort comparisons. Additional testing for multiple comparisons was not warranted.
- *Panel g*:
 - Data derived from burn cohort (GSE182616)
 - Distribution of SAS-1/MAS-1 profiles [outcomes] in burn trauma patients longitudinally sampled through hospitalization stratified by total burn surface area $\leq 20\%$ or $>20\%$ [predictors] (Source: GSE182616; median [IQR] age of 39 [30.5-55.0] years). P values were determined using GEE model with an exchangeable correlation structure, binomial family, and ANOVA for changes in proportion of SAS-1^{high}-MAS-1^{low} or SAS-1^{low}-MAS-1^{high} vs. rest over time.
 - Two hypothesis were tested in two separate groups of burn patients, examining whether recovery is associated with changes in SAS-1/MAS-1 profiles. Two P values per hypothesis is shown. Additional testing for multiple comparisons was not warranted.
- *Panel h*:
 - Data derived from sepsis cohort (GSE185263)
 - Distribution of SAS-1/MAS-1 profiles [outcomes] in healthy controls ($n=44$, median [IQR] age = 51 [29-59] years; females = 59%) and hospitalized patients with sepsis ($n=348$, median [IQR] age = 61 [44-72] years; females = 42%) stratified by mortality and SOFA score [predictors] (Source: GSE185263). SOFA categorical bins were defined as having a SOFA score of 0 ($n=95$), score of 1 ($n=43$), score of 2 ($n=57$), score of 3 ($n=39$), score of 4 ($n=29$), score of 5 ($n=20$), score of 6-7 ($n=20$), score of 8-12 ($n=29$), or a score of 13-16 ($n=13$). P values determined by Fisher's exact test.
 - Two separate hypotheses were examined, testing SAS-1/MAS-1 profile distributions according to survival and SOFA status. Single P value per comparison. Additional testing for multiple comparisons was not warranted.
- *Panel i*: pre-CC mice cohort

- Distributions of SAS-1/MAS-1 profiles [outcome] in pre-CC mice by high or low response (predictor] to influenza infection. *P* value determined by Fisher's exact test.
 - Only 1 comparison; no correction required.
- Supplementary Fig. 14:
 - Data derived from SIV– Chinese Rhesus macaques
 - Boxplots of percent PD1+CD4+ (*left*) and PD1+CD8+ (*right*) [outcomes] by age, IHG, and age & IHG [predictor]. *P* values determined by Kruskal-Wallis test across the indicated groups.
 - Two separate hypotheses were tested, examining trait levels by age and IHG status. Single *P* value reported per comparison. Additional testing for multiple comparisons was not warranted.
- Supplementary Fig. 15:
 - Flow cytometry gating strategy for T-B-NK cell antibody panel
 - No statistical analysis was performed.
- Supplementary Fig. 16:
 - Flow cytometry gating strategy for Treg antibody panel
 - No statistical analysis was performed.
- Supplementary Fig. 17:
 - Flow cytometry gating strategy and traits assessment related to the maturation stages of T-cell antibody panel
 - No statistical analysis was performed.
- Supplementary Fig. 18:
 - Flow cytometry gating strategy and traits assessment related to the circulating dendritic cell (cDC) antibody panel
 - No statistical analysis was performed.
- Figure embedded in Supplementary Section 1.1.1: Data derived from SardiNIA cohort
 - Panel top: Group 1
 - Distribution of IHGs (outcome) in the previous analysis of the SardiNIA cohort stratified by age and sex and age (predictor). *P* values determined by χ^2 test.
 - Panel middle: Group 2
 - Distribution of IHGs (outcome) in the updated analysis of the SardiNIA cohort stratified by age and sex and age (predictor). *P* values determined by χ^2 test.
 - Panel bottom: Comparison

- Distribution of IHGs in the previous (Group 1) and updated (Group 2) analysis of the SardiNIA cohort stratified by age. *P* values determined by χ^2 test.
 - These analyses in *panels a to c* were performed to quality control the two set of SardiNIA cohorts analyzed and corroborate the modeling data presented in Fig. 2g (main). Additional testing for multiple comparisons was not warranted.

- Supplementary Note 3 Figure:
 - *Panel a*:
 - Data derived from HIV– UCSD cohort, acute COVID-19, and COVID-19 convalescent (left to right).
 - *Top*: Median CD8+ cell counts, CD4+ cell counts, and CD4:CD8 ratio by CMV serostatus (outcomes) in overall and by IHG subdivided by CD4:CD8 ratio (predictors). *P* values comparing CD8+ T-cell counts by CMV serostatus determined using Wilcoxon rank sum test.
 - *Bottom*: Distribution of CMV serostatus (outcome) in overall and by IHG subdivided by CD4:CD8 ratio (predictors). *P* values determined by χ^2 test.
 - Two separate hypotheses are tested, examining differences in CD8+ counts by IHG status and CMV serostatus by IHG status in three different cohorts. No cross-cohort comparisons were performed. Additional testing for multiple comparisons was not warranted.
 - *Panel b*:
 - Data derived from HIV– UCSD cohort, acute COVID-19, and COVID-19 convalescent (left to right).
 - Distribution of CMV serostatus (outcome) by age strata (predictor). *P* values determined by χ^2 test.
 - In three different cohorts, the association of age and CMV serostatus across age is examined. Cross-cohort comparisons were not made. Single *P* value per cohort is reported. Additional testing for multiple comparisons was not warranted.
 - *Panel c*:
 - Data derived from HIV– UCSD cohort, acute COVID-19, and COVID-19 convalescent (left to right).
 - Line plots depicting the % CMV+ vs. Age strata in those with a CD4:CD8 ratio < 1.75 and those with a CD4:CD8 ratio \geq 1.75. *P* values comparing proportion CMV+ by CD4:CD8 ratio within each age strata were determined by χ^2 test.
 - In three different cohorts, the association of CD4:CD8 ratio with CMV serostatus across age is examined. The focus was on the consistency of the pattern across age and cross-cohort comparisons were not made. Additional testing for multiple comparisons was not warranted.
 - *Panel d*:
 - Data derived from RTR cohort
 - Distribution of CMV serostatus (top) and CD57+CD8+ high/low (bottom) [outcomes] by IHG (left) or IHG subdivided by CD4:CD8 ratio (right) [predictors]. Each participant was dichotomized into high/low CD57+CD8+ trait on the basis of

- a majority (i.e. >50%, referred to as CD57hi) or minority (CD57lo) of CD57+ cells within the CD8+ population. *P* value determined by Fisher's exact test.
- Two separate hypotheses are tested, examining the association of trait levels and CMV serostatus by IHG status. Single *P* value per comparison is shown. Additional testing for multiple comparisons was not warranted.
- *Panel e*:
 - Data derived from HIV– UCSD cohort.
 - Distribution of IHGs (outcome) by CMV serostatus and urine drug test status (predictors). *P* values determined by χ^2 test.
 - Two separate hypotheses were tested, examining differences in IHG distributions by CMV serostatus, after controlling for drug use, and differences in IHG distributions in CMV+ persons with vs. without positive urine drug test. Single *P* value reported for each comparison. Additional testing for multiple comparisons was not warranted.
 - *Panel f*:
 - Data derived from HIV– UCSD cohort, acute COVID-19, and COVID-19 convalescent (left to right).
 - Distribution of IHGs (outcome) by sex and CMV serostatus (predictors). *P* values comparing females vs. males within CMV+ was determined by Fisher's exact test.
 - In three different cohorts/groups of individuals, differences in IHG distributions by sex in CMV+ persons were examined. A single *P* value per comparison is reported. Additional testing for multiple comparisons was not warranted.
- Supplementary Note 4 Figure:
 - Same as **Fig. 4f** except an additional schema about the groups is presented in the right.
 - Distribution of IHGs (outcome) by indicated infectious outcome groups (predictor) in uninfected counterparts of CC-RIX mice infected with Ebola. *P* value determined by Fisher's exact test comparing mice groups lethal vs. resistant to Ebola infection.
 - One *P* value; no correction required.
 - Supplementary Note 5 Figure:
 - *Panel a*:
 - Schema. No statistical analysis was performed.
 - *Panel b*:
 - Data derived from HIV– UCSD cohort
 - Distribution of rs2524054 genotypes (outcome) by IHG subdivided by CD4:CD8 ratio groupings (predictors) in overall (left) and in European Americans only (right). *P* value determined using a trend test in which the number of minor alleles for rs2524054 was used as the dependent variable and the indicated IHG ratio groups as a continuous variable (in the same order as depicted in the figure) in a linear model. The group order shown is based on highest to lowest CD8 counts within each ratio strata.
 - One *P* value; no correction required.

- *Panel c:*
 - Data derived from HIV– UCSD cohort
 - Distribution of CMV serostatus (outcome) by indicated CD4:CD8 ratio groupings (predictor). P values determined by χ^2 test.
 - A single hypothesis was tested, examining whether decreasing ratio values associated with increasing CMV seropositivity status. Additional testing for multiple comparisons was not warranted.
- Supplementary Note 6 figure:
 - *Panel a:*
 - Data derived from SardiNIA cohort.
 - The n, median, and IQR for CD4+, CD8+ T-cell counts, and CD4:CD8 ratio (outcomes) in participants with ages 15-39 (younger) or ≥ 70 (older) years old by IHG (predictors) with the %difference between older and younger (outcome). P values determined by Wilcoxon rank-sum test.
 - Three separate hypotheses are tested, examining the differences in three distinct variables by age according to IHG status. Mechanistic hypothesis driven questions addressed. Additional testing for multiple comparisons was not warranted.
 - *Panel b:*
 - Data derived from SardiNIA cohort
 - Line plots of median CD4+, CD8+ T-cell counts, and CD4:CD8 ratio (left to right; outcome) by age strata (predictor) in overall, males, and females.
 - *Panel c:*
 - Data derived from SardiNIA cohort
 - Line plot of the median CD4:CD8 ratio (outcome) by age strata (predictor) in those with IHG-I or IHG-II.
 - *Panel d:*
 - Data derived from HIV– UCSD cohort
 - Kernel density estimate plot of CD4:CD8 ratio. P value determined by using the Kolmogorov-Smirnov test evaluating the differences in distribution of CD4:CD8 ratio between participants with aged <40 vs. ≥ 40 years old.
 - *Panel e:*
 - Data derived from SardiNIA cohort
 - Kernel density estimate plot of CD4:CD8 ratio. P value determined by using the Kolmogorov-Smirnov test evaluating the differences in distribution of CD4:CD8 ratio between participants with ages <40 vs. ≥ 40 years old.
 - Note: one participant with CD4:CD8 ratio >300 was eliminated from this analysis. The plot was trimmed at CD4:CD8 ratio cutoff of 7. The estimated density for participants aged <40 years old spans only till CD4:CD8 ratio of 6.84. However, for participants aged ≥ 40 years old, the CD4:CD8 ratio spans till 12.13. The tail of density distribution (between CD4:CD8 ratio of 7 to 12.13) in participants aged ≥ 40 years old accounts for only 1.4% of the population and is not shown.
 - Panels d and e show results of a single hypothesis tested in two separate cohorts. Overall P value in differences in distributions (kernel density estimate plots by age) is depicted. Single P value per comparison is shown. Additional testing for multiple comparisons was not warranted.

- Supplementary Note 8 figure:
 - *Panel a*:
 - Data derived from SardiNIA cohort and same as **Fig. 10a**.
 - 75 Immune traits sub-grouped into 19 signatures that classify into four groups, as described in Supplementary Section 11.2.
 - *Panel b*:
 - Data derived from SardiNIA cohort and similar to **Fig. 10b**, except that additional signatures are shown. Boxplots of the indicated immune traits (outcome) by age (<40 and ≥70) within IHG-I and IHG-II and by IHGs (predictor). The trait levels are presented as inverse-normalized residuals after adjusting for sex and/or age using linear regression.
 - *P* values determined by Wilcoxon rank-sum test and corrected for multiple comparisons using conservative Bonferroni method. Instead of using $P < 0.05$, a stringent $P < 1.67 \times 10^{-4}$ (adjusting for multiple comparisons: 75 traits compared x 4 comparisons: IHG-I vs. IHG-III; IHG-II vs. IHG-IV; younger vs. older within IHG-I and younger vs. older within IHG-II) was used to ascribe significance of traits as detailed in the Section 11.2 of the supplementary methods. Detailed methods are as described in Supplementary Section 11.2.

All Supplementary Tables

- *P* values were corrected for multiple comparisons where appropriate if the comparisons were all testing the same hypothesis or sample groups.

References

1. Perneger TV. What's wrong with Bonferroni adjustments. *BMJ* 1998;316:1236-8. PMID: 9553006
2. Savitz DA, Olshan AF. Multiple comparisons and related issues in the interpretation of epidemiologic data. *Am J Epidemiol* 1995;142:904-8. PMID: 7572970
3. Rothman KJ. No adjustments are needed for multiple comparisons. *Epidemiology* 1990;1:43-6. PMID: 2081237
4. Bacchetti P. Peer review of statistics in medical research: the other problem. *BMJ* 2002;324:1271-3. PMID: 12028986
5. Armstrong JS. Significance Tests Harm Progress in Forecasting. *International Journal of Forecasting* 2007;23:321-7. PMID:
6. Cohen J. The earth is round ($p < .05$). *American Psychologist* 1994;49:997–1003. PMID:
7. Gigerenzer G. Mindless statistics. *Journal of Behavioral and Experimental Economics (formerly The Journal of Socio-Economics)* 2004;33:587-606. PMID:
8. Goodman SN. Toward evidence-based medical statistics. 1: The P value fallacy. *Ann Intern Med* 1999;130:995-1004. PMID: 10383371

9. Gardner MJ, Altman DG. Confidence intervals rather than P values: estimation rather than hypothesis testing. *Br Med J (Clin Res Ed)* 1986;292:746-50. PMID: 3082422
10. Lecoutre B, Lecoutre, M.-P. and Poitevineau, J. . Uses, abuses and misuses of significance tests in the scientific community: won't the Bayesian choice be unavoidable? . *International Statistical Review* 2001;69:399–417. PMID:
11. Silva-Aycaguer LC, Suarez-Gil P, Fernandez-Somoano A. The null hypothesis significance test in health sciences research (1995-2006): statistical analysis and interpretation. *BMC Med Res Methodol* 2010;10:44. PMID: 20482841
12. Bottomley MJ, Harden PN, Wood KJ. CD8+ Immunosenescence Predicts Post-Transplant Cutaneous Squamous Cell Carcinoma in High-Risk Patients. *J Am Soc Nephrol* 2016;27:1505-15. PMID: 26563386

REVIEWERS' COMMENTS

Reviewer #1 (Remarks to the Author):

Thank you for revising again. Congratulations once more on this remarkable accomplishment. Of course, there remain many questions, but your data-rich paper will contribute to their resolution.

Reviewer #3 (Remarks to the Author):

I am satisfied with the responses of the authors to nearly all of the comments raised. This is an impressive study, which will contribute to the field and despite some weaknesses (like all studies) will inspire other work that will build upon it.

My remaining concern relates to the availability of data. As i stated this study has benefitedd enormously from public data sets so the authors need to recipricate and also make their data sets publically available. Stating that the data is "available from the corresponding author on reasonable request" is no longer sufficient as an open data sharing process. I think this statement will also be in conflict with the policy of the journal.

NCOMMS-21-36187B – Point-by-point response

Reviewer #1 (Remarks to the Author):

Thank you for revising again. Congratulations once more on this remarkable accomplishment. Of course, there remain many questions, but your data-rich paper will contribute to their resolution.

Response: We thank the reviewer for his/her laudatory comments.

Reviewer #3 (Remarks to the Author):

I am satisfied with the responses of the authors to nearly all of the comments raised. This is an impressive study, which will contribute to the field and despite some weaknesses (like all studies) will inspire other work that will build upon it.

Response: We thank the reviewer for his/her laudatory comments.

My remaining concern relates to the availability of data. As i stated this study has benefited enormously from public data sets so the authors need to reciprocate and also make their data sets publicly available. Stating that the data is "available from the corresponding author on reasonable request" is no longer sufficient as an open data sharing process. I think this statement will also be in conflict with the policy of the journal.

Response: We have expanded the data availability section including the accession links to processed data for newly generated data required to reproduce the findings from our cohorts and also accession links to the publicly available datasets used. The resources to all publicly available datasets are detailed in Supplementary Table 15a and resources to access shared datasets from coauthors are listed in Supplementary Table 15c.